# Overconfidence in climate overshoot

Carl-Friedrich Schleussner[1,2,3 ✉], Gaurav Ganti[1,2,3], Quentin Lejeune[2,3], Biqing Zhu[1,4], Peter Pfleiderer[3,5], Ruben Prütz[2,6,7], Philippe Ciais[4], Thomas L. Frölicher[8,9], Sabine Fuss[2,6,10], Thomas Gasser[1], Matthew J. Gidden[1,3], Chahan M. Kropf[11,12], Fabrice Lacroix[8,9,13], Robin Lamboll[14], Rosanne Martyr[2,3], Fabien Maussion[15,16], Jamie W. McCaughey[11,12], Malte Meinshausen[1,17,18], Matthias Mengel[10], Zebedee Nicholls[1,17,18], Yann Quilcaille[11], Benjamin Sanderson[19], Sonia I. Seneviratne[11], Jana Sillmann[5,23], Christopher J. Smith[1,20,21], Norman J. Steinert[19], Emily Theokritoff[2,3,7], Rachel Warren[22], Jeff Price[22] & Joeri Rogelj[1,7,14]

Global emission reduction efforts continue to be insufficient to meet the temperature goal of the Paris Agreement[1]. This makes the systematic exploration of so-called overshoot pathways that temporarily exceed a targeted global warming limit before drawing temperatures back down to safer levels a priority for science and policy[2–5]. Here we show that global and regional climate change and associated risks after an overshoot are different from a world that avoids it. We find that achieving declining global temperatures can limit long-term climate risks compared with a mere stabilization of global warming, including for sea-level rise and cryosphere changes. However, the possibility that global warming could be reversed many decades into the future might be of limited relevance for adaptation planning today. Temperature reversal could be undercut by strong Earth-system feedbacks resulting in high near-term and continuous long-term warming[6,7]. To hedge and protect against high-risk outcomes, we identify the geophysical need for a preventive carbon dioxide removal capacity of several hundred gigatonnes. Yet, technical, economic and sustainability considerations may limit the realization of carbon dioxide removal deployment at such scales[8,9]. Therefore, we cannot be confident that temperature decline after overshoot is achievable within the timescales expected today. Only rapid near-term emission reductions are effective in reducing climate risks.

The possibility of surpassing and subsequently returning below dangerous levels of global warming has been a topic of discussion for decades[10] with large-scale carbon dioxide removal (CDR) identified early on as playing an important part in this temperature reversal[11,12]. Since the adoption of the Paris Agreement in 2015 the issue has risen to further prominence.

The temperature goal of the Paris Agreement allows for some ambiguity in its interpretation but establishes 1.5 °C of global warming as the long-term upper limit for global temperature increase[13,14]. This means that if 1.5 °C is temporarily exceeded (subsequently referred to as overshoot), a reversal of warming below it is part of meeting the long-term ambition of the Paris Agreement[13]. The Paris Agreement text does not indicate that temperature must stabilize but instead establishes upper limits below which temperatures must peak and may then decline. This understanding is further strengthened when considering other elements of the Paris Agreement. Achieving global net-zero greenhouse gas (GHG) emissions, as implied by Article 4.1 of the Agreement, is expected to lead to declining temperatures[6,13].

Global GHG emission pathways have a central role in informing the development of policy benchmarks in line with the Paris Agreement and are a core part of climate change assessments by the Intergovernmental Panel on Climate Change (IPCC)[2,15]. These assessments categorize pathways principally based on their peak temperature outcome[2,15]. Because a peak and gradual reversal of global warming turns out to be a fundamental feature of Paris-compatible pathways[16], we propose to henceforth categorize pathways in terms of their peak and decline characteristics (Table 1).

Peak and decline pathways are differentiated by the stringency of emission reduction efforts in the near term and up to achieving net-zero $CO_2$ emissions, and the assumed net-negative $CO_2$ emissions in the long term[16]. The former determines the maximum cumulative $CO_2$ emissions of a pathway and thereby approximately the magnitude and time

[1]International Institute for Applied Systems Analysis (IIASA), Laxenburg, Austria. [2]Geography Department and IRITHESys Institute, Humboldt-Universität zu Berlin, Berlin, Germany. [3]Climate Analytics, Berlin, Germany. [4]Laboratoire des Sciences du Climat et de l'Environnement, LSCE, Gif-sur-Yvette, France. [5]Research Unit for Sustainability and Climate Risks, University of Hamburg, Hamburg, Germany. [6]Mercator Research Institute on Global Commons and Climate Change (MCC), Berlin, Germany. [7]Grantham Institute for Climate Change and the Environment, Imperial College London, London, UK. [8]Climate and Environmental Physics, Physics Institute, University of Bern, Bern, Switzerland. [9]Oeschger Centre for Climate Change Research, University of Bern, Bern, Switzerland. [10]Potsdam Institute for Climate Impact Research, Potsdam, Germany. [11]Department of Environmental Systems Science, ETH Zürich, Zürich, Switzerland. [12]Federal Office of Meteorology and Climatology, MeteoSwiss, Zürich, Switzerland. [13]Institute of Geography, University of Bern, Bern, Switzerland. [14]Centre for Environmental Policy, Imperial College London, London, UK. [15]Department of Atmospheric and Cryospheric Sciences, University of Innsbruck, Innsbruck, Austria. [16]School of Geographical Sciences, University of Bristol, Bristol, UK. [17]School of Geography, Earth and Atmospheric Sciences, The University of Melbourne, Melbourne, Victoria, Australia. [18]Climate Resource, Melbourne, Victoria, Australia. [19]Centre for International Climate and Environmental Research, Oslo, Norway. [20]Met Office Hadley Centre, Exeter, UK. [21]School of Earth and Environment, University of Leeds, Leeds, UK. [22]Tyndall Centre for Climate Change Research and School of Environmental Sciences, University of East Anglia, Norwich, UK. [23]Present address: Centre for International Climate and Environmental Research, Oslo, Norway. ✉e-mail: schleussner@iiasa.ac.at

**Table 1 | Conceptual categories of peak and decline emission pathways**

| Pathway category | Temperature characteristics | Emission characteristics (best estimates) |
|---|---|---|
| PD: peak and decline pathways | Pathways that aim to achieve temperature peak and a sustained long-term temperature decline of at least several decades in duration | Emission reductions in all GHGs towards achieving net-zero $CO_2$ emissions, and net-negative $CO_2$ emissions thereafter |
| PD-OS: overshoot pathways | PD pathways establish a target warming level to be achieved at some point in the far future but allow it to be exceeded with high likelihood over the near term in the conviction that warming can be reversed again at a later stage. These pathways typically envision temperature to be kept at the target level upon returning after overshoot | As peak and decline pathways, but rate of emission reduction, carbon budget, timing of net-zero $CO_2$ and amount of net-negative emissions depend on the characteristics of the envisaged overshoot including considerations of climate response uncertainties |
| PD-EP: enhanced protection pathways | PD pathways that aim to keep peak global warming as low as possible and gradually reverse warming thereafter to reduce climate risks. Given the timescales involved for warming reversal, these pathways typically do not reach an ultimate lower target temperature level within the scenario time frame considered | Stringent and rapid GHG emission reduction as much and as early as possible, achieving net-zero $CO_2$ emissions as soon as possible while minimizing residual emissions, and achieving sustainable levels of net-negative $CO_2$ emissions thereafter in order to potentially reach net-zero or net-negative GHGs |

See Extended Data Table 1 for a comparison with categories proposed in the scientific literature.

of peak warming for median climate outcomes[6,16] (Fig. 1a). The latter determines the pace of potential temperature reversal[16]. Both aspects are further dependent on the temporal evolution of non-$CO_2$ emissions.

Several categories of peak and decline pathways have been proposed in the scientific literature[2,17] (Extended Data Table 1). A prominent example is the latest contribution of Working Group III (WGIII) to the Sixth Assessment Report (AR6) of the IPCC, which includes two pathway categories explicitly referring to the term overshoot (Extended Data Table 1). Temperature overshoot pathways are a sub-category in the peak and decline categorization we present here, with the distinguishing characteristic of these pathways being that their intended maximum temperature limit (1.5 °C) is temporarily exceeded.

Although defined in terms of probabilities of temporarily exceeding 1.5 °C, the IPCC AR6 pathway categories frame a possible overshoot concretely: limited overshoot (C1) refers to exceeding the specified limit by up to about 0.1 °C, whereas high overshoot (C2) refers to exceeding it by more than 0.1 °C and up to 0.3 °C (refs. 2,15) (Extended Data Table 1). This seems to suggest that temperature overshoots in these pathway categories are constrained to a few tenths of a degree with high certainty. But this is not the case. These overshoot numbers refer only to median outcomes and substantially higher warming cannot be ruled out as shown below. A strong focus on median outcomes might lead to overconfidence in the risks under overshoot pathways.

In the following, we outline the dimensions of overconfidence in overshoot from emission pathways to adaptation implications (Fig. 1b). We start by exploring the uncertainties in global temperature outcomes and their implications for the required net-negative $CO_2$ emissions to achieve the intended reversal of warming. Based on these insights, we then discuss the consequences for mitigation strategies considering the feasibility and sustainability constraints of deploying gigatonne-scale CDR. Yet, even if global temperatures were in decline, it is an open question if and how this translates into a reversal of climatic impact drivers[6] and subsequent impacts and risks. We provide insights for both long-term regional climate changes and irreversible risks such

as sea-level rise. Finally, we discuss what considering or experiencing temperature overshoot implies for climate change adaptation. Based on this comprehensive perspective, we contend that it is essential to redirect the overshoot discussion towards prioritizing the reduction of climate risks in both the near term and long term and that overconfidence in the controllability and desirability of climate overshoot should be avoided.

## Uncertain climate response and reversal

Peak warming depends on the cumulative $CO_2$ emissions until global net-zero $CO_2$ and the stringency of reductions in non-$CO_2$ GHGs. Achieving net-negative $CO_2$ emissions (NNCE) after peak warming can result in a long-term decline in warming[6]. Most estimates of NNCE consistent with a long-term reversal of warming in peak and decline pathways have focused on median warming outcomes[15]. However, to comprehensively assess overshoot risks and NNCE requirements for warming reversal, uncertainties in the climate response must also be considered. These include uncertainties during the warming phase (for example, high warming outcomes due to amplifying warming feedbacks)[18] and in the long-term state (potential for continued warming post-net-zero $CO_2$ and the response of the climate system to NNCE)[7].

We explore NNCE requirements for an illustrative pathway with the following characteristics (Fig. 2a): (1) it achieves net-zero $CO_2$ around mid-century; (2) limits median peak warming close to 1.5 °C above pre-industrial levels; and (3) requires no NNCE to do so (for the median warming outcome). We use 2,237 ensemble members of the simple carbon cycle and climate model Finite Amplitude Impulse Response (FaIR) v.1.6.2 to estimate the range of physically plausible warming outcomes for this pathway, consistent with the uncertainty assessment of IPCC AR6 (Fig. 2a and Methods). Two groups of plausible futures stand out. The first includes relatively low-risk futures in which warming peaks below 1.5 °C at the time of, or before, net-zero $CO_2$ is achieved (Fig. 2b, bottom left); in these cases, no NNCEs are required. We also identify relatively high-risk futures in which warming exceeds 1.5 °C at the time of net-zero $CO_2$ and continues beyond (Fig. 2b, top right).

For each respective FaIR run, we estimate the NNCE requirement to return warming to 1.5 °C in 2100 (Methods). We find that a need for large NNCE deployment cannot be ruled out because of the heavy-tailed climate response uncertainty distribution[18] (Fig. 2c). The scale of this deployment (interquartile range: 0 to −400 Gt $CO_2$ cumulatively until 2100, or 0 to −10 Gt $CO_2$ yr$^{-1}$ after 2060) is of the same order of magnitude as the spread of deployed NNCE across the scenarios assessed in IPCC AR6 WGIII (Fig. 2c). Although we find that NNCE requirements resulting from a higher-than-average peak warming due to a strong transient climate response dominate, cumulative NNCE until 2100 of up to 200 Gt $CO_2$ (or 5 Gt $CO_2$ yr$^{-1}$, upper 95% percentile, Fig. 2c) could be required to hedge against further warming past net zero[19]. Our results show that a narrow focus on scenario uncertainty and median warming alone is insufficient to assess potential CDR deployment requirements even for merely achieving a stable global mean temperature in the twenty-first century.

CDR requirements here refer to additional carbon removal due to anthropogenic activity in line with the conventions and definitions of the models underlying our assessment. It is important to note that parties to the United Nations Framework Convention on Climate Change use a different definition for defining land-based carbon fluxes, which results in an approximately 4–7 Gt $CO_2$ yr$^{-1}$ difference between national GHG inventories and scientific models that needs to be considered when translating these insights into policy advice[20].

Our simple illustrative approach has several limitations that would benefit from further exploration, including with dedicated state-of-the-art Earth system models (ESMs)[21]. Particularly relevant questions arise around issues of asymmetry in the Earth system response to either positive or negative $CO_2$ emissions[22,23] (Methods).

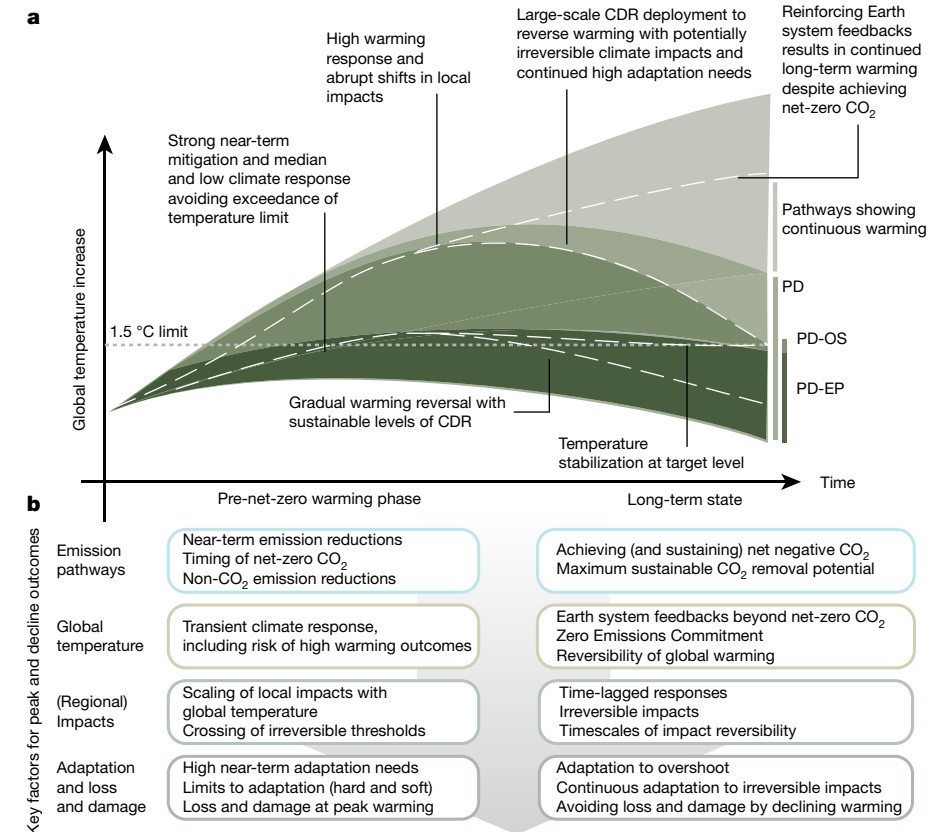

**a**

Strong near-term mitigation and median and low climate response avoiding exceedance of temperature limit

High warming response and abrupt shifts in local impacts

Large-scale CDR deployment to reverse warming with potentially irreversible climate impacts and continued high adaptation needs

Reinforcing Earth system feedbacks results in continued long-term warming despite achieving net-zero $CO_2$

Pathways showing continuous warming

PD

PD-OS

PD-EP

1.5 °C limit

Gradual warming reversal with sustainable levels of CDR

Temperature stabilization at target level

Global temperature increase

Pre-net-zero warming phase

Long-term state

Time

**b**

Key factors for peak and decline outcomes

**Emission pathways**
- Near-term emission reductions
- Timing of net-zero $CO_2$
- Non-$CO_2$ emission reductions
- Achieving (and sustaining) net negative $CO_2$
- Maximum sustainable $CO_2$ removal potential

**Global temperature**
- Transient climate response, including risk of high warming outcomes
- Earth system feedbacks beyond net-zero $CO_2$
- Zero Emissions Commitment
- Reversibility of global warming

**(Regional) Impacts**
- Scaling of local impacts with global temperature
- Crossing of irreversible thresholds
- Time-lagged responses
- Irreversible impacts
- Timescales of impact reversibility

**Adaptation and loss and damage**
- High near-term adaptation needs
- Limits to adaptation (hard and soft)
- Loss and damage at peak warming
- Adaptation to overshoot
- Continuous adaptation to irreversible impacts
- Avoiding loss and damage by declining warming

**Fig. 1 | Illustrative climate outcomes under different conceptual categories of peak and decline pathways. a**, Different classes of pathways with a peak and decline of global mean temperature (see also Table 1). Stylized individual pathways (dashed lines) are highlighted to illustrate the specific impact, adaptation and CDR dimensions associated with the different categories. **b**, An overview of key factors affecting pathway and potential peak and decline outcomes along the impact chain for the warming phase until net-zero $CO_2$ and for the long term beyond net zero. PD, peak and decline pathways; PD-EP, enhanced protection pathways; PD-OS, overshoot pathways.

Owing to the lack of appropriate training data, the response of simple climate models to NNCE is not well constrained. Moreover, the ESMs used to calibrate simple climate models may miss nonlinear responses in the climate system, including abrupt destabilization of natural carbon sinks[24] (for example, permafrost $CO_2$ and $CH_4$ release, peat carbon loss from climate change and degradation or conversion of peatland, extreme fires and drought mortality of forests). We explore permafrost and peatland responses to overshoot below (Fig. 4).

## Relying on CDR

Achieving NNCE requires the deployment of CDR that exceeds residual emissions in hard-to-abate sectors. Pathways assessed by the IPCC WGIII deploy CDR in different ways and to different extents[3]. Scale-up of CDR is most rapid in pathways with the lowest peak warming (low or no overshoot 1.5 °C pathways, C1, Extended Data Fig. 3). Across the ensemble of emission pathways, CDR levels by the end of the century are generally higher in high overshoot (C2) pathways, but the full (5–95%) range is similar to the C1 pathway range. Pathways that keep warming below 2 °C but do not limit warming to 1.5 °C in 2100 (C3) see a substantial CDR ramp-up in the second half of the twenty-first century reaching levels comparable to C1 pathways by 2080 (Extended Data Fig. 3). The total CDR amount deployed in pathways until 2100 depends predominantly on the effective reduction of residual positive $CO_2$ emissions and mitigation of non-$CO_2$ GHGs[17].

In the previous section, we showed how the extent of CDR required to achieve stable temperatures in the twenty-first century might be strongly underappreciated. Here we highlight that there are multiple areas in which current pathways might be overconfident in their assumed use of CDR (Extended Data Table 2). Upscaling of CDR may be constrained considerably[9] by factors such as lack of policy support and business models, technological uncertainty and public opposition (for example, perceived risks of delaying mitigation[25]). Even if technical removal potentials prove to be large, sustainability and equity considerations would limit acceptable deployment scales[8,9]. Insufficient technological readiness may be an important bottleneck, as current removal rates from CDR methods other than afforestation and reforestation are minuscule (about 2 Mt $CO_2$ yr$^{-1}$)[26] and would require a more than 1,000-fold increase by 2050 (ref. 27). Beyond technological concerns, an array of unintended or uncertain permanence issues and system feedback (Extended Data Table 2) might reduce or offset the contribution of CDR to mitigation[26,28].

Squaring these feasibility concerns with the potential need for gigatonne-scale CDR deployment to address climate uncertainty (Fig. 2) is challenging. We argue that deployment pathways that address this challenge should be guided by the principle of harm prevention[29] under enhanced protection pathways (Table 1). This approach requires two complementary actions: (1) reduce gross $CO_2$ emissions rapidly to reduce the total CDR requirements and (2) address feasibility concerns to facilitate the deployment of CDR beyond the achievement of net-zero $CO_2$ to hedge against potentially high warming outcomes.

## Regional climate change reversibility

The proposition of overshoot pathways is that failure to keep warming below a desired temperature limit is acceptable provided global warming is returned below a certain level, that is, 1.5 °C, in the long run. Even if global temperatures are reversed, this is not a given for regional

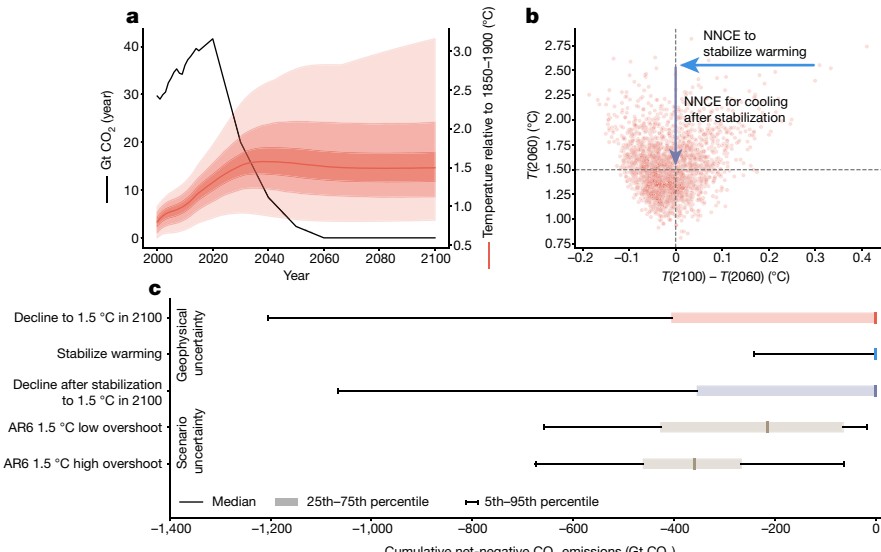

**Fig. 2 | Estimating cumulative NNCE needs when accounting for climate response uncertainty. a**, Net $CO_2$ emissions for the PROVIDE REN_NZCO2 pathway (black line) and the warming outcome uncertainty (derived using FaIR v.1.6.2; Methods). The median warming outcome is the red solid line, with each subsequent plume of varying transparency representing the 25th–75th percentile, 5th–95th percentile, and minimum to maximum ranges, respectively. **b**, Warming at the time of net-zero $CO_2$ (2060) compared with the change in temperature between net-zero $CO_2$ and 2100. **c**, Estimated NNCE to return warming for each peak warming outcome shown in **b** to 1.5 °C in 2100 (Methods). These estimates reflect NNCE implied by geophysical uncertainty of the warming outcome based on the REN_NZCO2 pathway (from top to bottom: NNCE to achieve 1.5 °C in 2100, NNCE to stabilize warming, NNCE for decline after stabilization). For comparison, the scenario uncertainty across the C1 and C2 categories from the IPCC AR6 WGIII report is shown (bottom rows). Note that this scenario uncertainty considers only median estimates of the geophysical response to emissions.

climatic changes. Therefore, understanding the implications of a global temperature overshoot for regional changes is important. Even if global warming is stabilized at a certain level without overshoot, the climate system continues to change as its components keep adjusting and equilibrate[30], with implications for regional climate patterns. The question then becomes what additional imprints on regional climate may originate directly from the overshoot.

Here we explore a unique set of dedicated modelling simulations comparing overshoot and long-term stabilization in two ESMs and find substantial differences in regional climate impact drivers on multi-century timescales (Fig. 3 and Extended Data Fig. 5). We use the results of the NorESM2-LM model following an emission-driven protocol conceptualizing an overshoot of the carbon budget, as well as GFDL-ESM2M simulations following the Adaptive Emission Reduction Approach (AERA) to match a predefined global mean temperature trajectory (Methods and Extended Data Fig. 4). Despite these differences in the modelling protocols, we find some features within the overshoot versus stabilization regional patterns emerging in both modelling simulations, in particular in high northern latitudes as a result of a time-lagged response of the Atlantic Meridional Overturning Circulation (AMOC)[4,31].

In the NorESM2-LM model, we observe a reversal of regional temperature scaling with Global mean surface air temperature (GMST) change for the North Atlantic and adjacent European land regions under overshoot (Fig. 3c), leading to a temporary regional cooling and subsequent regional recovery and warming[32] (Fig. 3e). The pattern in which the North Atlantic cools regionally despite planetary warming is also present in the stabilization scenario but is less pronounced. In the GFDL-ESM2M model, the imprint of overshoot and stabilization on regional climate is less pronounced. But temperature changes associated with a time-lagged AMOC recovery about 100 years after peak warming and to higher levels than in the stabilization scenario are also evident (Fig. 3d,f). We note that these simulations do not include increased Greenland meltwater influx that may suppress a potential AMOC recovery under overshoot[33]. Similarly pronounced features emerge for precipitation in

both models, in particular, related to movements of the Inter-Tropical Convergence Zone in response to changes in the AMOC[4] (Extended Data Fig. 5). Multi-model transient overshoot simulations further corroborate the finding that AMOC dynamics and related changes in regional climate are a dominant feature of overshoot pathways[5,32] (Methods and Extended Data Figs. 7 and 8). They also indicate a continuous warming of the Southern Ocean relative to the rest of the globe as a result of fast and slow response patterns, and changes in regional climate following reduced aerosol loadings (in particular in South and East Asia)[18]. Taken together, our results suggest that regional climate changes cannot be approximated well by GMST after peak warming.

We find substantial long-term imprints of overshoot on regional climate (Fig. 3c,d) that are distinct from transient changes in stabilization scenarios (Extended Data Fig. 6). However, substantial differences in model dynamics (compare Fig. 3e,f) remain. Dedicated multi-model intercomparison experiments are required to further investigate the long-term consequences of overshoot compared with stabilization[21]. We also note the importance of biophysical climate feedback of land-cover changes associated with large-scale land-based CDR deployment (Extended Data Table 2) that could be explored in these experiments.

## Time-lagged and irreversible impacts

For a range of climate impacts, there is no expectation of immediate reversibility after an overshoot. This includes changes in the deep ocean, marine biogeochemistry and species abundance[34], land-based biomes, carbon stocks and crop yields[35], but also biodiversity on land[36]. An overshoot will also increase the probability of triggering potential Earth system tipping elements[33]. Sea levels will continue to rise for centuries to millennia even if long-term temperatures decline[37].

Comprehensively assessing future climate risks under peak and decline pathways requires a focus not only on the (irreversible) consequences of a temporary overshoot but also on the benefits of long-term

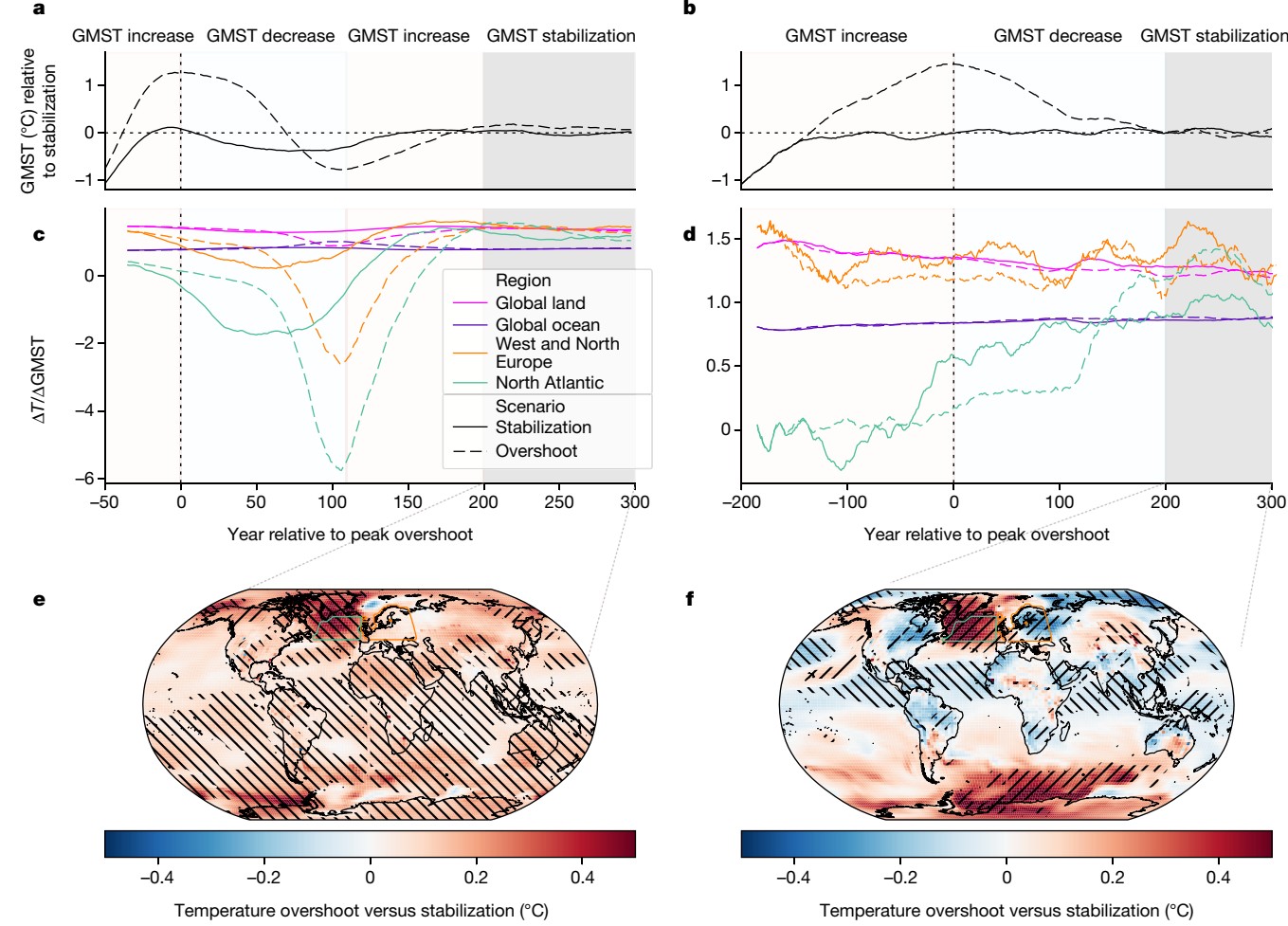

**Fig. 3 | Evolution of regional temperatures before and after overshoot compared with global temperature stabilization.** Results for a carbon budget overshoot protocol with the NorESM model[4] (**a,c,e**) and a global temperature-focused protocol (GFDL-ESM2M)[49] (**b,d,f**). **a,b**, GMST trajectories for dedicated climate stabilization (solid) and overshoot (dashed) scenarios. **c,d**, Temporal evolution of scaling coefficients of annual regional temperatures with GMST for the global land and ocean areas as well as the North Atlantic Ocean (north of 45° N) and Western and Northern Europe (31-year averaged anomalies relative to 1850–1900). **e,f**, Regional differences in annual temperature between overshoot and stabilization scenarios over 100 years of long-term GMST stabilization (grey shaded area in **a,b**). Hatching in **e,f** highlights grid cells in which the difference exceeds the 95th percentile (is below the 5th percentile) of comparable period differences in piControl simulations (Methods).

temperature reversal, compared with stabilization at higher levels. Here we explore the consequences of overshoot in an ensemble of peak and decline pathways (Methods) that achieve net-zero GHGs and thereby long-term temperature decline compared with stabilization at peak warming (by maintaining net-zero $CO_2$).

For global sea-level rise, we find that every 100 years of overshoot above 1.5 °C leads to an additional sea-level rise commitment of around 40 cm by 2300 (central estimate) apart from a baseline of about 80 cm without overshoot (Fig. 4a). For high-risk outcomes, the 2300 sea-level rise commitment could be about three times (95th percentile) above the central estimate[37] (Extended Data Fig. 10). Long-term temperature decline at about 0.03–0.04 °C per decade (broadly consistent with achieving net-zero GHGs) avoids about 40 cm of 2300 sea-level rise (median estimate, 95th percentile about 1.5 m) compared with stabilization at peak warming (Fig. 4b).

A similar pattern emerges for 2300 permafrost thaw and northern peatland warming leading to increased soil carbon decomposition and $CO_2$ and $CH_4$ release (Fig. 4 and Extended Data Fig. 9). The effect of permafrost and peatland emissions on 2300 temperatures increases by 0.02 °C per 100 years of overshoot (best estimate, upper 95% percentile 0.04 °C, Extended Data Fig. 10), whereas achieving long-term declining temperatures would reduce the additional 2300 temperature

increase by a similar order of magnitude. We warn that the diagnosed linear relationship between overshoot length and impact outcome may depend on the set of pathways that it was derived from. The underlying pathways assume overshoots starting from a period of delay in climate action followed by a steady reduction to net-zero GHG emissions implying a similar rate of long-term temperature decline in all pathways. The relationship could be different for more, or less extreme overshoot outcomes.

## Socioeconomic impacts

The severity of climate risks for human systems under overshoot depends markedly on their adaptive capacity[38], as well as the potential transgression of limits to adaptation[39]. An overshoot above 1.5 °C would likely emerge during the first half of the twenty-first century, a period still characterized by comparably low adaptive capacity in large parts of the globe even under optimistic scenarios of socioeconomic development[38]. The coincidence of overshoot and low adaptive capacity can amplify climate risks. This has profound consequences for the ability to achieve climate-resilient and equitable development outcomes under overshoot, in particular, for the most vulnerable countries, communities and peoples.

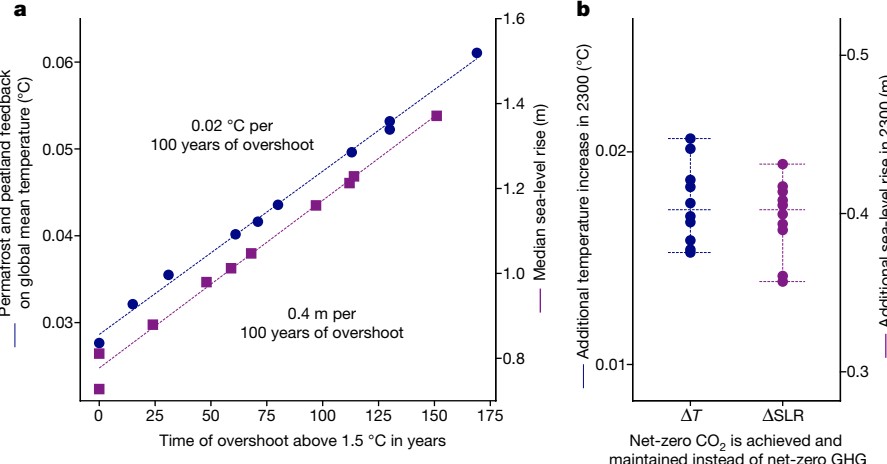

**Fig. 4 | Long-term irreversible permafrost, peatland and sea-level rise impacts of overshoot. a**, Feedback on 2300 global mean temperature increase by permafrost and peatland emissions (blue markers and left axis) and 2300 global median sea-level rise (SLR, purple markers and right axis, from ref. 37) as a function of overshoot duration. Circles (squares) mark results for temperature change (sea-level rise) for individual scenarios from ref. 37. **b**, Additional global mean temperature increase from warming-induced permafrost and peatland emissions and sea-level rise implied by stabilizing temperatures at peak warming (achieving and maintaining net-zero $CO_2$ emissions) compared with a long-term temperature decline resulting from achieving and maintaining net-zero GHGs. Dashed horizontal lines in **b** provide the ensemble median and minimum and maximum range.

Climate impacts on health, ecosystem services, livelihoods and education can leave lasting and intergenerational negative effects on the well-being of people[40] such as climate-related excess deaths linked to heat extremes during an overshoot period. Overshoots might also leave a long-term legacy in the economic performance of countries, particularly those least developed, because of the lasting impacts of climate change on economic growth[41]. Therefore, overshoot entails deeply ethical questions of how much additional climate-related loss and damage people, especially those in low-income countries, would need to endure.

## Adaptation decision-making and overshoot

In contrast to the prominence of overshoot pathways in the mitigation literature, their implications for adaptation planning have not been widely explored[42]. This poses the question of whether the possibility of impact reversal in the long-term future is relevant for adaptation planning today, in comparison with the more imminent threat of near-term climate change and the magnitude of peak warming[43].

Even under the optimistic assumption of nearly full reversibility of a climate impact driver under overshoot, a planning horizon of 50 years or more might be required before prospects of a long-term decline would start to affect adaptation decisions today or in the immediate future (Fig. 5a). Few adaptation plans and policies operate on these timescales: for example, the EU Adaptation Strategy spans three decades, whereas other national adaptation plans have similar or shorter time horizons[44]. Adaptation planning horizons and lifetimes of infrastructure can differ widely (Fig. 5b). At the long end of the planning scale, a hydropower dam may operate for a century or more, yet the management of that dam (and whether management should include flood control as an objective) would occur in concession periods (decades) as well as annual and sub-annual budget cycles (Fig. 5b).

The application of cost–benefit approaches in adaptation measures, and the time scale over which these are assessed, requires decisions on intergenerational equity reflected in the choice of the intertemporal discount rate[45]. Higher discount rates limit the time horizon relevant for economic adaptation decision-making to a few decades (Fig. 5b), in which case adapting to peak warming might always be preferable to adapting to a lower long-term outcome.

It therefore seems that long-term impact driver reversibility after overshoot may be of relevance only in specific cases of adaptation decision-making. A notable exception is adaptation against time-lagged irreversible impacts such as sea-level rise for which overshoots will affect the long-term outlook (Fig. 4). However, as we have shown above, long-term global temperature decline cannot be relied on with certainty. Thus, a resilient adaptation strategy cannot be based on betting on overshoot, and only limiting peak warming can effectively reduce adaptation needs.

Limits to adaptation, both soft and hard, constrain the option space available for adaptation[39]. This includes hard limits in which, for example, adaptation is reliant on ecosystem-based measures that are themselves negatively affected by climate change, as well as soft limits such as lack of resources or governance systems[38]. Transgressing hard adaptation limits, for example, by destroying sensitive ecosystems as a result of unbridled climate change, and high peak warming levels may render these measures unavailable under future warming reversal, reducing the available pool of adaptation measures compared with a no-overshoot case. The risk of transgressing adaptation limits, rather than uncertain prospects of long-term reversibility, seem to be most consequential for adaptation decision-making under overshoot.

## Reframing the overshoot discussion

In this Article, we argue that it is misleading to frame overshoot as an alternative way to achieve a similar climate outcome. We show that several climate impacts in a pre- and post-overshoot world are different, indicating impact reversibility is not a given. Even in cases in which impacts are reversible, the timescales for reversibility may be longer than typical decision horizons for adaptation planning, with peak warming impacts (as opposed to expected longer-term impacts) providing the backdrop for global adaptation needs assessments. From a climate justice perspective, overshoot entails socioeconomic impacts and climate-related loss and damage that are typically irreversible and fall most severely on poor people. This ethical dimension should be explicitly considered when assessing overshoot pathways and the possibilities to limit overshoot risks by near-term emissions reductions.

It has been argued that climate impacts during overshoots could be reduced or masked by the deployment of solar geoengineering (SG) intervention techniques[46] that would temporarily cool the planet. This idea is referred to as peak-shaving. These suggestions, however, make

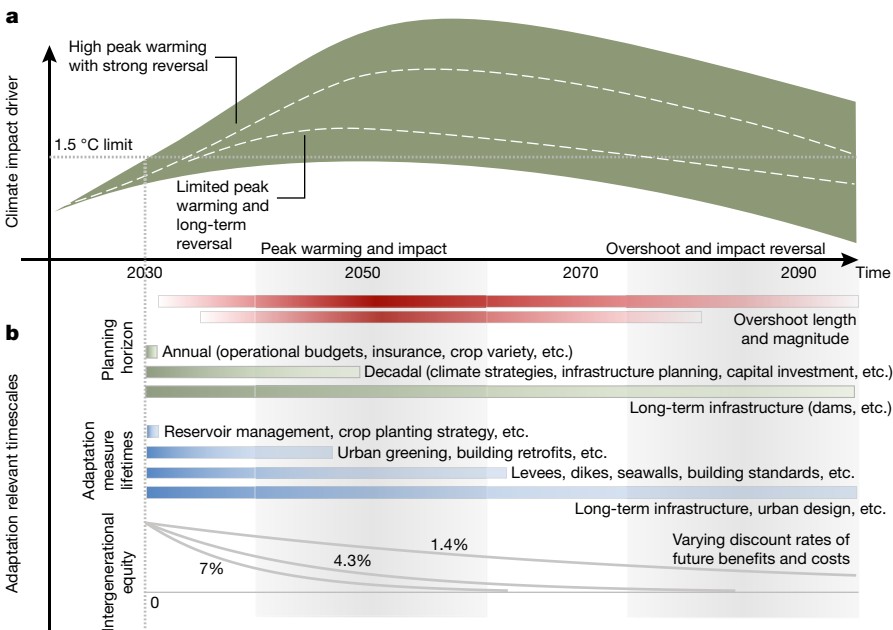

**Fig. 5 | Adaptation-relevant timescales and overshoot. a**, Stylized temporal evolution of a reversible climate impact driver under a peak and decline scenario. Dashed lines indicate a low and high overshoot outcome with median timescales of GMST reversibility typically in line with those from the IPCC AR6 database. **b**, A stylized illustration of adaptation-relevant timescales starting in 2030, including different planning horizons for adaptation planning and lifetimes of individual adaptation measures (horizontal bars, illustrative from years to decades[50], actual time frames vary strongly by context), and the effect of applying discounting (reflecting societal preferences towards intergenerational equity) to future damages and adaptation benefits. We show the effect of discounting for three illustrative discount rates.

strong assumptions about the applicability, effectiveness and governance of SG interventions. Accounting for uncertainties in the physical climate response, and in the evolution of future emissions after SG is deployed, implies that an SG intervention aimed at peak-shaving an overshoot could result in a multi-century commitment of both SG and CDR deployment[23]. Apart from the fundamental concerns about SG deployment in general[47], a peak-shaving discourse is prone to the same overconfidence in reversibility and effectiveness we have conceptualized in this Article.

A central motivation to pursue a long-term temperature draw-down under peak and decline scenarios is to reduce climate impacts. We have shown that this temperature draw-down would be effective in reducing the time-lagged impact emergence over centuries, including sea-level rise and cryospheric changes. The consequences of multi-metre long-term sea level rise will affect coastal regions globally and drawing down global temperatures is important to minimize these long-term risks. Similarly, the probability of crossing irreversible thresholds may remain substantial in the long term unless global mean temperature is brought back down below 1 °C above pre-industrial levels[33].

Based on these insights, we argue for a reframing of the science and policy discourse on overshoot to focus on minimizing climate risks in peak and decline temperature pathways (Table 1). We draw two overarching conclusions:

First, emissions reductions need to be accelerated as quickly as possible to slow down temperature increase and reduce peak warming. Pursuing such an enhanced protection pathway (Table 1) is the only robust strategy to, if not avoid then, at least minimize, far-reaching climate risks over the twenty-first century.

Second, we suggest that there is a need to prepare for an environmentally sustainable CDR capacity to hedge against long-term high-risk outcomes resulting from stronger-than-expected climate feedbacks. We find that this preventive CDR capacity might need to be of the order of several hundred gigatonnes of cumulative NNCE, a scale that might be just about possible within sustainable limits of CDR deployment[9] leaving little room for CDR use for offsetting residual emissions beyond hard-to-abate sectors. This further underscores the importance of very stringent near-term emission reductions to limit long-term risks. Although we argue that the build-up of a preventive CDR capacity is required to hedge against high warming outcomes, this same CDR capacity could, in case high warming outcomes do not materialize, also be deployed to draw down long-term temperatures and thereby reduce climate risks.

The need for a preventive capacity has implications for the design of stringent emission reduction pathways in light of constraints that limit overall CDR deployment. Pathways relying on large amounts of CDR to merely achieve net-zero $CO_2$ often exhaust or exceed sustainability limits[15], leaving little to no room for course corrections in case of high warming outcomes. By contrast, pathways that do not plan for the future development of CDR may fail to build up the technological solutions required to establish a preventive CDR capacity, thereby exposing future generations and, in particular, the most vulnerable communities to risks that could at least be partly hedged against. Incorporating preventive CDR in pathway design requires further reflection, including regarding risks and policy design, but also about how to assign responsibilities and incentivize different actors for providing for this preventive CDR capacity[48].

As a consequence of ever-delayed emission reductions, there is a high chance of exceeding global warming of 1.5 °C, and even 2 °C, under emission pathways reflecting current policy ambitions[1]. Even if global temperatures are brought down below those levels in the long term, such an overshoot will come with irreversible consequences. Only stringent, immediate emission reductions can effectively limit climate risks.

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

## Methods

### Evaluating net-negative $CO_2$ emissions needs reflecting climate uncertainty

In our illustrative analysis, we assess the NNCE for the PROVIDE REN_NZCO2 scenario[51]. The REN_NZCO2 scenario follows the emission trajectory of the Illustrative Mitigation Pathway (IMP) REN from the AR6 of IPCC[52–54] until the year of net-zero $CO_2$ (2060 for this scenario). After the year of net-zero $CO_2$, emissions (of both GHGs and aerosol precursors) are kept constant.

**Deriving climate response metrics.** For this analysis, we derive three metrics that capture different elements of the climate response during the warming phase and the long-term phase:

1. The effective transient response to cumulative emissions (up), or eTCREup: this metric captures the expected warming for a given quantity of cumulative emissions until net-zero $CO_2$.
2. The effective transient response to cumulative emissions (down), or eTCREdown: this metric captures the expected warming or cooling for a given quantity of cumulative net-negative emissions after net-zero $CO_2$. This is a purely diagnostic metric and also incorporates the effects of the effective Zero Emissions Commitment (eZEC).
3. The eZEC: the continued temperature response after net-zero $CO_2$ emissions are achieved and sustained[7]. Here eZEC is evaluated over 40 years (between 2060 and 2100).

To estimate eTCREup (equation (1)), we directly use the warming outcomes reported in the PROVIDE ensemble. The warming outcomes are evaluated using the simple climate and carbon cycle model FaIR v.1.6.2 (ref. 55) in a probabilistic setup with 2,237 ensemble members consistent with the uncertainty assessment of IPCC AR6[56]. Each ensemble member has a specific parameter configuration that allows for the assessment of ensemble member-specific properties such as the climate metrics introduced above across different emission scenarios. This probabilistic setup of FaIR is consistent with the assessed ranges of equilibrium climate sensitivity, historical global average surface temperature and other important metrics assessed by IPCC AR6 WGI (ref. 18).

$$eTCRE_{up}(n) = \frac{T_{2060}(n) - T_{2000}(n)}{\sum_{2000}^{2060} E_{t'}} \qquad (1)$$

where $n$ refers to the ensemble member from FaIR, $t'$ is the time step, $E_{t'}$ is the net $CO_2$ emissions in time step $t'$ and $T_{t'}(n)$ refers to the warming in the time step $t'$ for a given ensemble member.

We need to take a different approach for estimating the second metric ($eTCRE_{down}$) because the PROVIDE REN_NZCO2 does not have NNCE by design. We adapt this scenario with different floor levels of NNCE ranging from 5 Gt $CO_2$ yr$^{-1}$ to 25 Gt $CO_2$ yr$^{-1}$ (Extended Data Fig. 1) that are applied from 2061 to 2100. The scenario is unchanged before 2060. We then calculate the warming outcomes for each of these scenarios applying the same probabilistic FaIR setup and identify the scenario (in this case, REN_NZCO2 with 20 Gt $CO_2$ yr$^{-1}$ net removals) for which all ensemble members are cooling between 2060 and 2100 (Extended Data Fig. 1). This is required to get an appropriate measure of the effect of NNCE emissions. From this adapted scenario, we evaluate the eTCREdown for each ensemble member using

$$eTCRE_{down}(n) = \frac{T_{2100}(n) - T_{2060}(n)}{\sum_{2060}^{2100} E_{t'}} \qquad (2)$$

**Calculating cumulative NNCE for each ensemble member.** Each ensemble member demonstrates a different level of peak warming that depends on eTCRE$_{up}$ (Fig. 2c). We calculate the cumulative NNCE (per ensemble member) that is necessary to ensure post-peak cooling to 1.5 °C in 2100 using

$$NNCE(n) = 0 \quad \text{if } T_{2060}(n) < 1.5 \quad \text{else } \frac{1.5 - T_{2060}(n)}{eTCRE_{down}(n)} \qquad (3)$$

Estimating the effective Zero Emissions Commitment (eZEC) allows us to separate the stabilization and decline components of NNCE. We evaluate eZEC using the post-2060 warming outcome of the original PROVIDE REN_NZCO2 scenario as follows:

$$eZEC(n) = T_{2100}(n) - T_{2060}(n) \qquad (4)$$

We assess the component of NNCE($n$) to compensate for a positive eZEC using

$$NNCE_{stabilization}(n) = 0 \quad \text{if } T_{2060}(n) < 1.5 \quad \text{else } \frac{eZEC(n)}{eTCRE_{down}(n)} \qquad (5)$$

We then assess the component of this NNCE($n$) for cooling after stabilization using

$$NNCE_{decline}(n) = NNCE(n) - NNCE_{stabilization}(n) \qquad (6)$$

**Estimating FaIR v.1.6.2 ensemble member diagnostics for validation.** To evaluate the robustness of our NNCE estimates, we evaluate our FaIR model ensemble against the IPCC AR6 assessments for two key idealized model diagnostics—Equilibrium Climate Sensitivity (ECS) and the Zero Emissions Commitment (ZEC). ECS refers to the steady state change in the surface temperature following a doubling of the atmospheric $CO_2$ concentration from pre-industrial conditions[57]. ZEC is the global warming resulting after anthropogenic $CO_2$ emissions have reached zero and is determined by the balance between continued warming from past emissions and declining atmospheric $CO_2$ concentration that reduces radiative forcing after emissions cease[7].

The ECS is defined[58] as

$$ECS = F_{2\times}/\lambda \qquad (7)$$

where $F_{2\times}$ is the effective radiative forcing from a doubling of $CO_2$ and $\lambda$ is the climate feedback parameter. $F_{2\times}$ and $\lambda$ are parameters that are both used directly in FaIR, and therefore ECS can be calculated for each ensemble member.

We diagnose the ZEC for each ensemble member by performing the bell-shaped ZEC experiments from the Zero Emissions Commitment Model Intercomparison Project (ZECMIP) modelling protocol (corresponding to the B1–B3 experiments in ref. 7). These experiments are $CO_2$-only runs, with a bell-shaped emissions profile with a cumulative emissions constraint (750, 1,000 and 2,000 PgC, respectively) applied over a 100-year time period from the beginning of the simulation period. All non-$CO_2$ forcers are fixed at pre-industrial levels. The ZEC$_{50}$ estimate per ensemble member is then calculated as the difference between the temperatures in years 150 and 100 of the simulation. This ZEC$_{50}$ estimate is purely used for diagnostic purposes and differs from our eZEC estimate, with the latter dependent on the specific characteristics of the emission pathway we apply. However, as the bell experiments approach zero emissions gradually from above and are similar to the actual mitigation scenario emissions profiles, they are good analogues for eZEC.

As expected, following the extended calibration of FaIR against AR6, we find very good agreement between the distribution of ECS and ZEC across members of the FaIR ensemble and the AR6 assessment (compare Extended Data Fig. 2a,b). We also report agreement of the modelled historical warming across the ensemble compared with the observational record (Extended Data Fig. 1d). Based on this evaluation,

we cannot rule out high ECS/ZEC ensemble members that drive the tail of our NNCE distribution (Extended Data Fig. 2c). Yet, we find high NNCE outcomes also materialize for moderate-high ECS and ZEC outcomes.

### Overshoot reversibility for annual mean temperature and precipitation

To investigate the role of stabilization and overshoot for regional revers-ibility, we use simulations of two different ESMs that (1) stabilize GSAT at approximately 1.5 °C of global warming with respect to pre-industrial times and (2) overshoot this level by around 1.5 °C (Extended Data Fig. 4). GFDL-ESM2M[59,60] simulations were performed using the AERA[61], which adapts $CO_2$ forcing equivalent ($CO_2$-fe) emissions successively every 5 years to reach stabilization (1.5 °C) and temporary overshoot (peak warming of 3.0 °C) levels, before returning and stabilizing at 1.5 °C of global warming in the latter case. In this setup, the remaining $CO_2$-fe emissions budget is determined every 5 years based on the relationship of past global anthropogenic warming and $CO_2$-fe emissions simulated by the model. The remaining anthropogenic $CO_2$ emissions or removals are then computed assuming non-$CO_2$ and land use change emissions following the RCP 2.6. Future $CO_2$ emissions are then redistributed fol-lowing a cubic polynomial function, constrained to smoothly reach any given temperature level. Details for the stabilization case are given in the AERA model intercomparison simulation protocol[62] and analysis[49].

Simulations using NorESM2-LM[63] were performed following ideal-ized emission trajectories, including phases of positive and negative $CO_2$ emissions[4]. These simulations are emission-driven, meaning atmos-pheric $CO_2$ concentrations change in reaction to both $CO_2$ emissions and exchanges between the atmosphere and ocean or land. The only applied forcing is $CO_2$ emissions into the atmosphere, whereas land use and non-$CO_2$ GHG forcings remain at pre-industrial levels. The idealized cumulative emission trajectories adhere to the ZECMIP protocol[64]. These emissions are represented as bell-shaped curves, with 50 years of increasing emissions followed by 50 years of decreasing emissions. Negative cumulative emission trajectories follow a similar pattern but with a negative sign. The reference stabilization simulation has cumula-tive carbon emissions of 1,500 Pg during the first 100 years followed by zero emissions for 300 years. The reference simulation reaches global warming levels of approximately 1.7 °C in the long term. NorESM2-LM has a low transient climate response to cumulative emissions (TCRE) of 1.32 K (Eg C)$^{-1}$. For the overshoot simulation, the emission trajec-tory involves cumulative carbon emissions of 2,500 Pg over the first 100 years, following the same emissions profile as the reference sce-nario but with higher emissions rates. It is followed by the application of CDR (in this case assumed as direct air capture) removing 1,000 Pg of cumulative carbon over the period of another 100 years. After nega-tive emissions cease, it follows an extended phase of 200 years of zero emissions, such that the amount of cumulative carbon emissions is identical to the reference simulation for that period.

In both experimental protocols, non-$CO_2$ forcings, including aero-sols, are the same for the stabilization and overshoot scenarios. We thus find the experiments well suited to explore the long-term imprint of overshoots on regional climate compared with long-term climate stabilization 200 years after peak warming.

We note that none of the two protocols includes land cover changes beyond the reference pathway. This points to an implicit assumption that the additional CDR in these simulations is achieved using technical options with little to no land footprint such as Direct Air Capture with CCS (Extended Data Table 2). If the amount of CDR was to be achieved using land-based CDR methods, however, we would expect pronounced biophysical climate effects from the land cover changes alone[65]. The regional climate differences resulting from different CDR strategies should be explored in future modelling efforts.

**Regional averaging.** We compute spatially weighted regional aver-ages for land or ocean regions following IPCC AR6 regions. WNEU

corresponds to land grid cells in western central Europe (WCE) and northern Europe (NEU). NAO45 corresponds to ocean grid cells in the North Atlantic region above 45° N (see encircled area in Fig. 3e,f). AMZ and WAF are land regions.

**Scaling with GMST.** In Fig. 3 (Extended Data Fig. 5), we show surface air temperature (tas) anomalies (absolute precipitation anomalies, respectively) divided by 31-year smoothed GMST anomalies for dif-ferent regions. Anomalies are calculated with respect to 1850–1900.

**Period differences and statistical significance.** When compar-ing period averages between two scenarios (Fig. 3) or at different times in the same scenario (Extended Data Figs. 6–8), we compare the magnitude of the difference with random period differences of the same length in piControl simulations. If the difference exceeds the 95th percentile (or is below the 5th percentile) of differences found in piControl simulations, we consider the difference as sta-tistically significant outside of internal climate variability. When $n$ runs are available for the comparison of period averages, we select sets of $2n$ random periods and compute the difference between the first half and the second half of these random sets to mimic ensemble differences.

**CMIP6 analysis.** We analyse climate projections for the SSP5-34-OS and the SSP1-19 scenarios by 12 ESMs of the Coupled Model Intercom-parison Project Phase 6 (ref. 66): CESM2-WACCM, CanESM5, EC-Earth3, FGOALS-g3, GFDL-ESM4, GISS-E2-1-G, IPSL-CM6A-LR, MIROC-ES2L, MIROC6, MPI-ESM1-2-LR, MRI-ESM2-0 and UKESM1-0-LL.

We smooth the GMST time series by applying a 31-year running aver-age. In each simulation run, we identify peak warming as the year in which this smoothed GMST reaches its maximum. Next, we select the years before and after peak warming in which the smoothed GMST is closest to −0.1 K and −0.2 K below peak warming. There is a substantial, model-dependent asymmetry in the average time between the rate of change in GMST before and after peak warming (see ref. 5 for an over-view). In each run, we average yearly temperatures and precipitation for the 31 years around the above-described years of interest. Finally, for each ESM, these 31-year periods are averaged over all available runs of the ESM and an ensemble median for the 12 ESMs is computed for the displayed differences.

### 2300 projections for sea-level rise, permafrost and peatland

We project sea-level rise, permafrost and peatland carbon emissions with two sets of scenario ensembles as documented in ref. 37. Both sets of scenarios stabilize temperature rise below 2 °C, with one set of sce-narios achieving and maintaining the net-zero GHG emission objective of the Paris Agreement and the other set achieving net-zero $CO_2$ emis-sions only. Sea-level rise projections are taken from ref. 37, based on a combination of a reduced-complexity model of global mean temper-ature with a component-based simple sea-level model to evaluate the implications of different emission pathways on sea-level rise until 2300. We project carbon dynamics for permafrost and northern peatlands for the aforementioned scenario set using the permafrost module of the compact ESM OSCAR[67] and a peatland emulator calibrated on pre-viously published peatland intercomparison project[68]. The forcing data used to drive the permafrost and peatland modules are GMST change and the atmospheric $CO_2$ concentration change relative to pre-industrial levels. First, we simulated the $CO_2$ fluxes and $CH_4$ fluxes from both permafrost and northern peatlands (see Extended Data Fig. 9 for the responses of individual components). Next, we computed the net climate effects of these two systems using the GWP* following the method described in ref. 68. We use the following equation to derive the $CO_2$-warming-equivalent emissions ($E_{CO_2\text{-we}}$) of the $CH_4$ emissions, taking into account the delayed response of temperature to past changes in the $CH_4$ emission rate:

$$E_{CO_2\text{-we}^*} = GWP_H \times \left( r \times \frac{\Delta E_{CH_4}}{\Delta t} \times H + s \times E_{CH_4} \right) \qquad (7)$$

where $\Delta E_{CH_4}$ is the change in the emission rate of $E_{CH_4}$ over the $\Delta t$ preceding years; $H$ is the $CH_4$ emission rate for the year under consideration; $r$ and $s$ are the weights given to the impact of changing the $CH_4$ emission rate and the impact of the $CH_4$ stock. Following ref. 68, we use $\Delta t = 20$. Because of the dependency on the historical trajectory of the emission and carbon cycle feedback, the values of $r$ and $s$ are scenario-dependent. Here we use $r = 0.68$ and $s = 0.32$ (the values used in ref. 68 for RCP2.6), with $H = 100$ years, $GWP_{100}$ of 29.8 for permafrost and $GWP_{100}$ of 27.0 for peatland[18].

We then estimate the global temperature change ($\Delta T$) due to permafrost and peatland $CO_2$ and $CH_4$ emissions as the product of the cumulative anthropogenic $CO_2$-we emissions from permafrost and northern peatlands and the TCRE:

$$\Delta T_{\text{permafrost\&peatland}} = TCRE \times \left( \sum_{1861}^{2300} (E_{CO_2,2300} - E_{CO_2,\text{pre}}) + \sum_{1861}^{2300} (E_{CO_2\text{-we}^*,2300} - E_{CO_2\text{-we}^*,\text{pre}}) \right) \qquad (8)$$

where $E_{CO_2,2300}$ and $E_{CO_2,\text{pre}}$ are $CO_2$ emission rates from permafrost and northern peatlands in 2300 and in the pre-industrial era, respectively; $E_{CO_2\text{-we}^*,2300}$ and $E_{CO_2\text{-we}^*,\text{pre}}$ are $CO_2$-we* due to permafrost and northern peatland $CH_4$ emissions in 2300 and in the pre-industrial era, respectively. For TCRE, we take the median value of 0.45 °C per 1,000 Gt $CO_2$ (ref. 18).

## Data availability

The PROVIDE v.1.2 scenario data used for Fig. 2 is available at Zenodo[69] (https://doi.org/10.5281/zenodo.6963586). The data underlying the GFDL-ESM2M and NorESM2-LM simulations included in Fig. 3 and Extended Data Figs. 5 and 6 are available at Zenodo[70] (https://doi.org/10.5281/zenodo.11091132 and https://doi.org/10.11582/2022.00012). Data required to reproduce Extended Data Figs. 7 and 8 can be found at https://esgf-data.dkrz.de/search/cmip6-dkrz/. Data required to reproduce Fig. 4 and Extended Data Figs. 3, 4, 9 and 10 are included in the code repository.

## Code availability

The analysis was performed with Python and spatial projections rely on the cartopy package. The scripts to replicate Figs. 2–5 are available at Zenodo[71] (https://doi.org/10.5281/zenodo.13208166).

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

**Acknowledgements** We acknowledge support from the Horizon 2020 research and innovation programmes of the European Union under grant agreement no. 101003687 (PROVIDE). G.G. acknowledges support from the Bundesministerium für Bildung und Forschung (BMBF) under grant agreement no. 01LS2108D (CDR PoEt). T.G. also acknowledges support from the Horizon 2020 and Horizon Europe research and innovation programmes of the European Union under grant agreement nos. 773421 (Nunataryuk) and 101056939 (RESCUE). J.S. is funded by the German Research Foundation (DFG) under Excellence Strategy of Germany—EXC 2037:CLICCS—Climate, Climatic Change, and Society—project no. 390683824, contribution to the Center for Earth System Research and Sustainability (CEN) of Universität Hamburg. The GFDL ESM2M simulations were conducted at the Swiss National Supercomputing Centre. B.S. acknowledges support from the Research Council of Norway under grant agreement no. 334811 (TRIFECTA).

**Author contributions** C.-F.S., Q.L. and J.R. conceived the study. C.-F.S. designed the study and wrote the first draft with most of the contributions from Q.L., G.G. and J.R.; J.R. and C.-F.S.

developed the pathway classification and designed Fig. 1, Table 1 and Extended Data Table 1 with support by G.G. The global climate response section, including the analysis underlying Fig. 2 and Extended Data Figs. 1 and 2, was led by G.G. and supported by Z.N., C.J.S., R.L., C.-F.S. and J.R. The section on CDR, including Extended Data Table 2 and Extended Data Fig. 3, was led by R.P. with support from S.F., C.-F.S., M.J.G. and J.R. The section on climate change reversibility, including Fig. 3 and Extended Data Figs. 4–8, was led by P.P. with support from N.J.S., T.L.F., F.L., B.S. and C.-F.S.; F.L. conducted the GFDL ESM2M overshoot and stabilization simulations supported by T.L.F. The analysis underlying the section on time-lagged impacts was led by B.Z. supported by M. Mengel, T.G. and P.C. with inputs from R.W., J.P., F.M. and C.-F.S. The section on adaptation decision-making was led by C.M.K., J.W.M., E.T. and R.M. with inputs from J.S. and C.-F.S.; S.I.S., Y.Q. and M. Meinshausen provided inputs on the conceptualization of the entire Article. All authors contributed to the writing of the paper.

**Competing interests** The authors declare no competing interests.

**Additional information**
**Correspondence and requests for materials** should be addressed to Carl-Friedrich Schleussner.

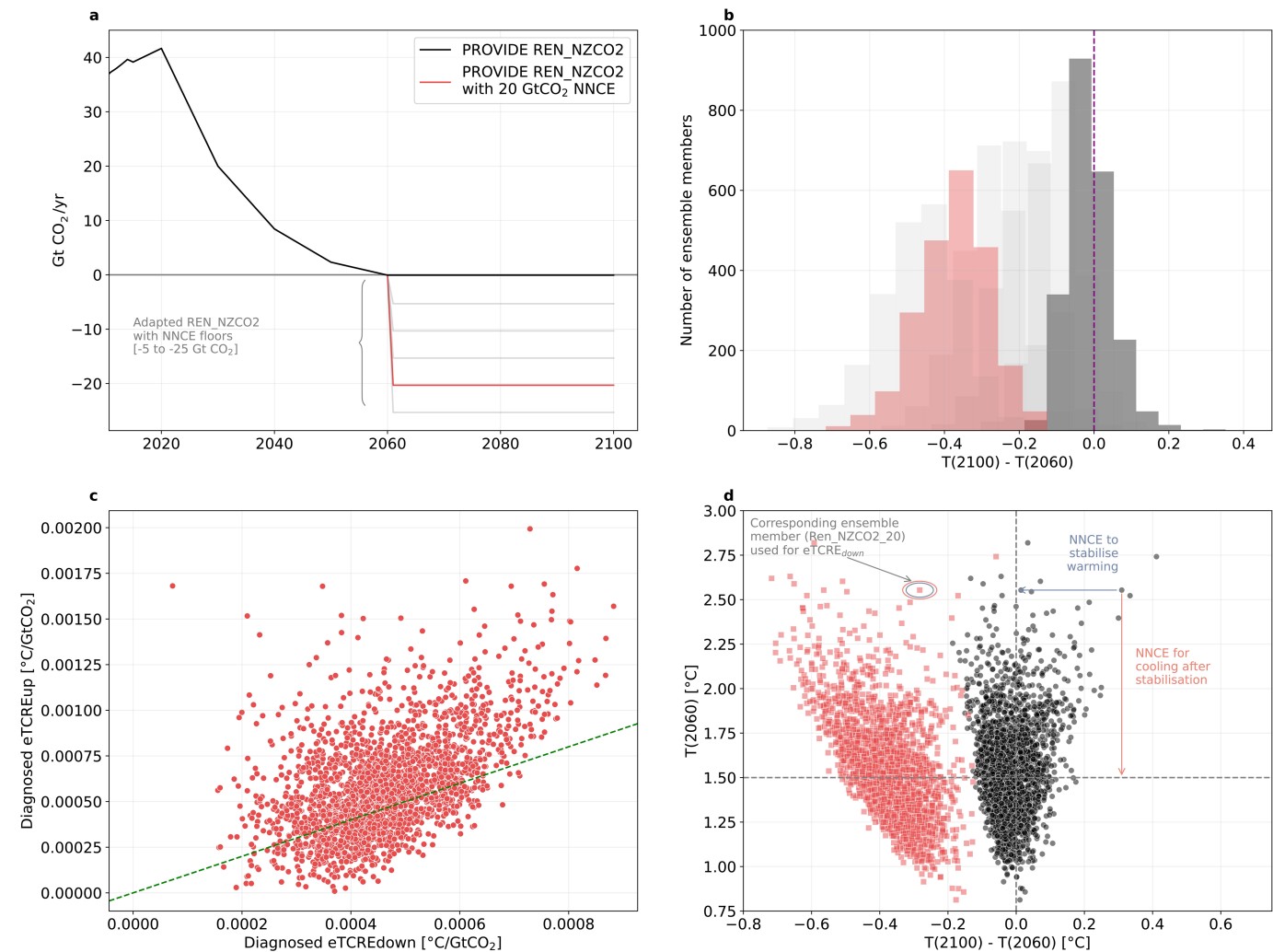

**Extended Data Fig. 1 | Method to derive net-negative CO2 emissions under climate uncertainty for PROVIDE REN_NZCO2. a**, The original PROVIDE REN_NZCO2 scenario (black) and the adapted PROVIDE REN_NZCO2 scenarios with different levels of net-negative CO$_2$ emissions. **b**, The difference between 2100 warming and 2060 warming across the scenarios with the original REN_NZCO2 in black and the adapted REN_NZCO2_20 with 20 Gt CO$_2$ highlighted in red. Estimates to the right of the purple line indicate ongoing warming after 2060. **c**, Diagnosed eTCREup and eTCREdown (estimated from PROVIDE REN_NZCO2_20), **d**, Cooling between 2100 and 2060 versus warming in 2060 for PROVIDE REN_NZCO2 and PROVIDE REN_NZCO2_20.

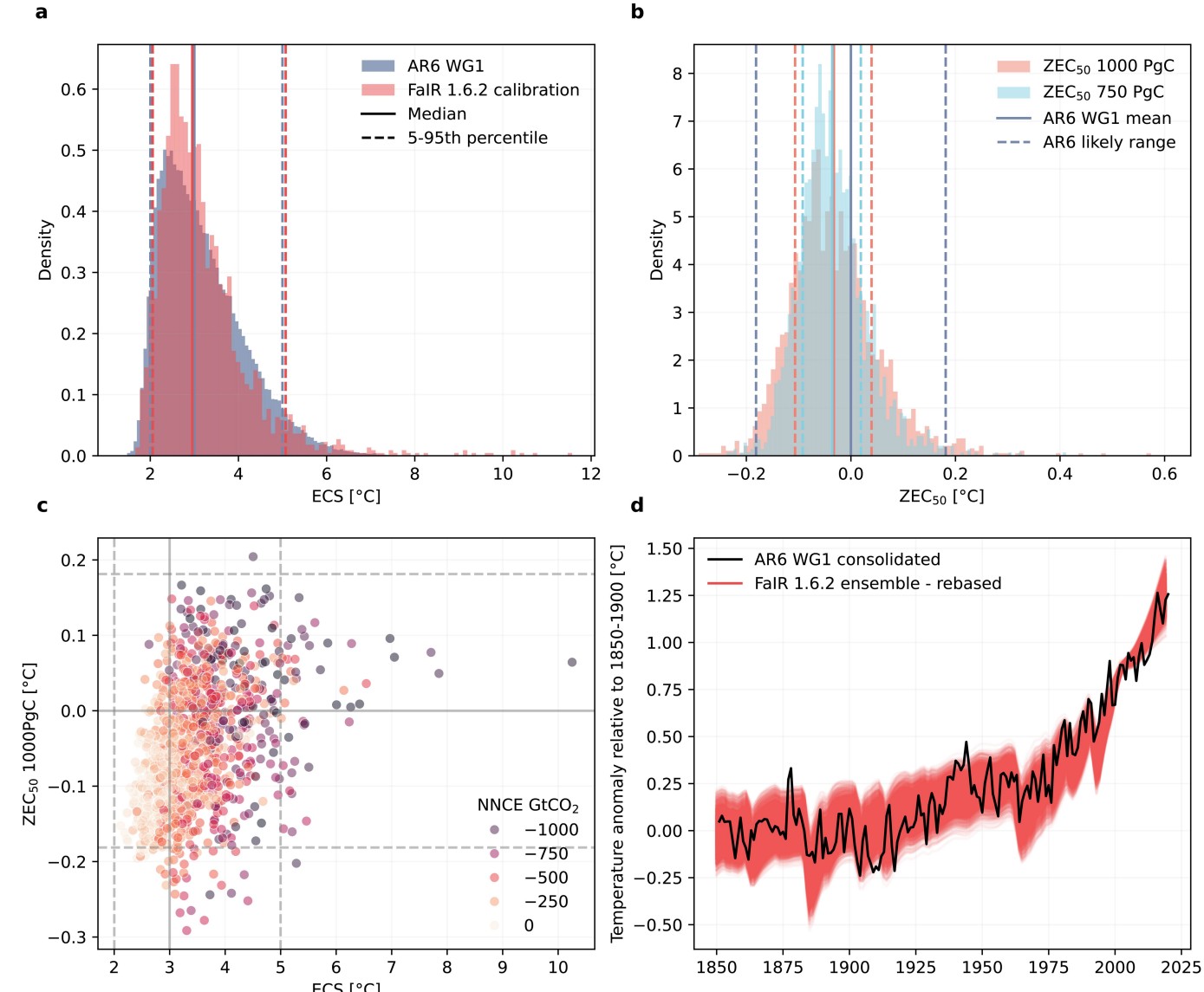

**Extended Data Fig. 2 | FaIR v1.6.2 ensemble diagnostics consistent with AR6 WG1 assessment. a**, Equilibrium Climate Sensitivity (ECS), **b** Zero Emissions Commitment (ZEC) over a 50 year period after CO2 emissions reach zero, **c** High ZEC and ECS drive high net-negative CO2 emissions estimates in ensemble members. Solid and dashed horizontal (vertical) lines indicate the median and 5–95% for ZEC (ECS) distributions as in panel **a**,**b**, respectively. **d**, Consistency of FaIR ensemble members (individual members shown) with the consolidated AR6 WG1 historical warming time series.

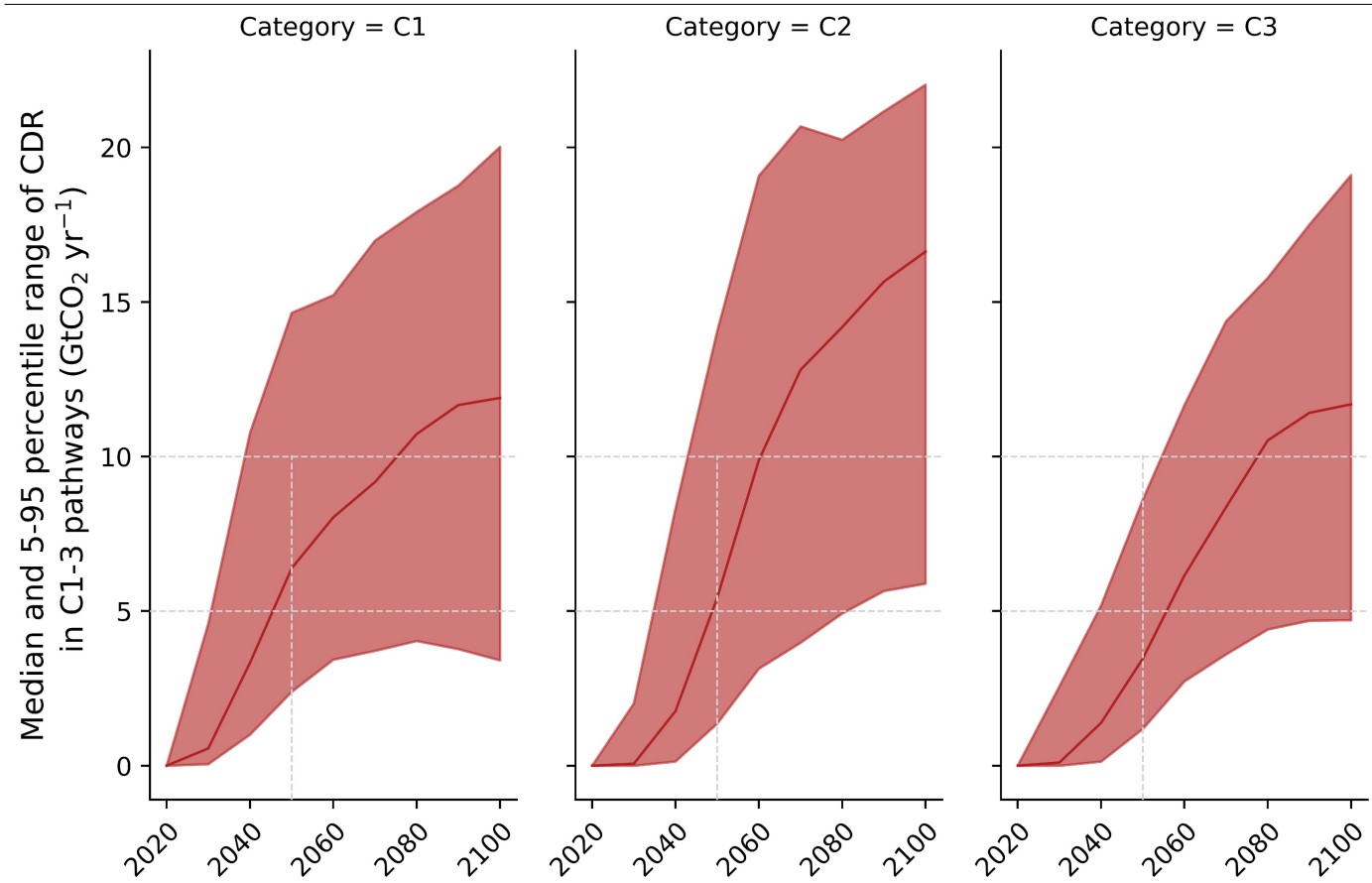

**Extended Data Fig. 3 | Median carbon dioxide removal ranges in AR6 for 2020–2100 across C1-3 with 5–95 percentile ranges.** The figure includes BECCS, DACCS, enhanced weathering, net-removal from AFOLU, and 'other' CDR. Net-removal from AFOLU is used as conservative proxy for land use sequestration to account for reporting inconsistencies for this variable.

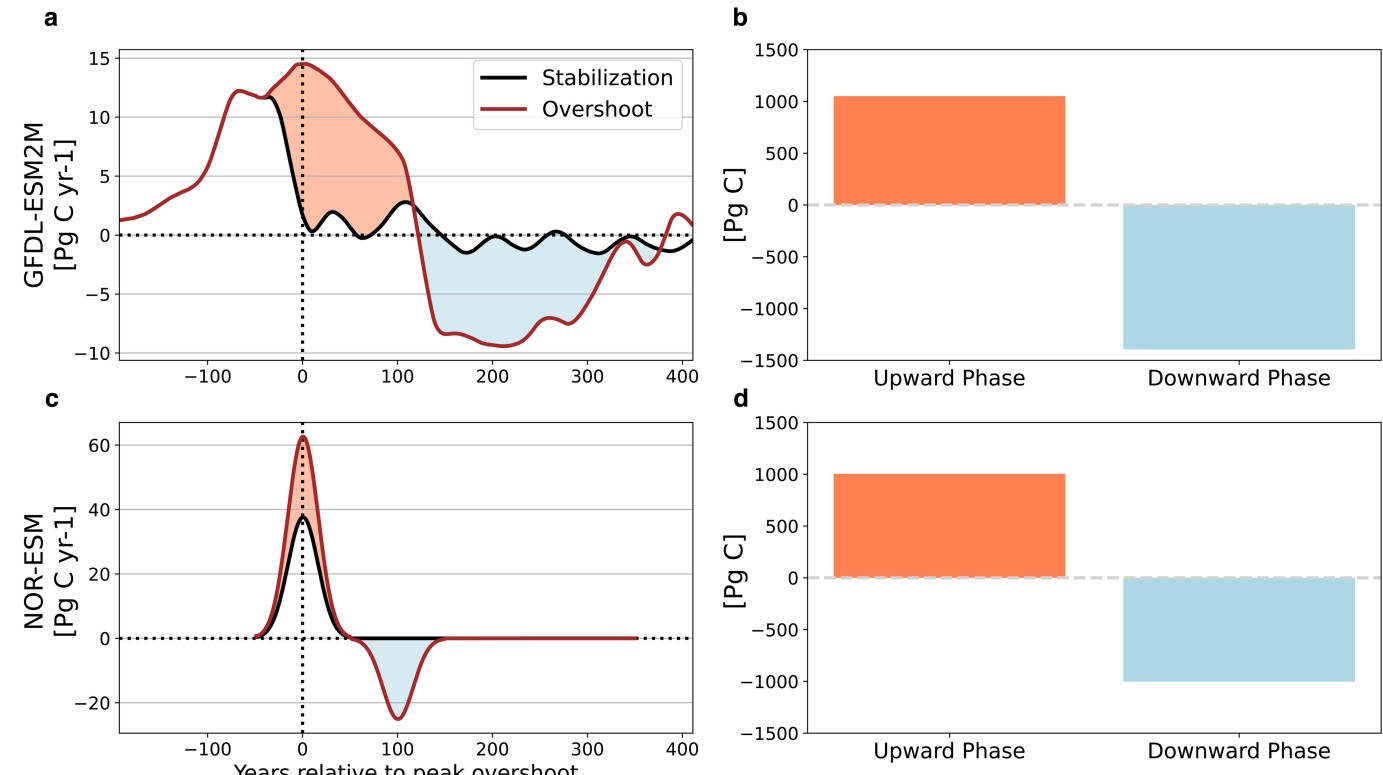

**Extended Data Fig. 4 | CO₂fe emissions in overshoot versus stabilisation experiments. a,c** show transient 31-year mean CO$_2$fe emission trajectories for the GFDL-ESM2M and NorESM experiments, respectively. **b,d** total cumulative carbon budget difference between the overshoot and stabilisation experiments for the GFDL-ESM2M and NorESM experiments during the upward (orange) and downward (blue) phases also highlighted in **a,c**.

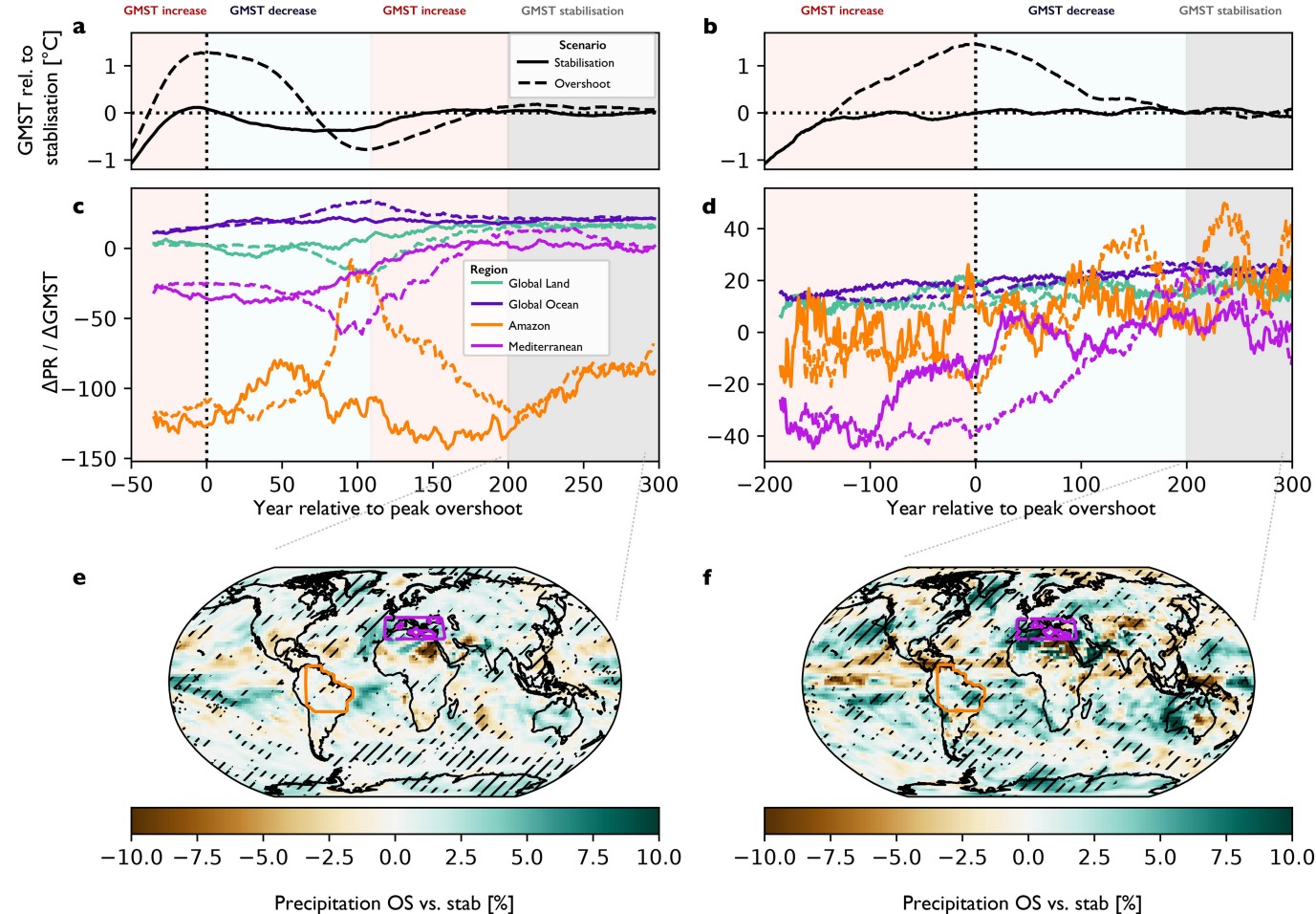

**Extended Data Fig. 5 | Evolution of regional precipitation before and after overshoot compared to global temperature stabilisation. a,c,e** show results for the NorESM Earth System Model, **b,d,f** for GFDL-ESM2M. **a,b** Global mean surface air temperature (GMT) trajectories for dedicated climate stabilisation (solid) and overshoot (dashed) scenarios. **c,d** temporal evolution of regional scaling coefficients of absolute annual precipitation changes with GMT for the global land and ocean areas as well as the Amazon and the Mediterranean region (31-year averaged anomalies relative to 1850-1900). **e,f** regional differences in annual precipitation between overshoot and stabilisation scenarios over hundred years of long-term GMT stabilisation (grey shaded area in panels **a,b**, hatching highlights grid-cells where the difference exceeds the 95th percentile (is below the 5th percentile) of comparable period differences in piControl simulations (see Methods).

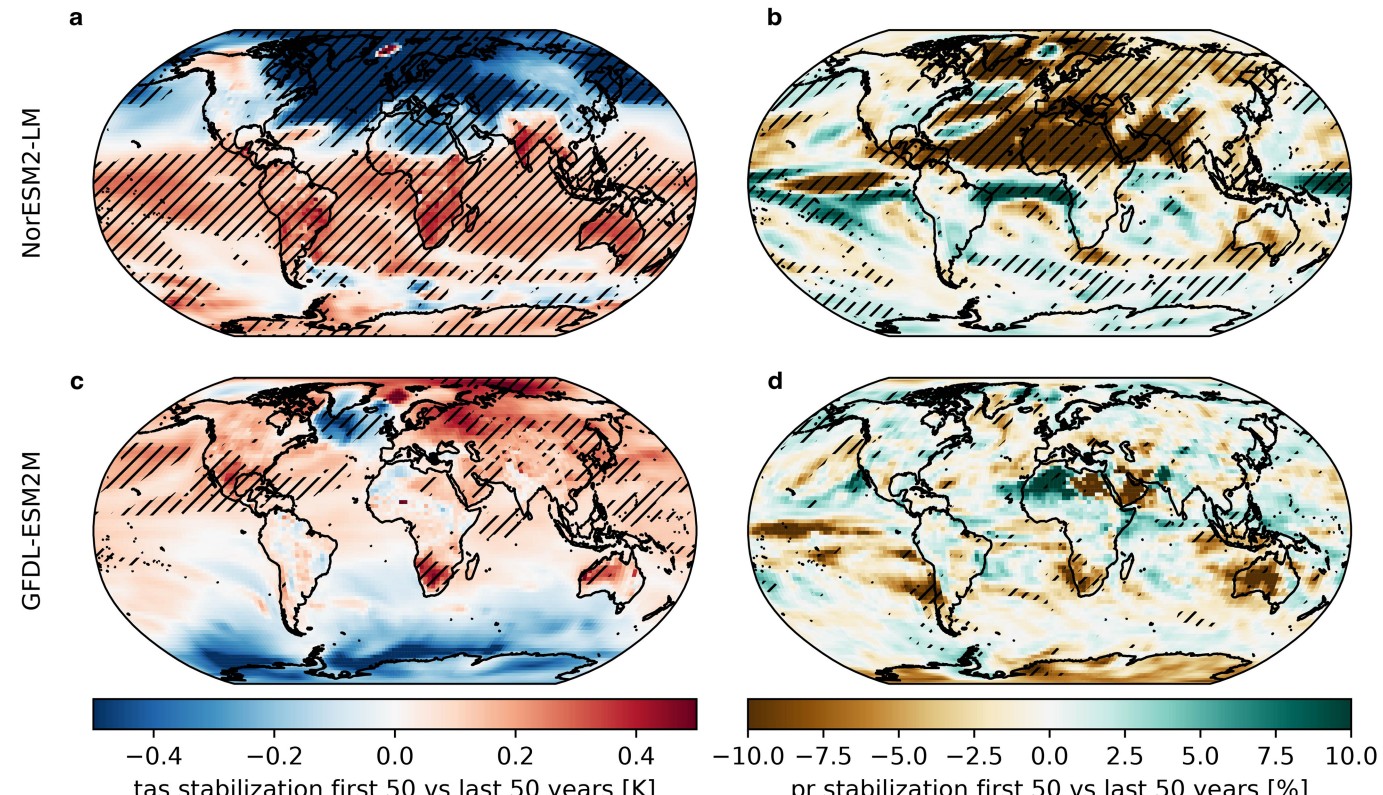

**Extended Data Fig. 6 | Transient regional differences in a GMT stabilisation scenario. a,b** show results for NorESM, **c,d** for GFDL-ESM2M, **a,c** for annual temperature over the first 50 years of GMT stabilisation vs. the last 50 years (compare Fig. 3a). Negative values mean the first period is cooler than the second. **c,d** like **a,c** but for annual precipitation. Hatching highlights grid-cells where the difference exceeds the 95th percentile (is below the 5th percentile) of comparable period differences in piControl simulations (see Methods).

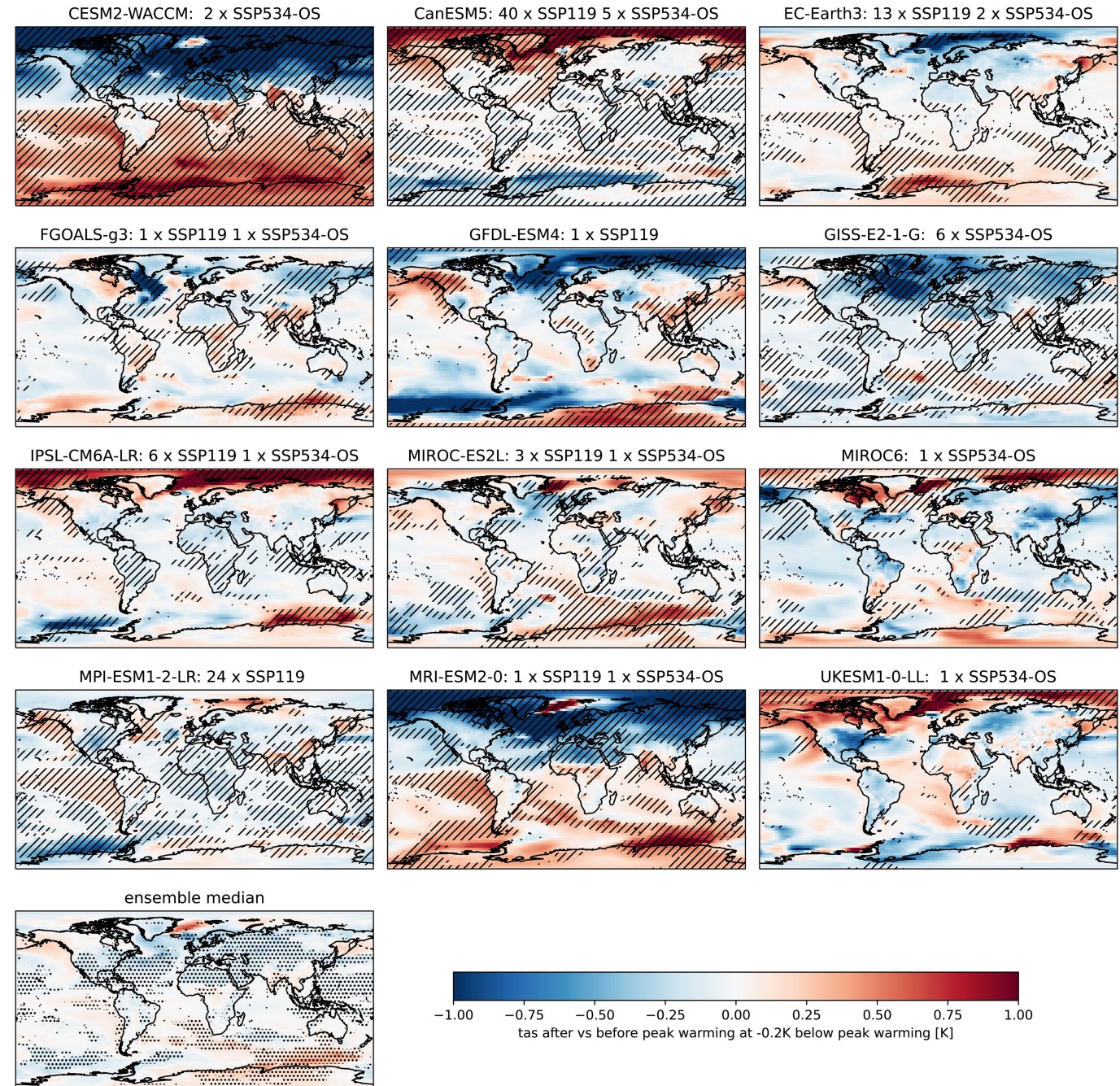

**Extended Data Fig. 7 | Differences between regional annual temperature before and after overshoot in a CMIP6 model ensemble.** Patterns are shown for centred 31 yr periods for GMT of −0.2 °C below peak warming before and after overshoot in the SSP5-34-OS and the SSP1-19 pathways (see Methods). In the first 12 panels, hatching highlights grid-cells where the difference exceeds the 95th percentile (is below the 5th percentile) of comparable period differences in piControl simulations (see Methods). For the ensemble median (last panel) stippling indicates a model agreement in the sign of change of at least 66% of the models.

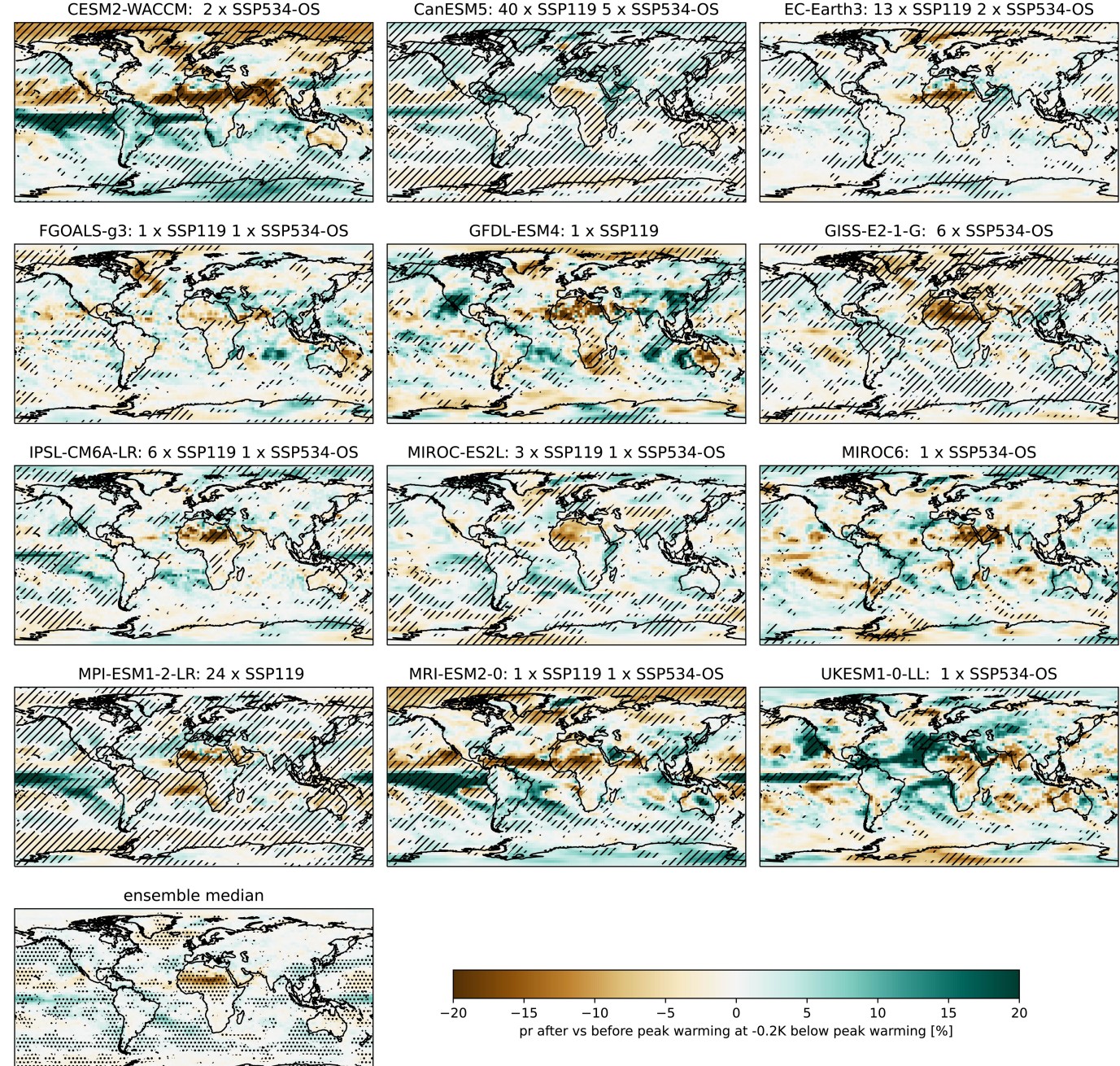

**Extended Data Fig. 8 | Differences between regional annual precipitation before and after overshoot in a CMIP6 model ensemble.** Patterns are shown for centred 31 yr periods for GMT of −0.2 °C below peak warming before and after overshoot in the SSP5-34-OS and the SSP1-19 pathways (see Methods). In the first 12 panels, hatching highlights grid-cells where the difference exceeds the 95th percentile (is below the 5th percentile) of comparable period differences in piControl simulations (see Methods). For the ensemble median (last panel) stippling indicates a model agreement in the sign of change of at least 66% of the models.

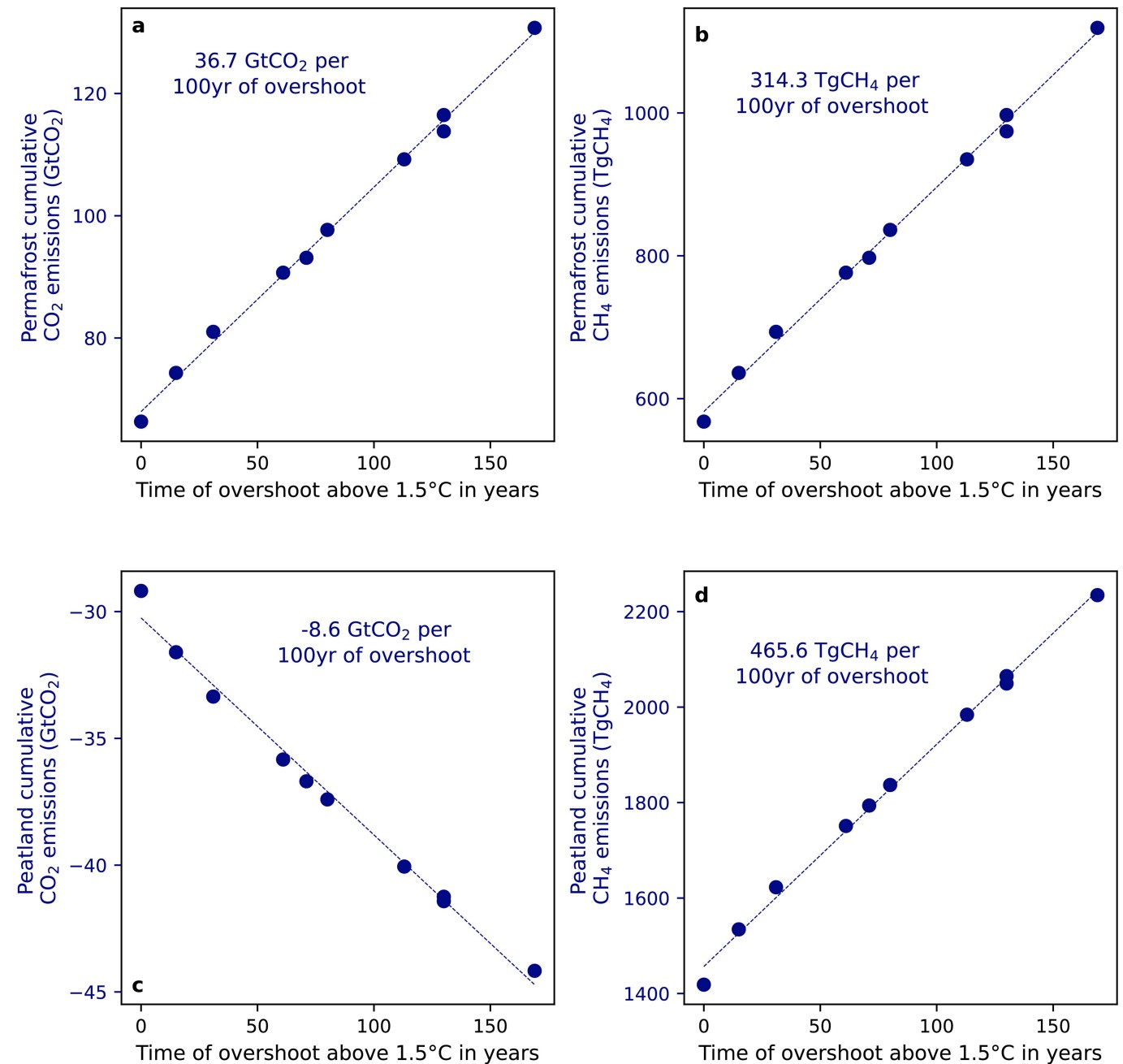

**Extended Data Fig. 9 | CO₂ and CH₄ emissions from permafrost and peatlands under overshoot. a**, Cumulative $CO_2$ emissions permafrost emissions as a function of length above 1.5 °C. **b**, $CH_4$ emissions from permafrost. **c**, $CO_2$ emissions from peatlands. **d**, $CH_4$ emissions from peatlands.

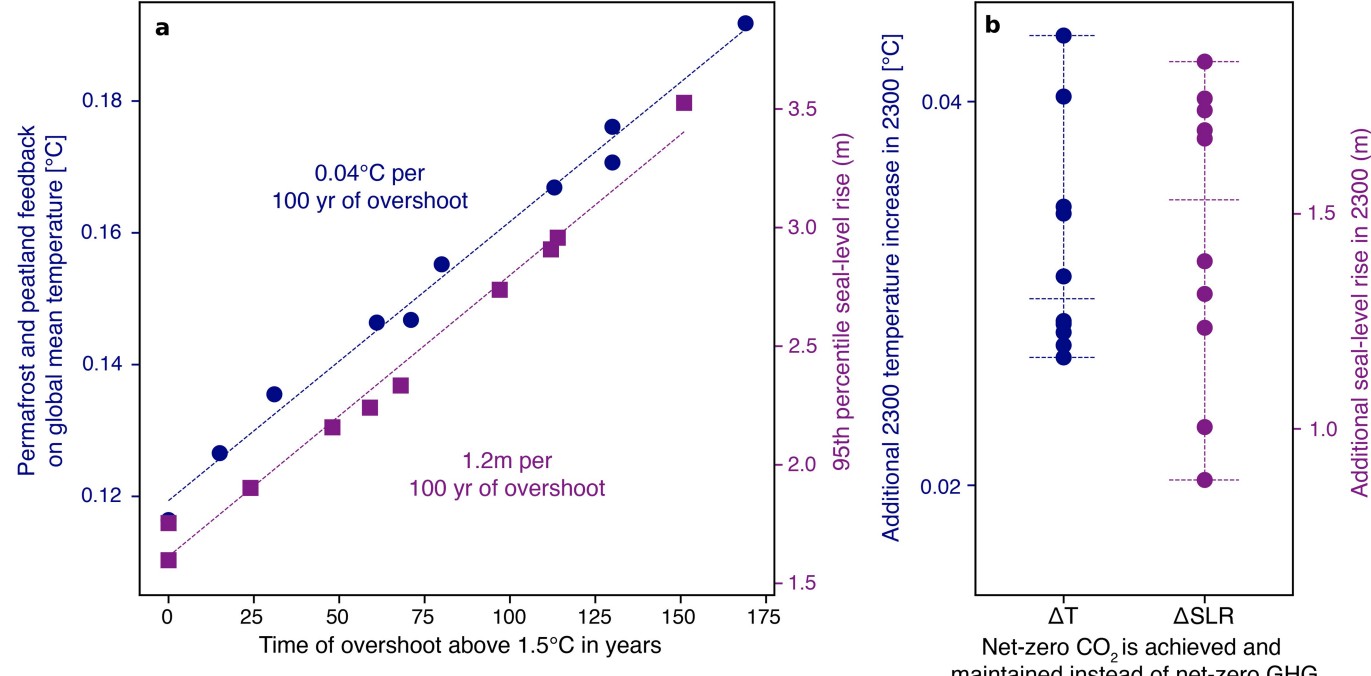

**Extended Data Fig. 10 | High-end long-term irreversible permafrost, peatland and sea-level rise impacts of overshoot.** As Fig. 4, but for the 95% quantile outcomes. **a**, Feedback on 2300 global mean temperature increase by permafrost and peatland emissions (blue markers and left axis) and 2300 global median sea-level rise (right axis) as a function of overshoot duration. Note that while the vertical axis provides 95% quantile outcomes, the overshoot length on the horizontal axis refers to the median overshoot length under a given scenario as in Fig. 4 to allow for direct comparability. **b**, Additional global mean temperature from warming-induced permafrost and peatland emissions and sea-level rise increase implied by stabilising temperatures at peak warming by achieving net-zero $CO_2$ emissions compared to a long-term temperature decline implied by achieving and maintaining net-zero GHGs. Circles (squares) mark results for temperature change (sea-level rise) for individual scenarios from ref. 37. Dashed horizontal lines in **b** provide the ensemble median and min/max range.

**Extended Data Table 1 | Literature categories of peak and decline emission pathways**

| Pathway Category | Temperature Characteristics | Emission Characteristics (Best Estimates) |
|---|---|---|
| Pathways that limit warming to 1.5°C (>50%) with no or limited overshoot (C1)[2] | Pathways that limit warming to 1.5°C in 2100 with a likelihood of greater than 50%, and reach or exceed warming of 1.5°C during the 21st century with a likelihood of 67% or less.<br><br>Limited overshoot refers to median estimates of global warming exceeding 1.5°C by up to about 0.1°C and for up to several decades. C1 pathways that achieve net-zero GHG are included in the sub-category C1a. | 2030 reductions of total GHG emissions relative to 2019:<br> 43% [34-60 %, 5th-95th percentile range]<br>Timing of net-zero $CO_2$:<br> 2050-2055 [2035-2070]<br>Timing of net-zero GHG (only category C1a pathways):<br> 2070-2075 [2050-2090]<br>Cumulative net-negative $CO_2$ after net-zero:<br> 220 $GtCO_2$ [20-660] |
| Pathways that return warming to 1.5°C (>50%) after a high overshoot (C2)[2] | Pathways that limit warming to 1.5°C in 2100 with a likelihood of greater than 50%, and exceed warming of 1.5°C during the 21st century with a likelihood of greater than 67%.<br><br>High overshoot refers to median global warming projections temporarily exceeding 1.5°C by 0.1-0.3°C for up to several decades | 2030 reductions of total GHG emissions relative to 2019:<br> 23% [0-44 %, 5th-95th percentile range]<br>Timing of net-zero $CO_2$:<br> 2055-2060 [2045-2070]<br>Timing of net-zero GHG:<br> 2070-2075 [2055-...]<br>Cumulative net-negative $CO_2$ after net-zero:<br> 360 $GtCO_2$ [60-680] |
| Paris Agreement compatible pathways[17] | Pathways that reach or exceed warming of 1.5°C during the 21st century with a likelihood of 67% or less, and simultaneously do not exceed 2°C during the 21st century with a likelihood of 90% or more.<br>Achieve long-term declining temperature by reaching net-zero GHGs. Similar to C1 pathways in the near term and category C1a pathways in the long term (post-2050). | 2030 reductions of total GHG emissions relative to 2019:<br> 41% [38-44 %, interquartile range]<br>Timing of net-zero $CO_2$:<br> 2050 [2045-2055]<br>Timing of net-zero GHG:<br> 2065 [2060-2075]<br>Cumulative net-negative $CO_2$ after net-zero:<br> 453 $GtCO_2$ [127 - 690] |

**Extended Data Table 2 | Overview of constraints of large-scale CDR[72–89]**

| | Description of constraints and potential for overconfidence |
|---|---|
| **Readiness** | Current removal capacities are far from what is required to be compatible with the Paris Agreement. In the coming years, removal scales need to go up while costs need to come down – both at highly ambitious levels. Implementation gaps already arise, potentially precluding reliance on CDR to steer back from overshoot[27]. |
| **Permanence & Resilience** | Permanent and secure storage of removed carbon is key. Overconfidence may arise from neglected uncertainty of the geological storage potential[72] and overestimated storage durability of land and ocean sinks under progressing climate change. Carbon stored in soils and vegetation is especially susceptible to climate or non-climatic impacts, including fires or pest infestation, and may be constrained further if total sequestration potentials are lower than current best estimates[73-76]. Carbon sequestration in marine ecosystems is equally vulnerable to climate impacts[77]. |
| **System feedbacks** | Mitigation effects of CDR may be offset by weakened and potentially reversed land and ocean carbon sinks, and other undesired system feedbacks[78], e.g., unfavourable albedo changes, or emissions due to direct or (unintended) indirect land use change. Carbon uptake potential of land-based CDR is highly uncertain, depending on bioenergy crop yields in the case of bioenergy and carbon capture and storage (BECCS) and soil carbon response to land-use change and the rate of forest regrowth in the case of afforestation[79,80]. |
| **Policy response & Governance** | Betting on CDR effectiveness may lead to insufficient emission reductions if CDR underperforms, or physical climate feedbacks are stronger than expected. The outlook of potential future CDR availability could deter mitigation, meaning that required gross emission reductions may be delayed and/or weakened[25,81] - an effect that can also be observed in integrated assessment models[82,83]. Lacking monitoring and liability of removal additionality and permanence may pose an additional constraint[8]. |
| **Sustainability & Acceptability** | The extensive land use footprint associated with large-scale CDR may threaten environmental integrity[9,84-86] and/or agricultural production[73]. However, some types of CDR (for example, via restoration of natural ecosystems and their associated carbon) would be more synergistic. CDR often requires public acceptance – an aspect not reflected in current scenarios. Consensus is critical, as CDR can lead to undesired distributional impacts (e.g., concerning land tenure or food prices if large areas are allocated for CDR). Further constraints arise when considering (transnational) equity criteria, as the burden of CDR may not be evenly distributed between polluters, regions, and generations[48,87]. Even with strong CDR deployment by high-income countries, equitable mitigation outcomes may not be achieved[88,89]. |