## [Peer Review file · Nature]

Manuscript Title: Overconfidence in climate overshoot

Reviewer Comments & Author Rebuttals

Reviewer Reports on the Initial Version:

Referee #1 (Remarks to the Author):

The authors present a range of results from their research on overshoot scenarios using a range of methods (from simple climate models to analysis of results from complex ESMs). They argue to focus the assessment of overshoot pathways more on a risk-based framework instead of focusing on median outcomes, which bears the risk of being overconfident in the feasibility and "safety" of overshoot pathways. The manuscript is generally well written and provides interesting insights. It is, however, partly a bit of a mixed bag and some of the results fit better under the topic of potential overconfidence in overshoots than others. For example, it did not become clear to me how the difference between achieving net-zero CO₂ and net-zero GHG emissions would contribute to a potential overconfidence in overshoot pathways. I also have one major concern regarding the method to quantify the impacts of an overshoot (compared to a world avoided an overshoot) and a few other general and specific points that I list below.

General comments:

1) One of the main points of this manuscript is, as stated clearly in the abstract, "that global and regional climate change in a post-overshoot world would be substantially different from a world that avoided overshoot, bearing profound implications for adaptation needs." I am afraid that I cannot follow the authors here. The method used by the authors to determine the difference between pre-overshoot and post-overshoot is not suited to isolate the impact of the overshoot. The authors use results from CMIP6 Earth system models simulating two CMIP6 SSP scenarios (SSP5-3.4-os and SSP1-1.9), and find pairs of time-slices (for each model) that have the same (global mean) temperature difference relative to peak temperature (-0.1 and -0.2 degrees), one time slice before the peak, one time slice after the peak (for each temperature). The problem with this method is that it mixes transient changes and the effect of the overshoot. There is no reason to assume that the transient effect would be small, and the authors themselves point this out in lines 267-268: "Even if global warming were to be stabilised at a certain level, the climate system would continue to change as components of the climate system continue to adjust and equilibrate". The same is true if instead of stabilizing at a certain temperature level, there is a temperature decline.

For example, the authors find that the "most apparent difference in annual mean temperature patterns is a stronger cooling over land after overshoot, as the land-ocean contrast is reduced". However, a reduction of the land-ocean contrast is a general result of a more equilibrated climate state (King et al. 2020), it remains unclear how large the contribution of the overshoot for this result is. The same is true for the impacts of the AMOC, which is known to shape the regional temperature and precipitation responses significantly (e.g. Bellomo et al. 2021; Schwinger et al. 2022). AMOC changes evolve over multiple decades in response to climate forcing, and it is not possible to attribute the temperature and precipitation changes to an overshoot using the time-slice method as the authors do.

This is a shortcoming of the CMIP6 ScenarioMIP design - what would be urgently needed is (one or several) scenario pairs achieving the same 2100 temperature (or carbon budget) outcome with no/limited overshoot and with overshoot. Such scenario pairs could be compared in a meaningful way to investigate the impacts of an overshoot relative to a world that avoided overshoot. The comparison the authors present here is not suited to quantify such impacts.

The authors also claim that climate would be "substantially" different due to the effects of an overshoot, but they do not provide any quantitative measure of what they would consider "substantial". Examples from the recent literature that try to define such measures include measures related to the hysteresis (Kim et al. 2022) or measures relative to internal variability (Schwinger et al. 2022b).

2) A core element in this analysis is a large ensemble of scenario simulation with the simple climate and carbon cycle model FaIR. Fig. 2a shows that there are ensemble members (beyond the 95th percentile) that show rather extreme warming outcomes and would require rather large (and probably infeasible) amounts of CDR. Two things would be interesting to know: First, what is it that characterizes these extreme warming ensemble members? In other words, can we learn something that relates to the real world from this large ensemble, i.e. which parameter and their uncertainties do we need to constrain better? Is this just dominated by TCR/ECS or are there other first order parameters that set this uncertainty (and consequently a potential overconfidence)? Second, it is stated that "This probabilistic setup of FaIR is consistent with assessed ranges of equilibrium climate sensitivity, historical global average surface temperature and other important metrics assessed by IPCC AR6 WG1", but in this context it would be interesting to know more about how the parameter space is sampled and the stated "consistency". Are the extreme warming ensemble members as consistent with e.g. historical warming or historical carbon uptake as those ensemble members close to the median?

More specific comments:

lines 38-40: "Irrespective of the peak warming, we find that achieving declining global temperature remains critical for limiting long-term climate risks including sea-level rise and cryosphere changes." I find this sentence confusing. What exactly is meant by "irrespective of the peak warming"? Peak warming doesn't matter?

line 52-53: Is "potentially important" the right wording here?

lines 60-65: This paragraph is not very helpful in my opinion. Even if overshoot was used for atmospheric concentrations, the dominant framing today is that of overshooting a carbon budget, which directly translates into overshooting a temperature target if the paradigm of constant TCRE holds. Then we might have non-zero ZEC and non-CO2 forcings. I would find it more helpful for the reader to briefly recap these concepts here.

lines 65: Why is ref 14 cited in this context?

lines 72-73: "...would lead to declining temperatures (central estimate) in the long term" What are the uncertainties around this estimate?

line 99: "...can give rise to the impression that the temperature overshoot risk under such scenarios is constrained to a few tenths of a degree" Why is ref. 14 cited here? This can give rise to the impression that these authors would have published such misleading statements?

line 116: "...determines the pace of temperature reversal" This must be based on some assumptions? I don't have access to the cited publication, but I assume this is based on TCRE and assuming that faster removal will give faster temperature decline?

line 137-140: It might be good to clarify what assumptions are underlying this statement. One would be that TCRE is constant (and remains constant under negative emissions)?

line 143: The authors mean long term "median warming outcomes"?

line 154: "pathway PROVIDE REN_NZCO2" is the use of this acronym necessary here?

line 194: I find the wording "peat carbon loss from climate and anthropogenic land use change" confusing, maybe "peat carbon loss due to climate change and degradation or conversion of peat lands"?

lines 198-202: "...as there is no observational evidence available to constrain the modelled response to net-negative CO₂ emissions. It therefore appears plausible that substantial residual uncertainties about the Earth System response will remain beyond the time net-zero CO₂ is reached, implying the need for a preventive approach to hedge against undesirable warming outcomes in the long term." I agree that it is reasonable to "hedge against undesirable outcomes". I agree less with the statement that "uncertainties will remain beyond the time net zero is reached". Earth system processes are not suddenly fundamentally different if emission were to become negative. The problem is that we have too sparse observations for a too short time periods to constrain some key processes, but up to the time when net-zero is reached we will hopefully have much more complete and longer observational records.

Table 2: "...and may be constrained further by uncertain sink saturation^{38,42–44}" It is unclear to me what this is supposed to mean. Also what is the difference between this and "System feedbacks"?

line 286: Strong regional high northern latitude impacts of AMOC slowdown when emissions are phased out or become negative have been also highlighted in a recent paper by Schwinger et al (2022).

line 386: "...the order of around 0.05°C per decade" - this is due to limitations of the rate of CDR?

Section "Time-lagged and irreversible impacts": In this section the topic of comparing net-zero CO₂ and net-zero GHG pathways suddenly comes up. This has not properly introduced and comes a bit out of the blue. Also, it remains unclear why this contributes to overconfidence in overshoot pathways?

lines 440-442: "As we have shown, whether or not a long-term decline in global temperatures can be achieved depends on uncertain physical climate system feedbacks, but nevertheless needs to rely on large scale carbon dioxide removal." Fig. 2 shows that there is a substantial number of ensemble members that show a temperature decline after peak without CDR, right? If that is correct I don't understand the logic of this sentence ("nevertheless needs to rely on CDR").

lines 458-460: "Overshooting of 1.5°C (or even 2°C) is not something that can be planned for with certainty - it is a question of probabilities and might prove impossible due to physical climate system feedbacks under a range of emission pathways." Currently, it does not look like overshooting 1.5 or 2 degrees is impossible... Please reconsider the logic of this sentence.

References:

Bellomo, K., Angeloni, M., Corti, S. et al. Future climate change shaped by inter-model differences in Atlantic meridional overturning circulation response. *Nat Commun* 12, 3659 (2021).

<https://doi.org/10.1038/s41467-021-24015-w>

Kim, SK., Shin, J., An, SI. et al. Widespread irreversible changes in surface temperature and precipitation in response to CO₂ forcing. *Nat. Clim. Chang.* 12, 834–840 (2022).

<https://doi.org/10.1038/s41558-022-01452-z>

King, A.D., Lane, T.P., Henley, B.J. et al. Global and regional impacts differ between transient and equilibrium warmer worlds. *Nat. Clim. Chang.* 10, 42–47 (2020). <https://doi.org/10.1038/s41558-019-0658-7>

Schwinger, J., Asaadi, A., Goris, N., and Lee, H. (2022): Possibility for strong northern hemisphere high-latitude cooling under negative emissions, *Nat. Commun.*, 13, 1095, doi:10.1038/s41467-022-28573-5

Schwinger, J., Asaadi, A., Steinert, N. J., and Lee, H. (2022b): Emit now, mitigate later? Earth system reversibility under overshoots of different magnitudes and durations, *Earth Syst. Dynam.*, 13, 1641–1665, doi:10.5194/esd-13-1641-2022

Referee #2 (Remarks to the Author):

This paper provides an in-depth and comprehensive overview of a very important and timely policy-relevant topic, the understanding the implication of overshoot scenarios. The authors point out that the current framing of overshoot scenarios is misleading as it implies that declining temperature after a peak is roughly equivalent stabilizing at that peak. However, they point out that different overshoot pathways could lead to significantly different climate outcomes with important implications for climate adaptation and carbon dioxide removal (CDR) requirements.

This paper provides critical new perspectives that are needed to inform the emerging policy and business discussion on assessing climate risks and crediting CDR. I would encourage this to be published.

Here are a few suggestions to improve clarity in the paper.

(1)The paper is very dense, and thus a bit hard to get through. Given the importance of the topic I would suggest two options to guide the reader through

a.) Explicitly outline implications (with headings or a table) for: climate adaptation, Mitigation targets and remaining carbon budget?, and CDR targets and allocation.

b.)Consider having a news or commentary piece published alongside this to clarify the significance of the messages emerging from this work.

(2)Figure 5, part b. I do not understand what the different bars and colors represent in part b of this figure. What do the seven bars under lifetime vs four under planning horizon. And what are the significance of the different discount rates shown - Are these just illustrative? Suggest you provide more explanation in the caption and text.

(3)CDR implications. The analysis of this paper has significant implications for the how to evaluate the need, contribution and benefits of CDR. The discussion around the implications for CDR could be more succinctly outlined, in particular the idea of Preventive CDR capacity. Lots of important issues are raised, but do not come through as clearly as they could.

(4) Climate adaptation implications. I would suggest providing a bit more detail on the implications of not taking this more nuanced approach to overshoot pathways. I realize that in some ways that what I am suggestion could be a whole new paper. But a bit more unpacking of this, and the CDR would help to further clarify the importance the analysis outlined in the paper.

Referee #3 (Remarks to the Author):

Overconfidence in overshoot – review notes

With minor revisions I would recommend publication. This article adds new rationales (rooted in modelling of overshoot scenarios) for reserving (limited, sustainable) CDR capacity for drawdown, rather than offsetting residual emissions; and for accelerating emissions reductions in the immediate future. It highlights uncertainties related to the extent and climatic implications of temporary overshoot of climate temperature goals, and worrying implications for adaptation and loss and damage activities from additional committed climate impacts if overshoot pathways (even those within the 0.3C margins considered by IPCC) are permitted. I see no fundamental flaws, although it does exhibit a bias towards modelling and modelling literatures which limits its engagement with other literatures examining the same challenges, and its policy relevance.

General comments

The opening summary of the paper offers several broad conclusions, some generally accepted in scientific circles, but meriting repetition for policy makers (notably ‘emphasize early emissions reductions’ and ‘scaling CDR cannot be guaranteed’), and some more novel: notably that climate impacts will differ substantially in a ‘post-overshoot’ world from those where overshoot is avoided. As I read it, the central recommendations are two fold. Both are valid, but neither element is presented with the clarity it deserves. First, emissions reduction should be accelerated as quickly as possible (“stringent immediate” both lacks specificity, relative to current efforts, and is easily dismissed as politically impractical). Second, while CDR capacity should be developed (within social and sustainability constraints), it should not be allocated to counterbalance residual emissions (or at least, not those that can foreseeably be mitigated). The political implications of both these are profound, but remain somewhat opaque in the present text. For example, carbon trading and offsetting markets do not support these goals, to name but one climate canard they would fell.

In part I fear this lack of clarity regarding political and policy implications is a product of a bias towards the production of climate knowledge through modelling, and a tendency to cross-citation amongst scholars using such techniques, which leaves other relevant literatures and epistemologies under-represented, even where they have reached similar conclusions in advance of this author group. I would encourage the authors to undertake some minor revisions to highlight political implications, and to cite relevant literature which supports their analysis from different foundations. In reflecting on the effects of a modelling orientation, they might find Ho et al. (2019) useful.

The introduction in particular merits careful redrafting in particular, and the production of a dedicated abstract which reflects the needed clarity. At present some of the conclusions presented in the introduction are phrased in ways that seem contradictory: for example, while arguing that the technical development of CDR cannot be relied upon, the authors propose that a preventive capacity of hundreds of Gt should be sought nonetheless. Reading ahead I can see that they mean to suggest that if CDR were developed to the scales foreseen in many overshoot pathways, it may well exceed social and sustainability constraints, yet still might not be adequate to significantly reduce temperatures in an overshoot world, due to feedbacks (or tipping points) in warming systems). But this then makes the phrasing of another conclusion peculiar “irrespective of the peak warming, we find declining global temperature remains critical to reducing climate risks”. Clearly this is true in abstract, but the previous finding suggests rather that with rising peak temperatures the nature of those risks changes, and the prospect of delivering temperature decline may be reduced. And later, the text indicates many risks which are more directly related to peak warming than to the rate or scale of subsequent temperature decline. This section therefore merits careful attention to ensure consistency and clarity.

Further, the argument for “redirecting the discussion towards managing climate

risks both in the near and long-term” is rather vague and ambiguous. Perhaps the authors intend to suggest that both near and long-term risks should be considered, and this means both cutting emissions faster now, and developing a preventive (sustainable) CDR capacity (as opposed to doing one or the other). By contrast the current phrasing implies shifting attention from emissions to risks, an approach which might be expected to encourage consideration of SRM interventions, given existing ‘risk-risk’ literature in this space (eg Felgenhauer et al. (2022).

I’d strongly recommend that the authors redraft their conclusions to ensure clarity regarding these matters for both scientific and policy readers.

There are several other sections that merit minor revisions for clarity on approach, implications or limitations.

1. Categorising ‘peak and decline’ pathways. This is a useful effort to relate overshoot to the characteristics of the majority of policy relevant emissions pathways. However in some respects it seems detached from the most critical aspects of policy relevance in the existing and proposed net-zero pathways adopted by the majority of states globally. Being explicit that existing net-zero pathways do not necessarily avoid overshoot would seem critical. Scholarly and policy debate about ‘the area under the curve’ (the cumulative emissions allowed between now and net-zero) is intense

and significant (e.g. Armstrong and McLaren, 2022; Fankhauser et al., 2022; Kaya et al., 2019; Sun et al., 2021). Making such connections would add to the policy utility of this article.

2. Use of the FaIR model. I am not a modeller, but consider myself reasonably well versed in the limitations of modelling. I find it inappropriate to specify the choice of model without raising caveats about the extent to which the additional simplifications of FaIR over full ESMs might themselves generate excessive certainty in the output set. For example the asserted 'finding' at lines 175-9 would appear to be an artefactual product of the choice of model and input constraints. The discussion of FaIR limitations at 188-192 is helpful, but comes too late, and is incomplete with respect to the input constraints described.

3. Calculating and discussing NNCE figures merits greater specificity. Scholarly work on ZEC has highlighted a need to be clear about the distinctions between the response of natural sinks to net-zero; the role of enhanced natural sinks (eg through reforestation); and the role of specifically anthropogenic removals. This is of intense policy relevance as some countries seek to redefine natural sinks as removals in their net-zero calculations. I recommend a brief clear definition of what is included in 'CDR requirements' or NNCE in this context, which would also provide a critical opportunity to note that 'requirements' calculated in this way may easily exceed capacities which could be deployed in socially just and environmentally sustainable ways.

4. Constraints to CDR capacity (lines 231-248 and table 2). This discussion seems somewhat ad-hoc, and lacking in a rigorous framework to ensure comprehensive and balanced treatment. In particular, in the main text, sustainability and equity concerns appear as a tag-on to technical and climate feedback issues. This perhaps reflects a bias in the disciplinary skills of the authors, but should be rectified (relevant examples include: Waller et al., 2020; McLaren, 2012; Smith et al., 2015; Cox et al., 2018; Forster et al., 2020). The table is better (with more diverse and complete issues and sources), although still particularly poor on the issue of mitigation deterrence (here still described with the outdated terminology of 'moral hazard') which has a rich recent literature (e.g. McLaren, 2020; McLaren et al., 2021; Carton et al., 2023). Leakage of captured carbon into utilisation in CCUS approaches, or enhanced oil recovery is totally overlooked even though dominant in actual existing engineered carbon capture. Uncertainties relating to marine CDR are not mentioned (e.g. Boettcher et al., 2021; Mengis et al., 2023).

5. The discussion of SLR, permafrost and peatland commitments from overshoot (lines 327ff) seems over-simplified. It is important to demonstrate that these risks will continue to grow as a result of overshoot, but it seems highly unlikely that they will respond in the direct linear fashion suggested here. In particular, SLR responses will presumably be highly conditional upon the tipping points of ice-sheet dynamics. For permafrost thaw, there may be more linearity, but relationships with ocean-current changes could introduce significant non-linear responses. For peatlands, the relationship between precipitation changes, new growth and methanogenesis is far from simply linear. The

simplifications used here must be clarified, along with noting the relevant uncertainties. I'd also question whether reporting impacts in terms of 'per 100 years overshoot' is appropriately consistent with the rest of the article, which (eg fig 5) presents the implications of overshoot measured in decades (60-70 years).

6. SRM dilemma. Overshoot is a concept widely used in geoengineering literature, and the risks and uncertainties it involves are seen by some as offering a clear justification for SRM interventions (e.g. MacMartin et al., 2018). SRM is mentioned briefly at line 445, but not in the discussion of particular regional effects of overshoot, some of which might (theoretically) be susceptible to amelioration by tailored or modulated SRM interventions (eg the ITCZ shift – (see e.g. Haywood et al., 2013: for discussion of asymmetry and SRM)). I am highly sceptical of the technical and political capacity to deliver such interventions (see McLaren, 2018), but the authors might be advised to consider such possibilities.

7. In contrast to the summary text in the introduction, the material in the concluding section (line 477ff) is clear that the aim of preventive CDR capacity is to develop significant capacity within sustainability and social constraints which is not allocated to counterbalance residual emissions, but is dedicated to delivering drawdown of atmospheric GHG concentrations. In this they echo substantial scholarship on CDR, Net Zero and residual emissions (e.g. McLaren et al., 2019; Buck et al., 2023; Armstrong and McLaren, 2022) (see also Ho, 2023: <https://www.nature.com/articles/d41586-023-00953-x>). It would seem appropriate to acknowledge this prior scholarship, and not imply that the conclusions here are novel.

In addition some minor amendments may be merited at the following points

Line 53-4 I'd note that Azar et al were far from the earliest to identify a key role for large scale CDR / NETs / GGR in stabilizing and reversing temperatures. For example the UK Climate Change Committee already commission research on this before 2010. Azar et al may not have even been the first to model this (and these authors should be well aware that modelling does not hold a monopoly on relevant climate knowledge ...)

Line 138 The word 'determine' is inappropriate here, given possible hysteresis in response to rising and falling GHG concentrations – other factors are therefore also relevant (as recognised in lines 146ff). Using 'determine' risks a similar form of 'overconfidence' in projections that the authors seek to warn against.

Line 169-70 the description of the results in the upper right quadrant is ambiguous: the terminology used applies to the entire right hand part of the chart (both upper and lower quadrants)

Line 220-221 the choice of 'conventional' and 'novel' as categories of CDR is not established in the literature: it is a deliberate effort in the non-peer-reviewed report cited to establish new discursive categories. It should therefore not be used unquestioningly. It may indeed be preferable to distinctions based on capture method (biological vs chemical; or natural vs engineered), location (terrestrial, oceanic etc), or final storage location (biologic, geologic, oceanic) for example. But categories matter (Heyward, 2013), and the choice must be justified. This choice, for example appears to treat all marine or oceanic methods as 'novel', lumping them in with engineered, and hybrid approaches such as BECCS and DACCS.

Line 222-226 The description of the pathway sets is rather misleading, as the central estimates for low or no overshoot pathways are exaggerated by the asymmetric effects of geophysical uncertainty, thus disguising the significant additional CDR typically demanded in overshoot pathways. The effect is to undermine one of the key arguments: that reliance on CDR should be minimised by minimising near-term and thus cumulative emissions.

Line 227-9 seems to be a restatement of the point that cumulative emissions are the main factor, but phrased in a way that is hard to parse. This sentence could probably be deleted without any loss.

Line 273 ff presents a series of projected regional effects. These are all large scale, and likely very plausible, but should be prefaced with some caveats about the capacity of the relevant models (CMIP group ESMs) to project regional effects, the relevant degrees of uncertainty thus associated with such projections. Some caveats are given in the caption for Figure 3, but not explained. These effects and the figure appear to be at least partly derived from a reference which is incomplete/not yet published (note 65): please confirm that they are based on peer-reviewed material.

Line 355-358 – the conceptualisation of vulnerability here seems naïve. As I read this text vulnerability is seen as something abstract and objective, a product of a paucity of economic development. In reality, vulnerability is as much a social construct, a product of uneven development and the construction of sacrifice zones and economic precarity in the interests of extractive economic growth. While the authors may not feel qualified to examine such challenges, they should at least acknowledge that there are radically different ways of conceptualising vulnerability that do not lead to a presumption or impression that further economic development is the only, or best way to alleviate vulnerability. In considering how to 'reframe' overshoot, the question of how vulnerability to climate impacts might be constructed, or deconstructed might be a valuable additional element.

Line 397 ff It would seem inconsistent to imply that the uncertainties relating to overshoot and its impacts make cost-benefit calculations inappropriate or misleading for CDR development and mitigation choices, but to then retain a conventional cost-benefit approach to adaptation to the same uncertainties.

Line 492-496 This paragraph adds nothing, and risks confusing the reader through oversimplification. It could be deleted without loss.

References

Armstrong C and McLaren D (2022) Which Net Zero? Climate Justice and Net Zero Emissions. *Ethics and International Affairs* 36(4): 505-526.

Boettcher M, Brent K, Buck HJ, et al. (2021) Navigating Potential Hype and Opportunity in Governing Marine Carbon Removal. *Frontiers in Climate* 3(47).

Buck HJ, Carton W, Lund JF, et al. (2023) Why residual emissions matter right now. *Nature Climate Change*. DOI: 10.1038/s41558-022-01592-2.

Carton W, Hougaard I-M, Markusson N, et al. (2023) Is carbon removal delaying emission reductions? *WIREs Climate Change* 14(4): e826.

Cox E, Pidgeon N, Spence E, et al. (2018) Blurred Lines: the Ethics and Policy of Greenhouse Gas Removal at Scale. *Frontiers in Environmental Science* 6.

Fankhauser S, Smith SM, Allen M, et al. (2022) The meaning of net zero and how to get it right. *Nature Climate Change* 12(1): 15-21.

Felgenhauer T, Bala G, Borsuk M, et al. (2022) Solar Radiation Modification: A Risk-Risk Analysis. Reportno. Report Number |, Date. Place Published | : Institution |.

Forster J, Vaughan NE, Gough C, et al. (2020) Mapping feasibilities of greenhouse gas removal: Key issues, gaps and opening up assessments. *Global Environmental Change* 63: 102073.

Haywood JM, Jones A, Bellouin N, et al. (2013) Asymmetric forcing from stratospheric aerosols impacts Sahelian rainfall. *Nature Climate Change* 3(7): 660-665.

Heyward C (2013) Situating and Abandoning Geoengineering: A Typology of Five Responses to Dangerous Climate Change. *PS: Political Science & Politics* 46(1): 23-27.

Ho E, Budescu DV, Bosetti V, et al. (2019) Not all carbon dioxide emission scenarios are equally likely: a subjective expert assessment. *Climatic Change* 155(4): 545-561.

Kaya Y, Yamaguchi M and Geden O (2019) Towards net zero CO₂ emissions without relying on massive carbon dioxide removal. *Sustainability Science*. DOI: 10.1007/s11625-019-00680-1.

MacMartin DG, Ricke KL and Keith DW (2018) Solar geoengineering as part of an overall strategy for meeting the 1.5°C Paris target. *Philosophical Transactions of the Royal Society A: Mathematical, Physical and Engineering Sciences* 376(2119).

McLaren D (2012) A comparative global assessment of potential negative emissions technologies. *Process Safety and Environmental Protection* 90(6): 489-500.

McLaren D (2018) Whose climate and whose ethics? Conceptions of justice in solar geoengineering modelling. *Energy Research & Social Science* 44: 209-221.

McLaren D (2020) Quantifying the Potential Scale of Mitigation Deterrence from Greenhouse Gas Removal Techniques. *Climatic Change* 162: 2411–2428.

McLaren D, Willis R, Szerszynski B, et al. (2021) Attractions of delay: Using deliberative engagement to investigate the political and strategic impacts of greenhouse gas removal technologies. *Environment and Planning E: Nature and Space* 0(0): 25148486211066238.

McLaren DP, Tyfield DP, Willis R, et al. (2019) Beyond “Net-Zero”: A Case for Separate Targets for Emissions Reduction and Negative Emissions. *Frontiers in Climate* 1: 4.

Mengis N, Paul A and Fernández-Méndez M (2023) Counting (on) blue carbon—Challenges and ways forward for carbon accounting of ecosystem-based carbon removal in marine environments. *PLOS Climate* 2(8): e0000148.

Smith P, Davis SJ, Creutzig F, et al. (2015) Biophysical and economic limits to negative CO₂ emissions. *Nature Climate Change* 6: 42.

Sun T, Ocko IB, Sturcken E, et al. (2021) Path to net zero is critical to climate outcome. *Scientific Reports* 11(1): 22173.

Waller L, Rayner T, Chilvers J, et al. (2020) Contested framings of greenhouse gas removal and its feasibility: Social and political dimensions. *WIREs Climate Change* 11(4): e649.

Author Rebuttals to Initial Comments:

We want to express our gratitude to the three reviewers who have provided very insightful comments that helped us to improve our manuscript. We have substantially revised our manuscript in response to their requests.

We include a point-by-point response below with our responses being marked by *green* colour.

Referee #1 (Remarks to the Author):

The authors present a range of results from their research on overshoot scenarios using a range of methods (from simple climate models to analysis of results from complex ESMs). They argue to focus the assessment of overshoot pathways more on a risk-based framework instead of focusing on median outcomes, which bears the risk of being overconfident in the feasibility and "safety" of overshoot pathways. The manuscript is generally well written and provides interesting insights. It is, however, partly a bit of a mixed bag and some of the results fit better under the topic of potential overconfidence in overshoots than others. For example, it did not become clear to me how the difference between achieving net-zero CO₂ and net-zero GHG emissions would contribute to a potential overconfidence in overshoot pathways. I also have one major concern regarding the method to quantify the impacts of an overshoot (compared to a world avoided an overshoot) and a few other general and specific points that I list below.

We thank the referee for this positive evaluation of our work and the very thoughtful comments. We are addressing the individual comments in detail below.

General comments:

1) One of the main points of this manuscript is, as stated clearly in the abstract, "that global and regional climate change in a post-overshoot world would be substantially different from a world that avoided overshoot, bearing profound implications for adaptation needs." I am afraid that I cannot follow the authors here. The method used by the authors to determine the difference between pre-overshoot and post-overshoot is not suited to isolate the impact of the overshoot. The authors use results from CMIP6 Earth system models simulating two CMIP6 SSP scenarios (SSP5-3.4-os and SSP1-1.9), and find pairs of time-slices (for each model) that have the same (global mean) temperature difference relative to peak temperature (-0.1 and -0.2 degrees), one time slice before the peak, one time slice after the peak (for each temperature). The problem with this method is that it mixes transient changes and the effect of the overshoot. There is no reason to assume that the transient effect would be small, and the authors themselves point this out in lines 267-268: "Even if global warming were to be stabilised at a certain level, the climate system would continue to change as

components of the climate system continue to adjust and equilibrate". The same is true if instead of stabilizing at a certain temperature level, there is a temperature decline.

For example, the authors find that the "most apparent difference in annual mean temperature patterns is a stronger cooling over land after overshoot, as the land-ocean contrast is reduced". However, a reduction of the land-ocean contrast is a general result of a more equilibrated climate state (King et al. 2020), it remains unclear how large the contribution of the overshoot for this result is. The same is true for the impacts of the AMOC, which is known to shape the regional temperature and precipitation responses significantly (e.g. Bellomo et al. 2021; Schwinger et al. 2022). AMOC changes evolve over multiple decades in response to climate forcing, and it is not possible to attribute the temperature and precipitation changes to an overshoot using the time-slice method as the authors do.

This is a shortcoming of the CMIP6 ScenarioMIP design - what would be urgently needed is (one or several) scenario pairs achieving the same 2100 temperature (or carbon budget) outcome with no/limited overshoot and with overshoot. Such scenario pairs could be compared in a meaningful way to investigate the impacts of an overshoot relative to a world that avoided overshoot. The comparison the authors present here is not suited to quantify such impacts.

The authors also claim that climate would be "substantially" different due to the effects of an overshoot, but they do not provide any quantitative measure of what they would consider "substantial". Examples from the recent literature that try to define such measures include measures related to the hysteresis (Kim et al. 2022) or measures relative to internal variability (Schwinger et al. 2022b).

We thank the referee for this very pertinent observation and agree with her/his comments. Our intent to use scenarios CMIP6 ensemble results in our analysis was to illustrate that, at the same GMT level after overshoot, regional patterns in temperature and precipitation changes are indeed different. Yet, beyond that general observation and as the reviewer rightly pointed out, the available scenario runs do not allow for an attribution of any of these observed features to the preceding overshoot.

So while we do see the value of including the CMIP6 results to illustrate the fact that the 'after' may be different from the 'before', with the necessary caveats, we agree that in order to substantiate the point about the effects of overshoot additional analysis would be required. Fortunately, we have been able to perform such additional analysis utilizing dedicated simulations comparing overshoot and stabilisation from two Earth System Models (ESMs), GFDL-ESM2M, and NorESM. We find that some key features that emerge in the individual model simulations are also present in the CMIP6

model ensemble, thus providing complementary insights. A key insight being that we still understand way too little on the implications of overshoot.

Therefore, and also here we agree fully with the referee, it is of critical importance that the next round of CMIP model experiments includes dedicated, emission driven overshoot scenarios to allow for much more substantive insights into the dynamics than what is possible given the limited set of experiments available today.

To address the referee's comment, we have performed the following changes:

1. We are including a new Fig. 3 based on dedicated overshoot vs. stabilisation experiments based on two ESMs. The new Fig.3 illustrates global and regional temperature evolution over several centuries after stabilization as well as after overshoot. A particular focus is put on the dynamics of the Atlantic Overturning Circulation (AMOC) that also exhibits significant differences between both models.
2. We include two additional figures in the Extended Data section based on the same analysis showing overshoot vs. stabilisation differences for annual precipitation, and transient regional differences in GMT stabilisation scenarios.
3. We have fundamentally revised the approach on how we show CMIP6 results now showing individual ESM results and indicating significance of the changes in the Extended Data Figures 6 and 7.
4. We have fundamentally revised the manuscript section "Regional climate change reversibility" to address the comments by the referee and reflect the new analysis included.
5. We revised the methodology of how we assess the significance of regional climate changes under overshoot vs. stabilization scenarios. Specifically, we compare the magnitude of the difference with random period differences of the same length in piControl simulations. If the difference exceeds the 95th percentile (or is below the 5th percentile) of differences found in piControl simulations we consider the difference as statistically significantly outside of internal climate variability.

2) A core element in this analysis is a large ensemble of scenario simulation with the simple climate and carbon cycle model FaIR. Fig. 2a shows that there are ensemble members (beyond the 95th percentile) that show rather extreme warming outcomes and would require rather large (and probably infeasible) amounts of CDR. Two things would be interesting to know: First, what is it that characterizes these extreme warming ensemble members? In other words, can we learn something that relates to the real world from this large ensemble, i.e. which parameter and their uncertainties do we need to constrain better? Is this just dominated by TCR/ECS or are there other first order parameters that set this uncertainty (and consequently a potential overconfidence)? Second, it is stated that "This probabilistic setup of FaIR is consistent with assessed ranges of equilibrium climate

sensitivity, historical global average surface temperature and other important metrics assessed by IPCC AR6 WG1", but in this context it would be interesting to know more about how the parameter space is sampled and the stated "consistency". Are the extreme warming ensemble members as consistent with e.g. historical warming or historical carbon uptake as those ensemble members close to the median?

We thank the referee for this important comment. In response to the comment we have added additional analysis in Extended Data Fig. 1 of the manuscript that illustrates the consistency of the FaIR version used with key AR6 assessed diagnostics, namely ECS, ZEC and the historic warming record, in line with the extended evaluation of the calibrated FaIR versions used here in the Supplementary Information for Chapter 7 of the AR6 Working Group 1 report. We added an additional section describing the evaluation in the Methods.

We also show how the assessed NNCE depends on the ECS/ZEC characteristics of the individual ensemble members. Based on this evaluation, we cannot rule out high ECS/ZEC ensemble members that drive the tail of our NNCE distribution (Extended Fig. 1 c). Yet, we find high NNCE outcomes also materialise for moderate-high ECS and ZEC configurations, in line with our reported finding that finds several hundred Gt NNCE already for the 75% quantile across the model ensemble.

We hope that this provides further confidence in our findings and addresses the very valid concern of the referee.

More specific comments:

lines 38-40: "Irrespective of the peak warming, we find that achieving declining global temperature remains critical for limiting long-term climate risks including sea-level rise and cryosphere changes." I find this sentence confusing. What exactly is meant by "irrespective of the peak warming"? Peak warming doesn't matter?

We thank the referee for pointing that out. Of course we do not want to imply that peak warming doesn't matter. We have revised the sentence to read:

"We find that achieving declining global temperatures can limit long-term climate risks compared to a mere stabilisation of global warming, including for sea-level rise and cryosphere changes"

line 52-53: Is "potentially important" the right wording here?

Unless ZEC turns out to be strongly negative, large-scale CDR will be fundamental to reverse warming. We thus have deleted the word 'potentially'.

lines 60-65: This paragraph is not very helpful in my opinion. Even if overshoot was used for atmospheric concentrations, the dominant framing today is that of overshooting a carbon budget, which directly translates into overshooting a temperature target if the paradigm of constant TCRE holds. Then we might have non-zero ZEC and non-CO2 forcings. I would find it more helpful for the reader to briefly recap these concepts here.

We thank the referee for this comment. We have completely revised this section the introduction in response to this and other comments by the referees.

lines 65: Why is ref 14 cited in this context?

That's indeed a mistake and we'd like to thank the referee for pointing that out.

lines 72-73: "...would lead to declining temperatures (central estimate) in the long term" What are the uncertainties around this estimate?

We are addressing the uncertainties around long-term warming estimates under different emission scenarios in detail in a subsequent section. In order to avoid repetition, we would like to not expand further on this, also noting that we explicitly refer to a high-level IPCC finding.

line 99: "...can give rise to the impression that the temperature overshoot risk under such scenarios is constrained to a few tenths of a degree" Why is ref. 14 cited here? This can give rise to the impression that these authors would have published such misleading statements?

We can see how this referencing could be read in that way - this was not our intent. We have removed the reference here.

line 116: "...determines the pace of temperature reversal" This must be based on some assumptions? I don't have access to the cited publication, but I assume this is based on TCRE and assuming that faster removal will give faster temperature decline?

The paper cited (Rogelj et al. 2019, in Nature) explores this question in great detail and we have been recalling their insights here. The analysis presented in Rogelj et al. is based on the MAGICC6 simple climate model, but a range of different methods, from deploying a TCRE based approach (although TCRE_down may be different than TCRE_up, e.g. Tokarska et 2015) results using more complex models support this very general insight.

line 137-140: It might be good to clarify what assumptions are underlying this statement. One would be that TCRE is constant (and remains constant under negative emissions)?

We have reworded this section to avoid confusion. We note that these are merely the introductory paragraphs for a section that is addressing precisely the issues highlighted by the referee. The paragraph now reads:

“Peak-and-decline (PD) pathways are differentiated by the stringency of emission reduction efforts in the near term and up to achieving net-zero CO2 emissions and the assumed net-negative CO2 emissions in the long term¹⁶. The former determines the maximum cumulative CO2 emissions of a pathway and thereby approximately the magnitude and time of peak warming for median climate outcomes^{6,16}. The latter determines the pace of potential temperature reversal¹⁶. Both aspects are further influenced by non-CO2 emissions. ”

line 143: The authors mean long term "median warming outcomes"?

Thank you. Added 'long-term'.

line 154: "pathway PROVIDE REN_NZCO2" is the use of this acronym necessary here?

It is indeed not. We have adjusted accordingly.

line 194: I find the wording "peat carbon loss from climate and anthropogenic land use change" confusing, maybe "peat carbon loss due to climate change and degradation or conversion of peat lands"?

Thank you for this suggestion for rewording the sentence which we have implemented.

lines 198-202: "...as there is no observational evidence available to constrain the modelled response to net-negative CO₂ emissions. It therefore appears plausible that substantial residual uncertainties about the Earth System response will remain beyond the time net-zero CO₂ is reached, implying the need for a preventive approach to hedge against undesirable warming outcomes in the long term." I agree that it is reasonable to "hedge against undesirable outcomes". I agree less with the statement that "uncertainties will remain beyond the time net zero is reached". Earth system processes are not suddenly fundamentally different if emission were to become negative. The problem is that we have too sparse observations for a too short time periods to constrain some key processes, but up to the time when net-zero is reached we will hopefully have much more complete and longer observational records.

We thank the reviewer for this comment, and of course agree with the reasoning provided that earth system processes are not changing suddenly and that of course the length of the observational record is a key limiting factor here. But also decades of additional observational data (we may be hopeful it won't be centuries until net zero), may not eliminate all uncertainties involved. And so some things we may only find out on the go.

We have streamlined the respective section and deleted the parts the respective parts.

Table 2: "...and may be constrained further by uncertain sink saturation^{38,42–44}" It is unclear to me what this is supposed to mean. Also what is the difference between this and "System feedbacks"?

We thank the referee for pointing that out. Sink saturation in soils and vegetation refers to the fact that the carbon storage capacity of systems has an (uncertain) upper limit even under optimal management (e.g. Fuss et al. 2018). We have reworded the respective sentence to be more clear to:

"and may be constrained further if total sequestration potentials are lower than current best estimates"

This issue is specific to management practices and thereby differs from system feedbacks, that more relate to additional side-effects implied by the management strategies in the first place.

line 286: Strong regional high northern latitude impacts of AMOC slowdown when emissions are phased out or become negative have been also highlighted in a recent paper by Schwinger et al (2022).

Thank you for pointing that out. The whole section has been rewritten and the reference included.

line 386: "...the order of around 0.05°C per decade" - this is due to limitations of the rate of CDR?

Indeed, it's diagnosed from current IAM overshoot scenarios that deploy around NNCE of 10 Gt per year. The sentence has been adjusted to reflect that. Higher rates could in principle be envisioned, but may come with profound technological and sustainability challenges.

The reference to this number, however, has been removed from the manuscript for streamlining purposes.

Section "Time-lagged and irreversible impacts": In this section the topic of comparing net-zero CO₂ and net-zero GHG pathways suddenly comes up. This has not properly introduced and comes a bit out of the blue. Also, it remains unclear why this contributes to overconfidence in overshoot pathways?

We thank the referee for pointing this out and we have revised this section as well as the introduction to clarify this issue. The key comparison here is between a stabilisation scenario, and one of gradual warming reversal.

As we clarify now, net-zero GHGs will (in the median) lead to long-term declining temperatures and thus peak-and-decline pathways. The category of net-zero GHGs is introduced in Table 1 and a distinct feature of pathways that could be seen as Paris Agreement compatible. We therefore consider this distinction to be policy relevant.

Illustrating results as either achieving net-zero CO₂ or net-zero GHG in this section allows us to separate the effects of temperature stabilization vs. long-term decline for long-term risks and thereby substantiate one of our key insights that declining global temperatures can limit long-term climate risks including sea-level rise and cryosphere changes which is illustrated here.

lines 440-442: "As we have shown, whether or not a long-term decline in global temperatures can be achieved depends on uncertain physical climate system feedbacks, but nevertheless needs to rely on large scale carbon dioxide removal." Fig. 2 shows that there is a substantial number of ensemble members that show a temperature decline after peak without CDR, right? If that is correct I don't understand the logic of this sentence ("nevertheless needs to rely on CDR").

We want to thank the referee for correctly pointing out that this sentence is worded somewhat one-sidedly. We have revised the respective section fundamentally.

lines 458-460: "Overshooting of 1.5°C (or even 2°C) is not something that can be planned for with certainty - it is a question of probabilities and might prove impossible due to physical climate system feedbacks under a range of emission pathways." Currently, it does not look like overshooting 1.5 or 2 degrees is impossible... Please reconsider the logic of this sentence.

We agree that this sentence is not very clear and have deleted it.

References:

Bellomo, K., Angeloni, M., Corti, S. et al. Future climate change shaped by inter-model differences in Atlantic meridional overturning circulation response. *Nat Commun* 12, 3659 (2021).

<https://doi.org/10.1038/s41467-021-24015-w>

Kim, SK., Shin, J., An, SI. et al. Widespread irreversible changes in surface temperature and precipitation in response to CO2 forcing. *Nat. Clim. Chang.* 12, 834–840 (2022).

<https://doi.org/10.1038/s41558-022-01452-z>

King, A.D., Lane, T.P., Henley, B.J. et al. Global and regional impacts differ between transient and equilibrium warmer worlds. *Nat. Clim. Chang.* 10, 42–47 (2020). <https://doi.org/10.1038/s41558-019-0658-7>

<https://doi.org/10.1038/s41558-019-0658-7>

Schwinger, J., Asaadi, A., Goris, N., and Lee, H. (2022): Possibility for strong northern hemisphere high-latitude cooling under negative emissions, *Nat. Commun.*, 13, 1095, doi:10.1038/s41467-022-28573-5

Schwinger, J., Asaadi, A., Steinert, N. J., and Lee, H. (2022b): Emit now, mitigate later? Earth system reversibility under overshoots of different magnitudes and durations, *Earth Syst. Dynam.*, 13, 1641–1665, doi:10.5194/esd-13-1641-2022

Referee #2 (Remarks to the Author):

This paper provides an in-depth and comprehensive overview of a very important and timely policy-relevant topic, the understanding the implication of overshoot scenarios. The authors point out that the current framing of overshoot scenarios is misleading as it implies that declining temperature after a peak is roughly equivalent stabilizing at that peak. However, they point out that different overshoot pathways could lead to significantly different climate outcomes with important implications for climate adaptation and carbon dioxide removal (CDR) requirements.

This paper provides critical new perspectives that are needed to inform the emerging policy and business discussion on assessing climate risks and crediting CDR. I would encourage this to be published.

We thank the referee for this positive assessment of our work.

Here are a few suggestions to improve clarity in the paper.

(1)The paper is very dense, and thus a bit hard to get through. Given the importance of the topic I would suggest two options to guide the reader through

a.) Explicitly outline implications (with headings or a table) for: climate adaptation, Mitigation targets and remaining carbon budget?, and CDR targets and allocation.

b.)Consider having a news or commentary piece published alongside this to clarify the significance of the messages emerging from this work.

We thank the referee for this suggestion. We have reflected about the suggestion, but unfortunately we have very limited room for additional content or display items (in fact, we already need to scale back from the existing ones). We tried to implement the suggestion a) as part of Fig. 1 panel b and have put further emphasis on this now.

We have also taken the comment of the referee to heart when revising and streamlining the manuscript to ensure clearer messaging, in particular in the discussion section.

The suggestion for a commentary piece is not for us to decide upon, but we would of course view this suggestion very positively.

(2)Figure 5, part b. I do not understand what the different bars and colors represent in part b of this figure. What do the seven bars under lifetime vs four under planning horizon. And what are the significance of the different discount rates shown - Are these just illustrative? Suggest you provide more explanation in the caption and text.

We thank the referee for this suggestion. Indeed, Fig. 5 b is illustrative in nature. We have revised the caption to clarify the meaning of the bars and the discount rates shown. We have also revised the accompanying main text. The caption of Fig. 5 now reads (changes highlighted in **bold**):

b, A stylised illustration of adaptation relevant timescales starting in 2030 including different lifetimes of individual adaptation measures (from years to decades, **horizontal bars**), the planning horizons for adaptation planning (decades) and the effect of applying discounting (reflecting societal preferences towards intergenerational equity) to future damages and adaptation benefits. We show the effect of discounting for three illustrative discount rates.

(3)CDR implications. The analysis of this paper has significant implications for the how to evaluate the need, contribution and benefits of CDR. The discussion around the implications for CDR could be more succinctly outlined, in particular the idea of Preventive CDR capacity. Lots of important issues are raised, but do not come through as clearly as they could.

We thank the referee for pointing this out. Ref #3 also has similar remarks. We have reworked the respective parts of the discussion section as well as the abstract to further strengthen this specific point.

(4) Climate adaptation implications. I would suggest providing a bit more detail on the implications of not taking this more nuanced approach to overshoot pathways. I realize that in some ways that what I am suggestion could be a whole new paper. But a bit more unpacking of this, and the CDR would help to further clarify the importance the analysis outlined in the paper.

We thank the referee for this suggestion that we fully agree with. In order to address it we have decided to substantially revise the section on Adaptation decision-making and overshoot, and removed the illustrative example explored included in the supplementary material in order to allow for a more elaborate discussion of the central insights as suggested by the referee.

Referee #3 (Remarks to the Author):

With minor revisions I would recommend publication. This article adds new rationales (rooted in modelling of overshoot scenarios) for reserving (limited, sustainable) CDR capacity for drawdown, rather than offsetting residual emissions; and for accelerating emissions reductions in the immediate future. It highlights uncertainties related to the extent and climatic implications of temporary overshoot of climate temperature goals, and worrying implications for adaptation and loss and damage activities from additional committed climate impacts if overshoot pathways (even those within the 0.3C margins considered by IPCC) are permitted. I see no fundamental flaws, although it does exhibit a bias towards modelling and modelling literatures which limits its engagement with other literatures examining the same challenges, and its policy relevance.

We thank the referee for this positive assessment of our work and in particular for pointing us to a significant body of relevant work that has previously been underrepresented in our manuscript.

General comments

The opening summary of the paper offers several broad conclusions, some generally accepted in scientific circles, but meriting repetition for policy makers (notably 'emphasize early emissions reductions' and 'scaling CDR cannot be guaranteed'), and some more novel: notably that climate impacts will differ substantially in a 'post-overshoot' world from those where overshoot is avoided. As I read it, the central recommendations are two fold. Both are valid, but neither element is presented with the clarity it deserves. First, emissions reduction should be accelerated as quickly as possible ("stringent immediate" both lacks specificity, relative to current efforts, and is easily dismissed as politically impractical).

Second, while CDR capacity should be developed (within social and sustainability constraints), it should not be allocated to counterbalance residual emissions (or at least, not those that can foreseeably be mitigated). The political implications of both these are profound, but remain somewhat opaque in the present text. For example, carbon trading and offsetting markets do not support these goals, to name but one climate canard they would fell.

We thank the referee for these very thoughtful suggestions. In response, we have revised the conclusion section considerably to increase clarity with regards to the central insights and resulting policy recommendations. Specifically, we also discuss the implications with regards to offsetting.

We hope that this helps to address the very valid concern by the referee.

In part I fear this lack of clarity regarding political and policy implications is a product of a bias towards the production of climate knowledge through modelling, and a tendency to cross-citation amongst scholars using such techniques, which leaves other relevant literatures and epistemologies under-represented, even where they have reached similar conclusions in advance of this author group. I would encourage the authors to undertake some minor revisions to highlight political implications, and to cite relevant literature which supports their analysis from different foundations. In reflecting on the effects of a modelling orientation, they might find Ho et al. (2019) useful.

We thank the referee for pointing out a systemic bias in our incorporation of relevant interdisciplinary climate scholarship. We have substantially revised the relevant sections and, despite the need to cut the initial reference list by about half, taken on board many of the referee's excellent suggestions for novel literature. We are highly appreciative of the substantial input provided by the referee in this regard.

The introduction in particular merits careful redrafting in particular, and the production of a dedicated abstract which reflects the needed clarity. At present some of the conclusions presented in the introduction are phrased in ways that seem contradictory: for example, while arguing that the technical development of CDR cannot be relied upon, the authors propose that a preventive capacity of hundreds of Gt should be sought nonetheless. Reading ahead I can see that they mean to suggest that if CDR were developed to the scales foreseen in many overshoot pathways, it may well exceed social and sustainability constraints, yet still might not be adequate to significantly reduce temperatures in an overshoot world, due to feedbacks (or tipping points) in warming systems). But this then makes the phrasing of another conclusion peculiar "irrespective of the peak warming, we find declining global temperature remains critical to reducing climate risks". Clearly this is true in abstract, but the previous finding suggests rather that with rising peak temperatures the nature of those risks changes, and the prospect of delivering temperature decline may be reduced. And later, the text indicates many risks which are more directly related to peak warming than to the rate or scale of subsequent temperature decline. This section therefore merits careful attention to ensure consistency and clarity.

In response to the referee's comment, we have substantially revised the introduction as well as abstract. We hope the interdependencies, and potential contradictions, that the referee has been spelled out so well are now also reflected better in our manuscript.

Further, the argument for "redirecting the discussion towards managing climate

risks both in the near and long-term” is rather vague and ambiguous. Perhaps the authors intend to suggest that both near and long-term risks should be considered, and this means both cutting emissions faster now, and developing a preventive (sustainable) CDR capacity (as opposed to doing one or the other). By contrast the current phrasing implies shifting attention from emissions to risks, an approach which might be expected to encourage consideration of SRM interventions, given existing ‘risk-risk’ literature in this space (eg Felgenhauer et al. (2022).

I’d strongly recommend that the authors redraft their conclusions to ensure clarity regarding these matters for both scientific and policy readers.

We thank the referee for critically reviewing our argumentation and pointing us to unintended implications of the risk framing used here. This is incredibly helpful. We have no intention to imply a ‘risk-risk’ framing the way it is being introduced in discussing SRM approaches and have revised the conclusion fundamentally to both address risks of SRM (as highlighted by the referee further down), as well as to avoid implying a risk-risk framing altogether.

Minor Comments

There are several other sections that merit minor revisions for clarity on approach, implications or limitations.

1. Categorising ‘peak and decline’ pathways. This is a useful effort to relate overshoot to the characteristics of the majority of policy relevant emissions pathways. However in some respects it seems detached from the most critical aspects of policy relevance in the existing and proposed net-zero pathways adopted by the majority of states globally. Being explicit that existing net-zero pathways do not necessarily avoid overshoot would seem critical. Scholarly and policy debate about ‘the area under the curve’ (the cumulative emissions allowed between now and net-zero) is intense and significant (e.g. Armstrong and McLaren, 2022; Fankhauser et al., 2022; Kaya et al., 2019; Sun et al., 2021). Making such connections would add to the policy utility of this article.

We thank the reviewer for calling for clarity here, which we agree is very useful to provide. We added the following paragraph to the respective section to clarify that point.

“Peak-and-decline (PD) pathways are differentiated by the stringency of emission reduction efforts in the near term and up to achieving net-zero CO₂ emissions and the assumed net-negative CO₂ emissions in the long term¹⁶. The former determines the maximum cumulative CO₂ emissions of a pathway and thereby approximately the magnitude and time of peak warming for median climate outcomes^{6,16}. The latter determines the pace of potential temperature reversal¹⁶. Both aspects are further influenced by non-CO₂ emissions. ”

2. Use of the FaIR model. I am not a modeller, but consider myself reasonably well versed in the limitations of modelling. I find it inappropriate to specify the choice of model without raising caveats about the extent to which the additional simplifications of FaIR over full ESMs might themselves generate excessive certainty in the output set. For example the asserted ‘finding’ at lines 175-9 would appear to be an artefactual product of the choice of model and input constraints. The discussion of FaIR limitations at 188-192 is helpful, but comes too late, and is incomplete with respect to the input constraints described.

We thank the referee for this important comment. This is similar to a comment by Ref #1 above. In response to both comments, we have added additional analysis in Extended Data Fig. 1 of the manuscript that illustrates the consistency of the FaIR version used with key AR6 assessed diagnostics, namely ECS, ZEC and the historic warming record, in line with the extended evaluation of the calibrated FaIR versions used here in the Supplementary Information for Chapter 7 of the AR6 Working Group 1 report (Forster *et al* 2021). We added an additional section describing the evaluation in the Methods. We find very good agreement across the distribution for these key diagnostics against the AR6 assessment.

We hope that this provides further confidence in our findings and addresses the very valid concern of the referee.

3. Calculating and discussing NNCE figures merits greater specificity. Scholarly work on ZEC has highlighted a need to be clear about the distinctions between the response of natural sinks to net-zero; the role of enhanced natural sinks (e.g. through reforestation); and the role of specifically anthropogenic removals. This is of intense policy relevance as some countries seek to redefine natural sinks as removals in their net-zero calculations. I recommend a brief clear definition of what is included in ‘CDR requirements’ or NNCE in this context, which would also provide a critical opportunity to note that ‘requirements’ calculated in this way may easily exceed capacities which could be deployed in socially just and environmentally sustainable ways.

We thank the referee for pointing this out, and highlighting the important difference between removal accounting in climate models and under national GHG inventories. We very much agree

that this requires clarification and have added the following paragraph to the respective section in response:

CDR requirements here refer to additional carbon removal due to anthropogenic activity in line with the conventions and definitions of the models underlying our assessment. It is important to note that parties to the UNFCCC use a different definition for defining land-based carbon fluxes, which results in a ~4-7 Gt CO₂/yr difference between national GHG inventories and scientific models that needs to be considered when translating these insights into policy advice²⁰.

4. Constraints to CDR capacity (lines 231-248 and table 2). This discussion seems somewhat ad-hoc, and lacking in a rigorous framework to ensure comprehensive and balanced treatment. In particular, in the main text, sustainability and equity concerns appear as a tag-on to technical and climate feedback issues. This perhaps reflects a bias in the disciplinary skills of the authors, but should be rectified (relevant examples include: Waller et al., 2020; McLaren, 2012; Smith et al., 2015; Cox et al., 2018; Forster et al., 2020). The table is better (with more diverse and complete issues and sources), although still particularly poor on the issue of mitigation deterrence (here still described with the outdated terminology of 'moral hazard') which has a rich recent literature (e.g. McLaren, 2020; McLaren et al., 2021; Carton et al., 2023). Leakage of captured carbon into utilisation in CCUS approaches, or enhanced oil recovery is totally overlooked even though dominant in actual existing engineered carbon capture. Uncertainties relating to marine CDR are not mentioned (e.g. Boettcher et al., 2021; Mengis et al., 2023).

We thank the reviewer for highlighting important publications, which we had not cited in the previous version of our manuscript. While the pool of evidence on carbon dioxide removal is rapidly growing and many important aspects linked to CDR constraints are already documented in the literature, we need to restrict ourselves to a few key references in order to stay within the journal's reference limit. Nevertheless, we have now added several of the suggested references to our revised manuscript and edited the text to be more explicit concerning mitigation deterrence and other constraints linked to ocean-based CDR.

Our discussion of constraints is limited to carbon dioxide removal as defined by the IPCC, which is why we did not elaborate on constraints linked to CCS, CCU or CCUS.

5. The discussion of SLR, permafrost and peatland commitments from overshoot (lines 327ff) seems over-simplified. It is important to demonstrate that these risks will continue to grow as a result of overshoot, but it seems highly unlikely that they will respond in the direct linear fashion suggested

here. In particular, SLR responses will presumably be highly conditional upon the tipping points of ice-sheet dynamics. For permafrost thaw, there may be more linearity, but relationships with ocean-current changes could introduce significant non-linear responses. For peatlands, the relationship between precipitation changes, new growth and methanogenesis is far from simply linear. The simplifications used here must be clarified, along with noting the relevant uncertainties. I'd also question whether reporting impacts in terms of 'per 100 years overshoot' is appropriately consistent with the rest of the article, which (eg fig 5) presents the implications of overshoot measured in decades (60-70 years).

We thank the referee for pointing to this indeed somewhat surprising result of a linear scaling with overshoot length. But we note that this is not merely an 'over-simplification', but an actual finding of our analysis. The underlying modelling does indeed include potential tipping dynamics of ice-sheets (see e.g. (Mengel *et al* 2018) as well as non-linear peatland dynamics (Qiu *et al* 2022), including for high risk outcomes (compare Extended Data Fig. 9)

We also find that this near-linear scaling holds for individual GHG contributions from peatlands and permafrost as shown in the added extended figure.

We, however, agree that this linear scaling might indeed be related to the specific ensemble of scenarios used, and in particular the gradual, and across the ensemble, homogeneous CDR deployment. We have now revised the respective section of the manuscript to caveat this finding accordingly:

We caution that the diagnosed linear relationship between overshoot length and impact outcome may depend on the set of pathways that it was derived from. The underlying pathways assume overshoots starting from a period of delay in climate action followed by a steady reduction to net zero GHG emissions implying a similar rate of long-term temperature decline in all pathways. The relationship could be different for more, or less extreme overshoot outcomes.

6. SRM dilemma. Overshoot is a concept widely used in geoengineering literature, and the risks and uncertainties it involves are seen by some as offering a clear justification for SRM interventions (e.g. MacMartin *et al.*, 2018). SRM is mentioned briefly at line 445, but not in the discussion of particular regional effects of overshoot, some of which might (theoretically) be susceptible to amelioration by tailored or modulated SRM interventions (eg the ITCZ shift – (see e.g. Haywood *et al.*, 2013: for discussion of asymmetry and SRM)). I am highly sceptical of the technical and political capacity to

deliver such interventions (see McLaren, 2018), but the authors might be advised to consider such possibilities.

We thank the reviewer for making that important point. In fact the Baur et al. paper we cited does address some of the areas of ‘overconfidence’ in how the overshoot peak shaving idea is sometimes stipulated. In response to this comment, we have now expanded on this issue considerably in the discussion as follows:

It has been argued that climate impacts during overshoots could be reduced or masked by the deployment of solar geoengineering (SG) intervention techniques⁴⁵ that would temporarily cool the planet. This idea is referred to as peak shaving. These suggestions, however, make strong assumptions about the applicability, effectiveness and governance of SG interventions. Accounting for uncertainties in the physical climate response, and in the evolution of future emissions after SG is deployed, implies that a SG intervention aimed at peak-shaving an overshoot could result in a multi-century commitment of both SG and CDR deployment²³. In addition to fundamental concerns about SG deployment in isolation⁴⁶, a peak-shaving discourse is prone to the same overconfidence in reversibility and effectiveness we have conceptualised in this article.

7. In contrast to the summary text in the introduction, the material in the concluding section (line 477ff) is clear that the aim of preventive CDR capacity is to develop significant capacity within sustainability and social constraints which is not allocated to counterbalance residual emissions, but is dedicated to delivering drawdown of atmospheric GHG concentrations. In this they echo substantial scholarship on CDR, Net Zero and residual emissions (e.g. McLaren et al., 2019; Buck et al., 2023; Armstrong and McLaren, 2022) (see also Ho, 2023: <https://www.nature.com/articles/d41586-023-00953-x>). It would seem appropriate to acknowledge this prior scholarship, and not imply that the conclusions here are novel.

We thank the referee for pointing that out and have revised the introduction and in particular the concluding section accordingly, clearly acknowledging these points and previous scholarship. As we have to be highly selective with references to be included, we were not able to acknowledge all the previous scholarship highlighted by the referee, but tried to make an authoritative selection. Among other edits, we specifically added the following paragraph:

Second, we suggest that there is a need to prepare for an environmentally sustainable CDR capacity to hedge against long-term high-risk outcomes resulting from stronger than expected climate feedbacks. We find that such a preventive CDR capacity might need to be of

the order of several hundred gigatonnes of cumulative NNCE , a scale that might be just about possible within sustainable limits of CDR deployment⁹. Designed as a preventive measure, this must not be planned to counterbalance residual emissions, leaving little room for CDR deployment for offsetting of residual emissions⁴⁷. This further underscores the importance of very stringent near-term emission reductions to limit long-term risks⁴⁸.

In addition some minor amendments may be merited at the following points

Line 53-4 I'd note that Azar et al were far from the earliest to identify a key role for large scale CDR / NETs / GGR in stabilizing and reversing temperatures. For example the UK Climate Change Committee already commission research on this before 2010. Azar et al may not have even been the first to model this (and these authors should be well aware that modelling does not hold a monopoly on relevant climate knowledge ...)

The reviewer is actually right. The concept of BECCS goes way back, and e.g. Keith and Rhodes were presenting first work on BECCS by 2000. We also note that Wigley et al. (1996) also already include negative emission scenarios and have adjusted the referencing accordingly.

Line 138 The word 'determine' is inappropriate here, given possible hysteresis in response to rising and falling GHG concentrations – other factors are therefore also relevant (as recognised in lines 146ff). Using 'determine' risks a similar form of 'overconfidence' in projections that the authors seek to warn against.

We have revised the respective section.

Line 169-70 the description of the results in the upper right quadrant is ambiguous: the terminology used applies to the entire right hand part of the chart (both upper and lower quadrants)

Revised the respective section. Relevant sentence discussing these elements now reads:

“We also identify relatively higher risk futures, where warming exceeds 1.5°C at net zero CO₂ and continues beyond (Fig. 2b, top right quadrant).”

Line 220-221 the choice of ‘conventional’ and ‘novel’ as categories of CDR is not established in the literature: it is a deliberate effort in the non-peer-reviewed report cited to establish new discursive categories. It should therefore not be used unquestioningly. It may indeed be preferable to distinctions based on capture method (biological vs chemical; or natural vs engineered), location (terrestrial, oceanic etc), or final storage location (biologic, geologic, oceanic) for example. But categories matter (Heyward, 2013), and the choice must be justified. This choice, for example appears to treat all marine or oceanic methods as ‘novel’, lumping them in with engineered, and hybrid approaches such as BECCS and DACCS.

We fully agree that a focus on median warming outcomes is insufficient to assess CDR requirements in overshoot pathways and have explored this in detail in the previous section. Here, however, we are making factual statements about the ensemble spread in IPCC AR6 WG3 pathways (not considering physical climate uncertainty). We have revised the section considerably to streamline and increase clarity and removed the references to ‘conventional’ and ‘novel’ CDR as this is not the main focus of our analysis here.

Line 222-226 The description of the pathway sets is rather misleading, as the central estimates for low or no overshoot pathways are exaggerated by the asymmetric effects of geophysical uncertainty, thus disguising the significant additional CDR typically demanded in overshoot pathways. The effect is to undermine one of the key arguments: that reliance on CDR should be minimised by minimising near-term and thus cumulative emissions.

We fully agree that a focus on median warming outcomes is insufficient to assess CDR requirements in overshoot pathways and have explored this in detail in the previous section. Here, however, we are making factual statements about the ensemble spread in IPCC AR6 WG3 pathways (not considering physical climate uncertainty). We have revised the section to increase clarity:

“Scale-up of CDR is most rapid in pathways with the lowest peak warming (low or no overshoot 1.5°C pathways, C1, Extended Fig. 3). Across the ensemble of emission pathways, CDR levels by the end of the century are generally higher in high overshoot C2 pathways, but the full (5-95%) range is similar to the C1 pathways range. Pathways that keep warming below 2°C, but do not limit warming to 1.5°C in 2100 (C3) see a substantial CDR ramp-up in the second half of the 21st century reaching levels comparable to C1 pathways by 2080 (Extended Fig. 3).”

Line 227-9 seems to be a restatement of the point that cumulative emissions are the main factor, but phrased in a way that is hard to parse. This sentence could probably be deleted without any loss.

This sentence has been removed.

Line 273 ff presents a series of projected regional effects. These are all large scale, and likely very plausible, but should be prefaced with some caveats about the capacity of the relevant models (CMIP group ESMs) to project regional effects, the relevant degrees of uncertainty thus associated with such projections. Some caveats are given in the caption for Figure 3, but not explained. These effects and the figure appear to be at least partly derived from a reference which is incomplete/not yet published (note 65): please confirm that they are based on peer-reviewed material.

In response to comments by Ref #1, this section has been reworked in its entirety and new analysis has been added. We hope that these changes also address the concerns raised by the referee. We also note that the reference in question on overshoot analysis based on CMIP6 (which has been moved to the Extended Data Figures now), has since been accepted.

Line 355-358 – the conceptualisation of vulnerability here seems naïve. As I read this text vulnerability is seen as something abstract and objective, a product of a paucity of economic development. In reality, vulnerability is as much a social construct, a product of uneven development and the construction of sacrifice zones and economic precarity in the interests of extractive economic growth. While the authors may not feel qualified to examine such challenges, they should at least acknowledge that there are radically different ways of conceptualising vulnerability that do not lead to a presumption or impression that further economic development is the only, or best way to alleviate vulnerability. In considering how to ‘reframe’ overshoot, the question of how vulnerability to climate impacts might be constructed, or deconstructed might be a valuable additional element.

We share the reviewer's view that vulnerability is indeed multifaceted and socially constructed. We did not intend to imply otherwise. Unfortunately, we face a difficult word-count constraint, as we cover many other fields in this analysis, so we are not able to explore dimensions of social construction and deconstruction of vulnerability. We agree this direction of inquiry has great merit for further studies. Nonetheless we have edited the text to clarify that vulnerability is not simply a lack of socio-economic development, but depends on the nature of that development (italics indicate additions):

An overshoot above 1.5°C would likely emerge during the first half of the 21st century, a period still characterised by comparably low adaptive capacity in large parts of the globe even under optimistic scenarios of socio-economic development³⁸. The coincidence of overshoot and low adaptive capacity can amplify climate risks. This has profound consequences for the ability to achieve climate-resilient and equitable development outcomes under overshoot in particular for the most vulnerable countries, communities and peoples.

Line 397 ff It would seem inconsistent to imply that the uncertainties relating to overshoot and its impacts make cost-benefit calculations inappropriate or misleading for CDR development and mitigation choices, but to then retain a conventional cost-benefit approach to adaptation to the same uncertainties.

The point with respect to CBA in the context of mitigation is very well taken. Yet, for adaptation projects the environment is somewhat different as individual projects and their implementation will apply CBA with limited scope and focus on a small scale problem and investment. Given the prominence of CBA in adaptation decision making in general, we therefore still think this is useful to clarify and would suggest to retain the reference.

Line 492-496 This paragraph adds nothing, and risks confusing the reader through oversimplification. It could be deleted without loss.

We thank the referee for pointing that out, but feel it would be inappropriate to end the manuscript with a paragraph discussing some policy and pathway design implications. We would therefore like to retain this paragraph and hope that the referee may find this acceptable.

References

Armstrong C and McLaren D (2022) Which Net Zero? Climate Justice and Net Zero Emissions. *Ethics and International Affairs* 36(4): 505-526.

Boettcher M, Brent K, Buck HJ, et al. (2021) Navigating Potential Hype and Opportunity in Governing Marine Carbon Removal. *Frontiers in Climate* 3(47).

Buck HJ, Carton W, Lund JF, et al. (2023) Why residual emissions matter right now. *Nature Climate Change*. DOI: 10.1038/s41558-022-01592-2.

Carton W, Hougaard I-M, Markusson N, et al. (2023) Is carbon removal delaying emission reductions? *WIREs Climate Change* 14(4): e826.

Cox E, Pidgeon N, Spence E, et al. (2018) Blurred Lines: the Ethics and Policy of Greenhouse Gas Removal at Scale. *Frontiers in Environmental Science* 6.

Fankhauser S, Smith SM, Allen M, et al. (2022) The meaning of net zero and how to get it right. *Nature Climate Change* 12(1): 15-21.

Felgenhauer T, Bala G, Borsuk M, et al. (2022) Solar Radiation Modification: A Risk-Risk Analysis. Reportno. Report Number |, Date. Place Published | : Institution |.

Forster J, Vaughan NE, Gough C, et al. (2020) Mapping feasibilities of greenhouse gas removal: Key issues, gaps and opening up assessments. *Global Environmental Change* 63: 102073.

Haywood JM, Jones A, Bellouin N, et al. (2013) Asymmetric forcing from stratospheric aerosols impacts Sahelian rainfall. *Nature Climate Change* 3(7): 660-665.

Heyward C (2013) Situating and Abandoning Geoengineering: A Typology of Five Responses to Dangerous Climate Change. *PS: Political Science & Politics* 46(1): 23-27.

Ho E, Budescu DV, Bosetti V, et al. (2019) Not all carbon dioxide emission scenarios are equally likely: a subjective expert assessment. *Climatic Change* 155(4): 545-561.

Kaya Y, Yamaguchi M and Geden O (2019) Towards net zero CO₂ emissions without relying on massive carbon dioxide removal. *Sustainability Science*. DOI: 10.1007/s11625-019-00680-1.

MacMartin DG, Ricke KL and Keith DW (2018) Solar geoengineering as part of an overall strategy for meeting the 1.5°C Paris target. *Philosophical Transactions of the Royal Society A: Mathematical, Physical and Engineering Sciences* 376(2119).

McLaren D (2012) A comparative global assessment of potential negative emissions technologies. *Process Safety and Environmental Protection* 90(6): 489-500.

McLaren D (2018) Whose climate and whose ethics? Conceptions of justice in solar geoengineering modelling. *Energy Research & Social Science* 44: 209-221.

McLaren D (2020) Quantifying the Potential Scale of Mitigation Deterrence from Greenhouse Gas Removal Techniques. *Climatic Change* 162: 2411–2428.

McLaren D, Willis R, Szerszynski B, et al. (2021) Attractions of delay: Using deliberative engagement to investigate the political and strategic impacts of greenhouse gas removal technologies. *Environment and Planning E: Nature and Space* 0(0): 25148486211066238.

McLaren DP, Tyfield DP, Willis R, et al. (2019) Beyond “Net-Zero”: A Case for Separate Targets for Emissions Reduction and Negative Emissions. *Frontiers in Climate* 1: 4.

Mengis N, Paul A and Fernández-Méndez M (2023) Counting (on) blue carbon—Challenges and ways forward for carbon accounting of ecosystem-based carbon removal in marine environments. *PLOS Climate* 2(8): e0000148.

Smith P, Davis SJ, Creutzig F, et al. (2015) Biophysical and economic limits to negative CO₂ emissions. *Nature Climate Change* 6: 42.

Sun T, Ocko IB, Sturcken E, et al. (2021) Path to net zero is critical to climate outcome. *Scientific Reports* 11(1): 22173.

Waller L, Rayner T, Chilvers J, et al. (2020) Contested framings of greenhouse gas removal and its feasibility: Social and political dimensions. *WIREs Climate Change* 11(4): e649.

Reviewer Reports on the First Revision:

Referee #3 (Remarks to the Author):

I thank the authors for their detailed and thoughtful responses to my comments and suggestions.

I am content with the amendments made, recognising the limitations of word and reference counts, and would now recommend publication.

I note one amendment which requires minor further clarification: on p11 the text now reads "We explore NNCE requirements for an illustrative pathway with the following characteristics (Fig. 2a): (1) it achieves net -zero CO₂ around mid-century, (2) limits and aims to limit median peak warming close to 1.5°C above pre-industrial levels, and (3) has no NNCE." This appears contradictory (to explore NNCE requirements for pathways with no NNCE) although on reading further it becomes clear that the authors mean no 'initial' inclusion of NNCE. I trust this can be easily clarified.

Referee #4 (Remarks to the Author):

The referees' comments are mostly well addressed, but additional responses for the comment of Reviewer #1 on the term "substantially different" are needed. The authors provided some explanations in terms of statistical significance of regional changes, however it is still unclear how much changes were quantitatively observed from the analysis, and whether or not it is sufficient to claim "substantial".

In addition, even if the current explanation is sufficient to assert that the changes are substantially different in the regional level, the abstract still claims "global and regional climate change... are substantially different". The current responses look insufficient to support in terms of global changes anyway.

Additional explanation on these points are needed, or please consider rephrasing the words in the abstract.

Minor comments:

Please provide the definitions of the whiskers and outliers for the AR6 scenarios presented in Fig 2c. Is it 5-95th percentile range?

The link <https://zenodo.org/record/5886912> is provided as the source data for Fig. 2, but I found it links to the IPCC AR6 scenario database. Although I did not check the contents of this database carefully, does it really include the PROVIDE scenario data? Otherwise please provide a correct location.

Temperature unit is presented in Kelvin in Fig4, whereas other figures and texts use degrees. Are there any reason?

Referee #5 (Remarks to the Author):

Coming in as an additional reviewer, I attempted to follow both the previous reviewer's logic as well as the authors' responses to the best of my abilities. Please feel free to point out potential misconceptions on my end, yet I hope that a set of fresh eyes might be helpful after such a considerable revision.

That being said, the authors have made substantial changes to the manuscript in response to the reviewer's comment and made a huge effort to address all of the reviewer's concerns. Especially the inclusion of two additional model simulations as a response to referee #1 concerns on the regional reversibility experimental design are impressive, but maybe not the most relevant or scientifically meaningful.

My impression of the manuscript is that it consists of two interesting (but somewhat individual) storylines, and therefore fails to follow a red thread throughout (this might be related to the revision). I get this impression already since the argument structure and logic outlined in the abstract and the conclusion ...

(storyline 1: no perfect reversibility of climate impacts -> time lines of OS not the relevant metric for adaptation planning;

storyline 2: declining temperatures likely alleviate long-term/time-lagged climate impacts -> NNCE arise in case continued warming post net-zero -> there are techno-economic, social-political limitations to CDR deployment -> CDR should be foremost considered as preventive/back-up measure for possibility of high-risk futures)

... is different from the logic that the manuscript follows.

This goes to the point of the referees made about there being internal inconsistencies in the logic concerning adaptation and impacts considerations in the manuscript. After reviewing the spatial reversibility results, I would suggest removing this part from the manuscript, since the evidence unfortunately is not there yet to discuss spatial reversibility of overshoot scenarios. I would strongly suggest that the authors refocus the manuscript and expand on the second storyline.

Below, I am outlining where I think further adjustments might be necessary to address the reviewer's concerns, as well as some of my own comments.

Introduction:

Table 1: I find the two conceptual PD categories as described in the table rather vague, overlapping and it is unclear how and why they are/should be distinct from the ones already used in the literature. It is not specified what overshoot is referred to in the PD-OS (I am assuming temperature, but would argue that carbon budget would actually be the measure). It is not clear to me if the two categories are distinct through their near-term mitigation action, or the level of NNCE? When would we move from one category to the other. How is their narrative different from the highOS vs lowOS one? After reading the introduction, this becomes somewhat clearer, but this

should be reflected in the table, which is meant as a help for the reader (i.e., mention maximum cumulative CO2 emissions, and magnitude of NNCE, but again also when are we in one or the other category?).

Fig. 1: Shading/Coloring in Fig 1 is confusing. From the four color shades on the right, the reader would look for four pathways, but the PD: Peak and decline pathway is a descriptive category of the other two PD pathways. And then, if I understand correctly, the PD-OS and the Pathways showing continuous warming are the same but with different assumptions on Earth system feedbacks. Which case is figure 1b discussing?

Regional climate change reversibility:

While I see the immense effort the authors have put into revising this section, I remain unconvinced of the analysis due to two points:

- 1) You are displaying and comparing one result each of two vastly different experimental designs, even though there is data for multiple models from both experiments. The biggest concern regarding this figure is the implicit comparison of ZECMIP and AREA-MIP results; the former being an idealised CO2-only experiment aiming for the same carbon budget (MacDougall et al., 2020), and latter being an fully-forced experiment, including non-CO2 GHGs, aerosols, and land-use changes, that diagnoses CO2 emissions trajectories for the same end of the century temperature outcome (Silvy et al., 2024, in review). Displaying and comparing spatial patterns of temperature and precipitation with two experiments one including, one excluding time and spatial-varying aerosol patterns, and finding that there are substantial differences is not scientifically sound.
- 2) None of the experiments actually includes the impact from large-scale CDR deployment within their scenarios. An argument that is lacking throughout the entire manuscript, but would be one key to linking these analysis and results to the rest of the manuscript.

Adaptation decision-making and overshoot:

Figure 5: Based on the points raised by one of the referees, I think the usefulness of this illustrative figure remains to be substantially improved. It remains unclear what the horizontal bars actually stand for. The authors clearly have a basis for the length of the different bars, so why not provide the reader with this information (i.e. label the bars) rather than giving a repetition of the ylabel axis. In the same way, a label for the time scales displayed (i.e. a xlabel that shows years, decades, centuries) for figure b would be helpful for the reader.

I also suggest to re-label the first bars to “Median overshoot length and magnitude” along the logic of the manuscript.

For Figure 5a, I think displaying some form of probabilities of the two scenarios would be helpful for linking this figure to the adaptation discussion of high-risk/low-probability events (see comment below).

For adaptation decision making high-risk/impact-low-probability outcomes are considered, so I would argue that next to time scales it is also the level of peak warming that is more relevant, i.e. this is where the peak warming level is more important than long-term temperature outcomes. I

am missing the notion of such considerations in this section all-together, even though I find them highly relevant for the remaining arguments made in the manuscript including not betting on median outcomes.

The other aspect that is relevant for the strength of climate impacts and according adaptation is the rate of change (e.g., William et al., 2021), I would like to invite the authors to take this into consideration in the manuscript, since it would again align very nicely with the storyline and outline what kind of impacts are relevant for overshoot considerations.

Reframing the overshoot discussion:

The innovative conclusion of the manuscript of the development of a sustainable CDR capacity that is independent of the discussion of remaining emissions, comes from my understanding mostly from the possibility that declining global mean temperatures are likely beneficial to the lagged climate impacts and the analysis of the NNCE needs using the FAIR climate model. This suggestion is interesting, yet the authors do then fail to discuss the actual implications this suggestion would have. The discussion of what an environmental (and maybe also societal) level of sustainable CDR would entail, how the authors would define sustainable, or how to get to the information on sustainable levels of CDR and the associated potentials.

Also when talking about adaptation and time scales, I am missing this information wrt the environmental sustainable, preventive CDR pool. Where in figure 5 would this pool lie? Would this CDR pool be deployed in a timely enough fashion to hedge against impacts, given the current deployment gap?

Lastly, I feel like there is a disconnect between the title and the content of the manuscript including the main conclusions (i.e. preventive CDR pool). I would like to invite the authors to revisit the focus of the manuscript and then accordingly adjust the title.

Refs:

Williams, J.W., Ordonez, A. & Svenning, J.C. A unifying framework for studying and managing climate-driven rates of ecological change. *Nat Ecol Evol* 5, 17–26 (2021).
<https://doi.org/10.1038/s41559-020-01344-5>

Hwang, Y. T., Frierson, D. M., & Kang, S. M. (2013). Anthropogenic sulfate aerosol and the southward shift of tropical precipitation in the late 20th century. *Geophysical Research Letters*, 40(11), 2845-2850.

Williams, K. D., Jones, A., Roberts, D. L., Senior, C. A., & Woodage, M. J. (2001). The response of the climate system to the indirect effects of anthropogenic sulfate aerosol. *Climate Dynamics*, 17, 845-856.

Mitchell, J. F., Johns, T. C., Gregory, J. M., & Tett, S. F. B. (1995). Climate response to increasing levels of greenhouse gases and sulphate aerosols. *Nature*, 376(6540), 501-504.

Charlson, R. J., & Wigley, T. M. (1994). Sulfate aerosol and climatic change. *Scientific American*, 270(2), 48-57.

Silvy, Y., Frölicher, T. L., Terhaar, J., Joos, F., Burger, F. A., Lacroix, F., ... & Ziehn, T. (2024). AERA-MIP: Emission pathways, remaining budgets and carbon cycle dynamics compatible with 1.5 °C and 2 °C global warming stabilization. *EGUsphere*, 2024, 1-47.

Author Rebuttals to First Revision:

We want to express our gratitude to the three reviewers who have provided very insightful comments that helped us to improve our manuscript. We have substantially revised our manuscript in response to their requests. We are particularly grateful to reviewer 4 and 5 for stepping in and supporting the review process.

We include a point-by-point response below with our responses being marked by green colour.

Referee #3 (Remarks to the Author):

I thank the authors for their detailed and thoughtful responses to my comments and suggestions.

I am content with the amendments made, recognising the limitations of word and reference counts, and would now recommend publication.

We thank the reviewer for the excellent comments that have helped us greatly to improve the manuscript.

I note one amendment which requires minor further clarification: on p11 the text now reads "We explore NNCE requirements for an illustrative pathway with the following characteristics

(Fig. 2a): (1) it achieves net -zero CO₂ around mid-century, (2) limits and aims to limit median peak warming close to 1.5°C above pre-industrial levels, and (3) has no NNCE." This appears contradictory (to explore NNCE requirements for pathways with no NNCE) although on reading further it becomes clear that the authors mean no 'initial' inclusion of NNCE. I trust this can be easily clarified.

We thank the referee for this observation and have revised the sentence to clarify the issue as follows:

"(3) requires no NNCE to do so (for the median warming outcome)."

Referee #4 (Remarks to the Author):

The referees' comments are mostly well addressed, but additional responses for the comment of Reviewer #1 on the term "substantially different" are needed. The authors provided some explanations in terms of statistical significance of regional changes, however it is still unclear how much changes were quantitatively observed from the analysis, and whether or not it is sufficient to claim "substantial".

In addition, even if the current explanation is sufficient to assert that the changes are substantially different in the regional level, the abstract still claims "global and regional climate change... are substantially different". The current responses look insufficient to support in terms of global changes anyway.

Additional explanation on these points are needed, or please consider rephrasing the words in the abstract.

We thank the referee for pointing this out. We can see that the inclusion of the word 'substantial' continues to be problematic. We would maintain the point that climate changes on the global level are substantial as illustrated e.g. in Fig. 4 a where we show that 50 years of overshoot would commit to almost as much sea level rise than what has been observed since the pre-industrial period. Of course, this does not in the same way apply to all climate indicators.

But to avoid confusion we have reworded the abstract removing the word "substantially".

Minor comments:

Please provide the definitions of the whiskers and outliers for the AR6 scenarios presented in Fig 2c. Is it 5-95th percentile range?

We thank the reviewer for pointing that out and we have revised Fig. 2c accordingly.

The link <https://zenodo.org/record/5886912> is provided as the source data for Fig. 2, but I found it links to the IPCC AR6 scenario database. Although I did not check the contents of this database carefully, does it really include the PROVIDE scenario data? Otherwise please provide a correct location.

We want to apologise for the oversight on our end (indeed a mistake) and have added the correct link to the scenarios on PROVIDE Zenodo repository:
<https://zenodo.org/records/6963586>

Temperature unit is presented in Kelvin in Fig4, whereas other figures and texts use degrees. Are there any reason?

We thank the reviewer for pointing that out and have adjusted the temperature unit in Fig. 4 to °C, too.

Referee #5 (Remarks to the Author):

Coming in as an additional reviewer, I attempted to follow both the previous reviewer's logic as well as the authors' responses to the best of my abilities. Please feel free to point out potential misconceptions on my end, yet I hope that a set of fresh eyes might be helpful after such a considerable revision.

That being said, the authors have made substantial changes to the manuscript in response to the reviewer's comment and made a huge effort to address all of the reviewer's concerns. Especially the inclusion of two additional model simulations as a response to referee #1 concerns on the regional reversibility experimental design are impressive, but maybe not the most relevant or scientifically meaningful.

We want to thank the referees for stepping in to support the review process, and the very thoughtful comments that certainly help us to further improve the manuscript. We also want to thank the reviewer to acknowledge that we have done substantial revisions in response to the comments by Ref #1.

My impression of the manuscript is that it consists of two interesting (but somewhat individual) storylines, and therefore fails to follow a red thread throughout (this might be related to the revision). I get this impression already since the argument structure and logic outlined in the abstract and the conclusion ...

(storyline 1: no perfect reversibility of climate impacts -> time lines of OS not the relevant metric for adaptation planning;

storyline 2: declining temperatures likely alleviate long-term/time-lagged climate impacts -> NNCE arise in case continued warming post net-zero -> there are techno-economic, social-political limitations to CDR deployment -> CDR should be foremost considered as preventive/back-up measure for possibility of high-risk futures)

... is different from the logic that the manuscript follows.

We thank the reviewer for this very thoughtful comment that clearly outlines the need for revisions to clarify the storyline and scope of our manuscript. We are focussing on the risks of overconfidence in overshoot portrayed as “another way” to achieve a desirable climate outcome (i.e. limiting warming to 1.5°C). In order to do so, we need to provide a comprehensive picture of the relevant dimensions of (omitted) uncertainties from emission pathways to adaptation needs. Step by step we illustrate the omitted uncertainties that lead to overconfidence in overshoot outcomes. I.e., we show how global temperature reversibility cannot be assumed. We then show that even if we assumed it, regional impact driver reversibility is not, etc.

However, we fully take the point by the reviewer that this is not clear enough from the current manuscript and have rewritten the respective part of the introduction section to increase clarity. We hope this addressed the referee’s concern.

In the following, we will outline the dimensions of overconfidence in overshoot from emission pathways to adaptation implications. We start by exploring the uncertainties in global temperature outcomes and their implications for the required net-negative CO₂ emissions to achieve the intended reversal of warming. Based on these insights we then discuss the consequences for mitigation strategies considering feasibility and sustainability constraints of deploying gigatonne-scale CDR. Yet, even if global temperatures were in decline, it is an open question if and how this translates into reversal of climatic impact drivers and subsequent impacts and risks. We provide insights into this question both for long-term regional climate changes under overshoot versus stabilisation, as well as for irreversible risks such as sea-level rise. Finally, we discuss the implications of considering or experiencing temperature overshoot for climate change adaptation. Based on this comprehensive perspective, we argue for redirecting the discussion towards reducing climate risks both in the near and long-term, and to avoid overconfidence in the controllability and desirability of climate overshoot.

This goes to the point of the referees made about there being internal inconsistencies in the logic concerning adaptation and impacts considerations in the manuscript. After reviewing the spatial reversibility results, I would suggest removing this part from the manuscript, since the evidence unfortunately is not there yet to discuss spatial reversibility of overshoot scenarios. I would strongly suggest that the authors refocus the manuscript and expand on the second storyline.

We thank the referee for outlining these considerations. However, we have difficulties to parse the comment with regards to the internal consistencies to the remarks made by previous referees. We however tried to revise the manuscript to address the internal inconsistencies pointed out by the referee. We consider the insights on the spatial reversibility as a key part of our manuscript and will discuss the referee's remarks in greater detail below.

Below, I am outlining where I think further adjustments might be necessary to address the reviewer's concerns, as well as some of my own comments.

Introduction:

Table 1: I find the two conceptual PD categories as described in the table rather vague, overlapping and it is unclear how and why they are/should be distinct from the ones already used in the literature. It is not specified what overshoot is referred to in the PD-OS (I am assuming temperature, but would argue that carbon budget would actually be the measure). It is not clear to me if the two categories are distinct through their near-term mitigation action, or the level of NNCE? When would we move from one category to the other. How is their narrative different from the highOS vs lowOS one? After reading the introduction, this becomes somewhat clearer, but this should be reflected in the table, which is meant as a help for the reader (i.e., mention maximum cumulative CO₂ emissions, and magnitude of NNCE, but again also when are we in one or the other category?).

We thank the reviewer for raising these concerns in terms of the conceptual clarity of the categories. They are indeed designed in temperature space and while the PD-OS category specifies a 'target temperature level' (and allows for overshoot), the PD-EP does not. We revised the table to clarify that PD-OS pathways do

"establish a target warming level" and "typically envision temperature stabilisation after overshoot."

In contrast PD-EP pathways would *"Given the timescales involved, these pathways typically do not reach an ultimate lower target temperature level within the scenario timeframe considered."*

As the PD are conceptual categories, they do not include quantitative definitions in a way that literature categories like low or high OS pathways do (which are included in second part of the table).

The relation of highOS and lowOS pathways to these categories is not quite as straightforward. Most high-OS and a number of low-OS pathways would fall into the PD-OS category, depending on when they achieve 1.5°C and whether or not they achieve long-term stabilisation. Only Paris Agreement compatible pathways would fall into the PD-EP conceptual category.

Fig. 1: Shading/Coloring in Fig 1 is confusing. From the four color shades on the right, the reader would look for four pathways, but the PD: Peak and decline pathway is a descriptive category of the other two PD pathways. And then, if I understand correctly, the PD-OS and the Pathways showing continuous warming are the same but with different assumptions on Earth system feedbacks. Which case is figure 1b discussing?

We thank the reviewer for this in-depth engagement with the figure and the very helpful comments. We have revised the visualisation to clarify the distinction between the pathway categories. We also revised Fig. 1b to clarify that it more broadly refers to outcomes under peak and decline pathways.

Regional climate change reversibility:

While I see the immense effort the authors have put into revising this section, I remain unconvinced of the analysis due to two points:

- 1) You are displaying and comparing one result each of two vastly different experimental designs, even though there is data for multiple models from both experiments. The biggest concern regarding this figure is the implicit comparison of ZECMIP and AREA-MIP results; the former being an idealised CO₂-only experiment

aiming for the same carbon budget (MacDougall et al., 2020), and latter being an fully-forced experiment, including non-CO2 GHGs, aerosols, and land-use changes, that diagnoses CO2 emissions trajectories for the same end of the century temperature outcome (Silvy et al., 2024, in review). Displaying and comparing spatial patterns of temperature and precipitation with two experiments one including, one excluding time and spatial-varying aerosol patterns, and finding that there are substantial differences is not scientifically sound.

We thank the referee for engaging with these new results in detail. However, it appears that the additional analysis and modelling results we included as part of the major revisions we did in response to referee 1 have led to a misunderstanding of the scope of the section.

1) Isolation of the impact of overshoot

The fundamental concern of referee 1 was to “*isolate the impact of the overshoot*” from transient changes long-term changes. Referee 1 further stated that:

“What would be urgently needed is (one or several) scenario pairs achieving the same 2100 temperature (or carbon budget) outcome with no/limited overshoot and with overshoot. Such scenario pairs could be compared in a meaningful way to investigate the impacts of an overshoot relative to a world that avoided overshoot.”

In response to this comment, we have been able to identify two complementary, dedicated modelling experiments that allow us to address this very question whether regional climate changes under an overshoot scenario are different from those emerging under long-term global temperature stabilisation.

What we find is that in two different models and different modelling setups (one GMT focussed, extended AERA-MIP experiment, and carbon budget overshoot experiment in emissions space extending the ZECMIP protocol), we can identify robust regional climate differences between an overshoot and long-term stabilisation scenario over multi-decadal to centennial timescales. The robustness of this result despite the differences in modelling setups gives us further confidence in the conclusion that we

need to expect overshoot futures to be different from long-term stabilised climates even beyond the cessation of the overshoot. We consider this a major insight.

The purpose of our analysis is *not* to compare the AERA-MIP and ZECMIP based modelling results and the referee is right in pointing out that such a comparison of differences between the experiments would not be “scientifically sound.”

Our main insights lay in the qualitative similarities in terms of the regional climate changes between the two experiments, not their differences.

But we of course very much see the need to avoid any impression of an unduly comparison between the two experiments and have revised the manuscript and Fig. 3 accordingly.

2) Use of two different experimental setups

To us, the use of two different very modelling protocols, one exploring an overshoot in emissions space, one in temperature space, can support the robustness of the insights. And indeed, while we find pronounced differences in the quantitative response, the results yield qualitatively similar results with respect to key features under overshoot. Also the CMIP6 based analysis included in Extended Fig. 6 and 7 qualitatively confirms some general findings. To us, this provides additional confidence in our insights.

We also want to highlight that the simulations used by us are, to the best of our knowledge, a unique set of simulations going well beyond the default AERA-MIP and ZECMIP protocols.

While the setup of the stabilization scenario performed by NorESM2-LM follows the design of ZECMIP, the associated overshoot scenario is an addition developed under the Norwegian IMPOSE project. For the analysis provided in the manuscript, both

simulations are required. To date, NorESM2-LM is the only Earth System Model that has performed these simulations.

Similarly, the GFDL-ESM is the only ESM that, extending the AERA-MIP protocol, has performed stabilisation and overshoot experiments up to the year 2500. This is crucially required to investigate the stabilisation vs. overshoot question.

A multi-model comparison for different ESMs under the same protocol is thus not possible (yet).

3) The importance of non-CO2 and aerosol forcing

We fully agree with the referee that full consideration must be given to changes in non-CO2, aerosol forcings as well as land use changes when exploring overshoot questions. We also fully acknowledge the differences in the modelling experiments in that regard and are discussing these implications for the CMIP6 results shown in the Extended Figures.

We note that the GFDL overshoot and stabilisation scenario use the same aerosol forcing throughout and regional climate differences between the two experiments thus cannot be explained by differences in aerosol forcing.

We also note that non-CO2 forcing, as well as land-use and land cover changes in the AERA-MIP experiments (GFDL-ESM2M) follow the RCP2.6 scenario over the 21st century and are being held constant after the year 2100.

Fig.1: Non-CO₂ emissions (CO₂ forcing equivalent) in both experimental protocols (adapted from Silvy *et al* 2024 Fig. A1).

Given their short lived nature, we would therefore expect very limited to no influence of aerosols (and changes in land use change and non-CO₂ forcing agents) on the long-term climate outcomes up to 2500. The long-term climate outcome is therefore dominantly driven by fossil fuel CO₂ emissions only, as is in the NorESM simulations, which are by definition fossil fuel CO₂ emissions only.

We thus find the experiments well suited to explore the long-term imprint of overshoots on regional climate compared to long-term climate stabilisation 200 years after peak warming.

2) None of the experiments actually includes the impact from large-scale CDR deployment within their scenarios. An argument that is lacking throughout the entire manuscript, but would be one key to linking these analysis and results to the rest of the manuscript.

We fully agree that this is a critical point to highlight and we note the importance to consider the biophysical climate consequences including the dynamical response of the climate system to land cover changes (compare e.g. (De Hertog *et al* 2023)). However, the climate impacts of large-scale CDR strongly depend on the type of CDR deployed, and the regional distribution. However, we note that there are CDR options with limited to non biophysical climate response such as Direct Air Capture with CCS.

To explore the different biophysical climate consequences of different CDR deployment strategies, however, would constitute a full study on its own and is beyond the scope of the analysis presented here.

We, however, fully agree with the need to clarify this point in the manuscript presented and have done so as follows:

In the main manuscript:

We also note the importance of biophysical climate feedbacks of land-cover changes associated with large-scale land-based CDR deployment (Table ED1) that could be explored in such experiments.

And in the methods section:

*We note that none of the two protocols includes land cover changes beyond the reference pathway. This points to an implicit assumption that the additional CDR in these simulations is achieved using technical options with little to no land footprint such as Direct Air Capture with CCS (Extended Data Table 1). If the amount of CDR was to be achieved using land-based CDR methods, however, we would expect pronounced biophysical climate effects from the land cover changes alone (de Hertog *et al*, 2023). Exploring the regional climate differences resulting from different CDR strategies would be useful in future modelling efforts.*

In response to the referee's comment we have also added a new Extended Fig. 4 showing the temporal and cumulative evolution of emissions for both experiments.

4) Other lines of evidence

We also want to highlight that we also include an analysis of overshoot outcomes in CMIP6 ScenarioMIP pathways based on a much larger set of ESMs that we present in the extended figure section (see extended Figure 6 and 7). While these do not allow for conclusive analysis on overshoot impacts due to the overlap with stabilisation and aerosol removal effects (noting these are limited up to 2100), they qualitatively support some of the key features of overshoot pathways we document.

Adaptation decision-making and overshoot:

Figure 5: Based on the points raised by one of the referees, I think the usefulness of this illustrative figure remains to be substantially improved. It remains unclear what the horizontal bars actually stand for. The authors clearly have a basis for the length of the different bars, so why not provide the reader with this information (i.e. label the bars) rather than giving a repetition of the y-label axis. In the same way, a label for the time scales displayed (i.e. a xlabel that shows years, decades, centuries) for figure b would be helpful for the reader.

I also suggest to re-label the first bars to “Median overshoot length and magnitude” along the logic of the manuscript.

For Figure 5a, I think displaying some form of probabilities of the two scenarios would be helpful for linking this figure to the adaptation discussion of high-risk/low-probability events (see comment below).

We thank the reviewer for these very thoughtful comments that we have implemented in a revision of the figure.

For adaptation decision making high-risk/impact-low-probability outcomes are considered, so I would argue that next to time scales it is also the level of peak warming that is more relevant, i.e. this is where the peak warming level is more important than long-term temperature outcomes. I am missing the notion of such considerations in this

section all-together, even though I find them highly relevant for the remaining arguments made in the manuscript including not betting on median outcomes.

We thank the reviewer for this pertinent point that we fully agree with.

In fact, an earlier version of this manuscript included some dedicated modelling that illustrated how peak warming is more important than the long-term expectation. We removed this due to space constraints, but have now reinserted the notion of the importance of peak warming throughout the section.

The other aspect that is relevant for the strength of climate impacts and according adaptation is the rate of change (e.g., William et al., 2021), I would like to invite the authors to take this into consideration in the manuscript, since it would again align very nicely with the storyline and outline what kind of impacts are relevant for overshoot considerations.

Another excellent comment that we are in full agreement with and are happy to incorporate in full. We also note that for limits to adaptation, the pace of climate change fundamentally matters and have made a remark in that respect, too.

Reframing the overshoot discussion:

The innovative conclusion of the manuscript of the development of a sustainable CDR capacity that is independent of the discussion of remaining emissions, comes from my understanding mostly from the possibility that declining global mean temperatures are likely beneficial to the lagged climate impacts and the analysis of the NNCE needs using the FAIR climate model. This suggestion is interesting, yet the authors do then fail to discuss the actual implications this suggestion would have. The discussion of what an environmental (and maybe also societal) level of sustainable CDR would entail, how the authors would define sustainable, or how to get to the information on sustainable levels of CDR and the associated potentials.

We thank the referee for this very encouraging reflection on what core insights from our manuscript are in their view. We have included an analysis of this in the section on

“Relying on Carbon Dioxide Removal”. The need for such a proposed preventive CDR capacity, however, is linked to physical climate uncertainties and the risk of a high warming outcome that needs to be hedged against. Such a concept has profound implications for how we view CDR in emission reduction strategies that we also discuss in the manuscript:

The need for a preventive capacity has implications for the design of stringent emission reduction pathways in light of constraints that limit overall CDR deployment. Pathways relying on large amounts of CDR to even achieve net-zero CO₂ often exhaust or exceed sustainability limits by design, leaving little to no room for course adjustments in case of high warming outcomes. On the other hand, pathways that do not plan for future development of CDR may fail to build up the technological solutions required to establish a preventive CDR capacity, thereby exposing future generations and in particular most vulnerable communities to risks that could at least be partly hedged against. Incorporating preventive CDR in pathway design requires further reflection, including regarding risks and policy design, but also about how to assign responsibilities and incentivise different actors for providing for this preventive CDR capacity.

Also when talking about adaptation and time scales, I am missing this information wrt the environmental sustainable, preventive CDR pool. Where in figure 5 would this pool lie? Would this CDR pool be deployed in a timely enough fashion to hedge against impacts, given the current deployment gap?

We thank the referee for this comment and the link back to the adaptation considerations. The preventive CDR pool would be required to hedge against high risk outcomes in which warming reversibility is not a given as we discuss in section 2 of the manuscript. It thus comes “on top” the considerations put forward when discussing adaptation and overshoot further underscoring that betting on overshoot is not a resilient adaptation strategy. We have revised the manuscript to draw these links more explicitly to include the following:

“However, as we have shown above, long-term global temperature decline is nothing that can be relied on with certainty. Thus, a resilient adaptation strategy cannot be

based on betting on overshoot and only limiting peak warming can effectively reduce adaptation needs.”

Lastly, I feel like there is a disconnect between the title and the content of the manuscript including the main conclusions (i.e. preventive CDR pool). I would like to invite the authors to revisit the focus of the manuscript and then accordingly adjust the title.

We thank the authors for this comment that is linked to a comment above on the main narrative. As we have outlined above, and hopefully clarified in the manuscript, too, exploring the dimensions of overconfidence in overshoot is the main narrative connecting the elements of the manuscript. We thus feel that the current title is adequate in capturing the full narrative, and would prefer to maintain it.

References

De Hertog S J, Havermann F, Vanderkelen I, Guo S, Luo F, Manola I, Coumou D, Davin E L, Duveiller G, Lejeune Q, Pongratz J, Schleussner C-F, Seneviratne S I and Thiery W 2023 The biogeophysical effects of idealized land cover and land management changes in Earth system models *Earth Syst. Dyn.* 14 629–67 Full Editor email and response

Silvy Y, Frölicher T L, Terhaar J, Joos F, Burger F A, Lacroix F, Allen M, Bernadello R, Bopp L, Brovkin V, Buzan J R, Cadule P, Dix M, Dunne J, Friedlingstein P, Georgievski G, Hajima T, Jenkins S, Kawamiya M, Kiang N Y, Lapin V, Lee D, Lerner P, Mengis N, Monteiro E A, Paynter D, Peters G P, Romanou A, Schwinger J, Sparrow S, Stofferahn E, Tjiputra J, Tourigny E and Ziehn T 2024 AERA-MIP: Emission pathways, remaining budgets and carbon cycle dynamics compatible with 1.5 °C and 2 °C global warming stabilization *EGU sphere* 1–47

Reviewer Reports on the Second Revision:

Referee #5 (Remarks to the Author):

The author team has done an excellent job addressing my previously raised concerns. I think the clarity of the purpose of the manuscript scope has been much improved, and the limitations and future research needs are now more clearly outlined.

The only part of the manuscript that reads a bit weird now is a the following section, in which there is a direct contradiction within two sentences:

"We note the substantial differences in the modelling protocols, which means results are not directly comparable. However, despite these differences, we find similarities in the overshoot vs. stabilisation regional patterns emerging in both modelling simulations, which gives us confidence in the robustness of our findings."

I suggest to rephrase, maintaining the disclaimer about the limitation of results given their different modelling protocols and overshoot targets. Maybe towards: "Interestingly, despite these differences, we find some features within the overshoot vs. stabilisation regional patterns emerging in both modelling simulations, that warrant further analysis." And skip the part on robustness of your the findings.

After this section has been revised, I would recommend publication.

Author Rebuttals to Second Revision:

Referee #5 (Remarks to the Author):

The author team has done an excellent job addressing my previously raised concerns. I think the clarity of the purpose of the manuscript scope has been much improved, and the limitations and future research needs are now more clearly outlined.

The only part of the manuscript that reads a bit weird now is in the following section, in which there is a direct contradiction within two sentences:

"We note the substantial differences in the modelling protocols, which means results are not directly comparable. However, despite these differences, we find similarities in the overshoot vs. stabilisation regional patterns emerging in both modelling simulations, which gives us confidence in the robustness of our findings."

I suggest to rephrase, maintaining the disclaimer about the limitation of results given their different modelling protocols and overshoot targets. Maybe towards:

"Interestingly, despite these differences, we find some features within the overshoot vs. stabilisation regional patterns emerging in both modelling simulations, that warrant further analysis." And skip the part on robustness of your the findings.

After this section has been revised, I would recommend publication.

We thank the reviewer for the excellent comments that have helped us to improve the manuscript. We are happy to learn that our revisions have addressed the referee's concern.

With regard to the sentence at question, we are happy to implement the referee's suggestion with some small modifications (mostly that the part on 'further analysis' fits better further down in this section).

It now reads:

Despite these differences in the modelling protocols, we find some features within the overshoot vs. stabilisation regional patterns emerging in both modelling simulations.